# Rain-on-Snow (ROS) Understudied in Sea Ice Remote Sensing: A Multi-Sensor Analysis of ROS during MOSAiC

Julienne Stroeve[1,2,3], Vishnu Nandan[1], Rosemary Willatt[3], Ruzica Dadic[4,5], Philip Rostosky[6], Michael Gallagher[7], Robbie Mallett[3], Andrew Barrett[2], Stefan Hendricks[8], Rasmus Tonboe[9], Mark Serreze[2], Linda Thielke[6], Gunnar Spreen[6], Thomas Newman[3], John Yackel[10], Robert Ricker[11], Michel Tsamados[3], Amy Macfarlane[5], Henna-Reetta Hannula[12], and Martin Schneebeli[5]

[1]University of Manitoba, Centre for Earth Observation Science (CEOS), Winnipeg, Canada
[2]University of Colorado, Cooperative Institute for Research in Environmental Science (CIRES), National Snow and Ice Data Center (NSIDC), Boulder, CO, USA
[3]University College London, United Kingdom
[4]Victoria University of Wellington, Antarctic Research Centre, New Zealand
[5]WSL Institute for Snow and Avalanche Research SLF, Davos, Switzerland
[6]University of Bremen, Institute of Environmental Physics, Bremen, Germany
[7]University of Colorado, Cooperative Institute for Research in Environmental Science (CIRES), National Oceanographic and Atmospheric Administration (NOAA) Physical Sciences Laboratory, Boulder, CO, USA
[8]Alfred Wegener Institute, Bremerhaven, Germany
[9]Technical University of Denmark, Copenhagen, Denmark
[10]Department of Geography, University of Calgary, Canada
[11]Norwegian Research Centre, Tromsø, Norway
[12]Finnish Meteorological Institute, Helsinki, Finland

**Correspondence:** Julienne Stroeve (Julienne.Stroeve@umanitoba.ca)

**Abstract.** Arctic rain-on-snow (ROS) deposits liquid water onto existing snowpacks. Upon refreezing, this can form icy crusts at the surface or within the snowpack. By altering radar backscatter and microwave emissivity, ROS over sea ice can influence the accuracy of sea ice variables retrieved from satellite radar altimetry, scatterometers, and passive microwave radiometers. During the Arctic Ocean MOSAiC Expedition, there was an unprecedented opportunity to observe a ROS event using *in situ* active and passive microwave instruments similar to those deployed on satellite platforms. During liquid water accumulation in the snowpack, there was a four-fold decrease in radar energy returned at Ku- and Ka-bands. After the snowpack refroze and ice layers formed, this decrease was followed by a six-fold increase in returned energy. Besides altering the radar backscatter, analysis of the returned waveforms shows the waveform shape changed in response to rain and refreezing. Microwave emissivity at 19 and 89 GHz increased with increasing liquid water content and decreased as the snowpack refroze, yet subsequent ice layers altered the polarization difference. Corresponding analysis of CryoSat-2 waveform shape and backscatter as well as AMSR2 brightness temperatures further shows the rain/refreeze was significant enough to impact satellite returns. Our analysis provides the first detailed *in situ* analysis of the impacts of ROS and subsequent refreezing on both active and passive microwave observations, providing important baseline knowledge for detecting ROS over sea ice and assessing their impacts on satellite-derived sea ice variables.

## 1 Introduction

Over the past 50 years, the Arctic has warmed three times faster than the planet as a whole [AMAP, 2021]. While this amplified Arctic warming is most strongly manifested in autumn as a result of summer sea ice loss [e.g. Serreze et al., 2009], recent studies have also documented an increase in frequency and duration of winter warm spells [Graham et al., 2017]. As a result, there is some evidence that Arctic rain-on-snow (ROS) events are becoming more common [Meredith et al., 2019; Liston and Hiemstra, 2011]. A recent study further suggests Arctic precipitation will increase more strongly that previously projected through 2100, with an earlier transition from snow to rain [McCrystall et al., 2021]. When rain falls on snow, it can dramatically alter snowpack properties such as snow density, grain size, and snow water equivalent (SWE) content [Langlois et al., 2017; Grenfell and Putkonen, 2008]. Upon refreezing, ice layers may form within the snowpack. On land, ROS exacerbates flooding [Li et al., 2019], increases soil temperatures and snowmelt [Westermann et al., 2011; Putkonen and Roe, 2003; Rennert et al., 2009], while subsequent icy layers can impact wildlife, such as seal denning [Sterling and Smith, 2004], or inhibit foraging, leading to massive mortality of caribou, reindeer, and musk oxen [Putkonen et al., 2011; Forbes et al., 2016].

As a result of the ecological and societal importance of these events, several remote sensing techniques have been developed to detect ROS over land using active and passive microwave sensors. Wet snow causes backscatter to decrease, as the presence of liquid water increases the dielectric permittivity, and thus signal absorption [Kim et al., 1984; Webb et al., 2021], while refrozen ice layers lead to a strong increase in backscatter [Mortin et al., 2014]. Since the backscatter difference between melt and refreeze periods is generally larger than day-to-day variations, a threshold on temporal backscatter variability can reveal when ROS has occurred [Bartsch et al., 2010]. For passive microwave emissions, liquid water in the snowpack increases emissivity [Vuyovich et al., 2017]. The dependence is stronger at lower frequencies because of the change in emission depth associated with melt. The response is also polarization dependent; horizontal channels exhibit stronger responses [Anderson, 1997]. Taking advantage of these emissivity dependencies, Dolant et al. [2016] developed a detection algorithm based upon vertical (V) and horizontal (H) brightness temperature gradient ratios between 19 and 37 GHz.

ROS detection over sea ice has by comparison been largely unexplored. However, winter ROS events over sea ice can also generate ice layers at the surface or within the snowpack that could modify emitted and backscattered radar energy used to retrieve various sea ice geophysical variables, such as sea ice concentration, ice thickness, snow depth and the timing of melt onset and freeze-up. Voss et al. [2003] explored the error induced from ROS and subsequent refreezing on ice type classification using passive microwave and scatterometer data. Despite this study, detection of ROS over sea ice to date has been based on partitioning of precipitation phase from atmospheric reanalysis. For example, Dou et al. [2021] evaluated reanalysis products to track changes in spring ROS events over sea ice and conclude that they are occurring earlier than they did four decades ago.

An issue particular to sea ice is that surface crusts or ice layers in the snowpack can result in an apparent vertical upward shift in the dominant scattering surface surface, affecting retrievals of sea ice elevation and hence sea ice thickness. While ROS events have not been specifically researched with regards to their influence on radar altimetry, Willatt et al [2010] showed that morphological features including ice layers in snow-covered Antarctic sea ice reduced the probability that the snow/ice interface was the dominant scattering surface at Ku-band, while King et al. [2018] found ice lenses altered ice thickness

retrievals by a factor of two in the Arctic. Nandan et al. [2020] demonstrated a two-fold difference in dielectric constant between snow and refrozen ice layers, leading to an elevated peak that accounted for 15% of the total simulated Ku-band return power. Besides generating an ice layer upon refreezing, ROS further impacts the overall surface roughness of the snowpack [Seifert and Langleben, 1972]. Combined, these snow property changes alter radar backscatter. Given that currently all CryoSat-2 radar altimeter derived sea ice thickness data products assume the dominant scattering surface is the snow/ice interface, the presence of liquid water, changes to snow structure and/or ice layers resulting from a cold-season ROS event could bias thickness retrievals. Without accounting for this effect, derived snow depth obtained using a combination of SARAL/AltiKa (Ka-band) and CryoSat-2 (Ku-band) radar freeboards [e.g. Guerreiro et al., 2016], or CryoSat-2 radar freeboards with ICESat-2 (laser altimeter) snow freeboards [e.g. Kwok et al., 2020] would be similarly biased.

ROS events may also lead to distinct changes in microwave emissivity and backscatter that affect melt-onset timing derived from passive sensors (e.g. AMSR2) or Ku-band scatterometers (e.g. QuickSCAT). Furthermore, retrievals of ice concentration, ice type, snow depth, and thin ice thickness from passive microwave radiometers spanning L-band to 89 GHz may be similarly impacted. Voss et al. [2003] already demonstrated that increased surface roughness following a melt-refreeze period alters the emitted microwave energy, resulting in errors in retrieved first-year ice (FYI) and multiyear ice (MYI) fractions.

During the 2019-2020 Multidisciplinary drifting Observatory for the Study of Arctic Climate (MOSAiC) [Krumpen et al., 2020], a suite of surface-based, active and passive microwave systems was deployed to monitor the seasonal evolution of snow and ice properties, microwave backscatter, and emissivity. Sensors spanned 0.5 to 89 GHz and were deployed at the same location with overlapping footprints. Here, we present the first observations of a ROS event on sea ice observed by an *in situ* Ka/Ku-band radar, together with coincident observations from a microwave radiometer at 19 and 89 GHz. By taking advantage of this unique opportunity of having several remote sensing instruments deployed at the same time, we provide key baseline knowledge for developing ROS retrieval algorithms over sea ice and understanding ROS impacts on retrieved sea ice variables such as snow depth, ice thickness, and sea ice concentration.

## 2 Instrumentation and Data

The radar and radiometer were deployed on a large ice floe at the dedicated MOSAiC Remote Sensing (RS) site. The location of the RS site relative to R/V Polarstern and MET City (where the 12-m tall meteorological tower was installed), as well as locations for the snow pit sampling during Leg 5 (21 August - 20 September 2020) are shown in **Figure 1**. The ice thickness near the RS site was about 1.4 m. Details of the remote sensing site setup can be found in Nicolaus et al. [2021].

### 2.1 Surface-based Dual-Frequency Radar (KuKa radar)

The Ka- and Ku-band fully-polarimetric (VV, HH, HV and VH), surface-based radar (KuKa radar) was a newly-built system by ProSensing Inc and designed for polar deployments [see Stroeve et al., 2020]. FMCW (Frequency Modulated Continuous Wave) radars, such as the KuKa radar, are particularly suited to applications requiring high range resolution for which the observed range interval is small, and where the high processing gain allows low peak transmit powers. The KuKa radar transmits

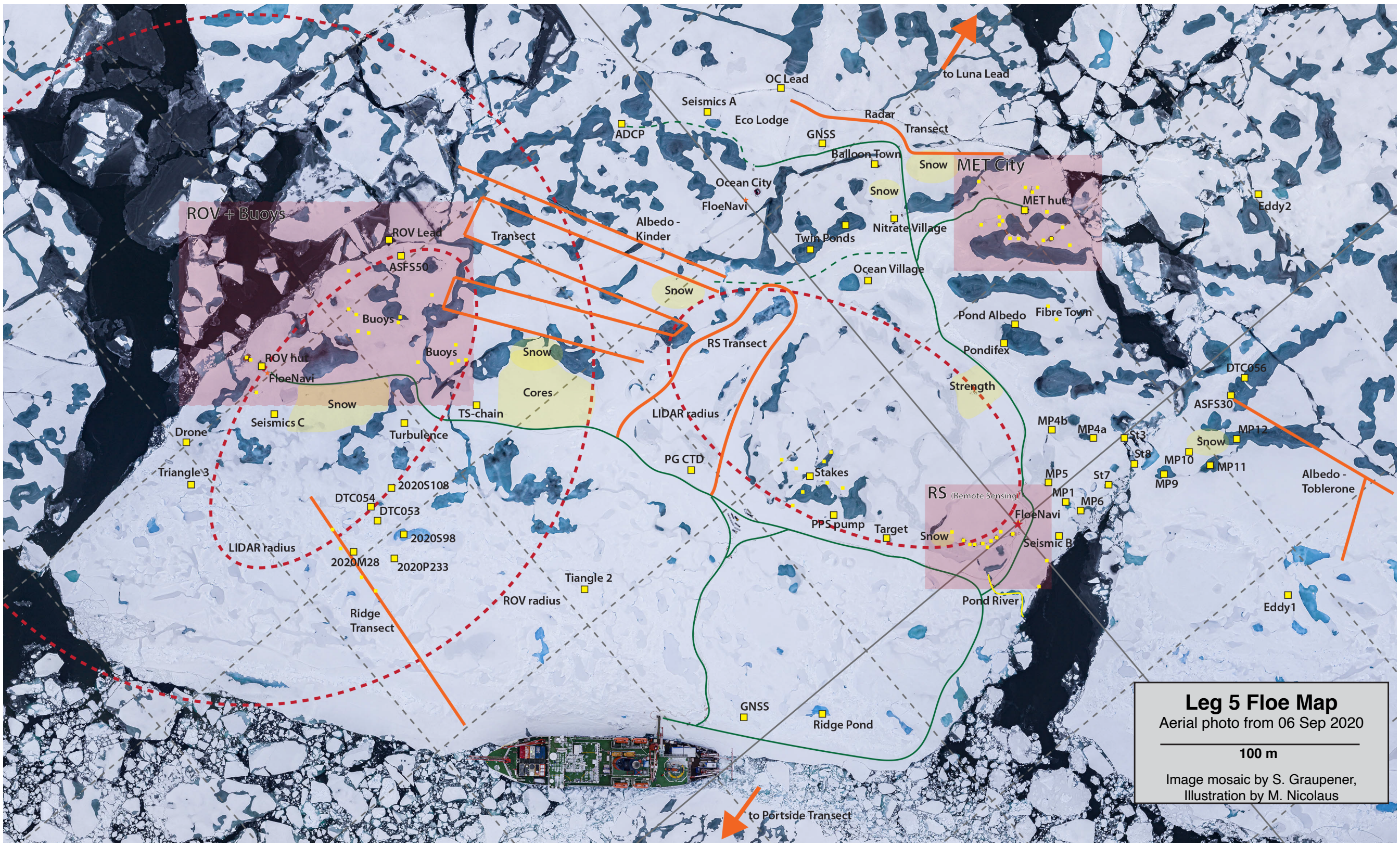

**Figure 1.** Aerial photo showing the location of R/V Polarstern (bottom center) relative to the position of Remote Sensing (RS) site (middle-right of image). Also shown are the locations of MET City where the meteorological observations used in this study were obtained (upper-right of image), as well as the Remote Sensing and the albedo transects. MET City was ∼100 m from the RS site. The floe was located at roughly 89°N/109°W. Snow pit areas are highlighted in yellow. RS and MET City as well as the ROV and buoy sites are highlighted in pink.

Ku-band waves at 12-18 GHz and Ka-band waves at 30-40 GHz, with bandwidths of 6 and 10 GHz, respectively (**Table 1**). The KuKa system uses separate transmit and receive antennas for both Ka- and Ku-bands that allows simultaneous transmission and reception of signals, and a 100% duty cycle. Deconvolved data are used to reduce sidelobes caused by chirp non-linearity (see Appendix for details on the deconvolution).

The KuKa radar was deployed during MOSAiC Legs 1-2 and 4-5, and operated both as a scatterometer (scan mode) and as an altimeter (nadir mode). In scan mode, the radar was mounted on a pedestal, scanning hourly along an azimuthal width of 60$^o$ (between -30$^o$ to +30$^o$ azimuth range), and from nadir to a viewing angle ($\theta$) of 60$^o$, in 3$^o$ increments. During nadir mode, the instrument was fixed to look at nadir and was towed along a transect. During MOSAiC, KuKa was configured to match the center frequencies of AltiKa at 35 GHz (Ka-band) and CryoSat-2 at 13.575 GHz (Ku-band). Compared to satellite systems, KuKa has significantly higher bandwidth, providing improved range resolution of 1.5 cm (Ka-band) and 2.5 cm (Ku-band),

Table 1. Operating parameters for each of the KuKa radar's radio frequency (RF) units.

| Specification | Ku-band | Ka-band |
|---|---|---|
| RF Output Frequency | 12-18 GHz | 30-40 GHz |
| Two-way beamwidth (Center Frequency) | 16.9° at 13.575 GHz | 11.9° at 35 GHz |
| Range resolution | 2.5 cm | 1.5 cm |
| Azimuth range | 60° (-30° to +30°) | |
| Incidence angle (steps) | 0° - 60° (3°) | |
| Noise Floor (Stroeve et al., 2020) | -70 dB (VV, HH)<br>-80 dB (HV, VH) | -90 dB (VV, HH, HV and VH) |

compared to 32 and 47 cm for AltiKa and CryoSat-2, respectively. On 16 January 2020, an external trihedral corner reflector calibration was conducted during Leg 2 at the RS site, positioned ∼ 10 m in the antenna's far field region.

In this study, we only evaluate the data from the scan mode while the instrument was deployed at the RS site, which also includes nadir scans. **Table A1** lists the measurement periods of KuKa during Leg 5. A total of 559 scans were obtained between 25 August and 19 September 2020. A data gap exists between 08:50 and 13:00 UTC on September 13 as a result of a power outage. We focus our analysis on the normalized radar cross-section (NRCS) values at nadir and 45°, mimicking $\theta$ of satellite radar altimeters and microwave scatterometers. NRCS (also termed sigma0), is given in decibels (dB) and describes the radar backscatter as a function of antenna range to the scattering particles and snow physical properties, and is frequency, polarization, and $\theta$ dependent.

One of our objectives is to better understand the implications of ROS on altering freeboard height retrievals and thus radar altimeter-derived ice thickness and snow depth. Therefore, we also examine radar-return waveform changes at nadir, including the dominant scattering surfaces observed before, during and after the ROS event. Since KuKa has a beam-limited footprint and operates at 1.5 m above the surface, as opposed to CryoSat-2 and AltiKa, which are pulse-limited footprints at ∼800 km above the surface, the geometries are different and waveform shapes and their changes are not directly comparable. Nevertheless, changes in relative backscatter contributions from surfaces, interfaces, and volumes as well as changes in waveform shape enable inference of potential impacts on retrieved freeboards. For this study, HH-polarized data, acquired at nadir, and averaged over scans from -30° to -10° (Ka) and 0° to 30° (Ku) azimuth angles were used. The full azimuth range was reduced in post-processing to avoid snowdrifts around the sled. See Stroeve et al. [2020] and the Appendix for more details on the KuKa instrument and data processing.

**Table 2.** System parameters of Radiometrics Surface Based Radiometer (SBR)

| | **AC1900 19 GHz** | **AC3700 37 GHz** | **AC8900 89 GHz** |
|---|---|---|---|
| Center Frequency (GHz) | 19.0 | 37.0 | 89.0 |
| Edge-to-edge IF bandpass (GHz) | 1 | 2 | 4 |
| Antenna HPBW (degrees) | 6 | 6 | 5.88 |
| Sidelobes (-dB) | -23 | -22 | -15 |
| Noise Figure (dB) | 3.8 | 3.5 | 6.85 |
| Sensitivity (K) | 0.04 | 0.03 | 0.08 |
| Receive Noise (K) | 390 | 360 | 1130 |
| Antenna Type | Lensed | Lensed | Corrugated Horn |
| Size (cm) | 51 x 23 x 13 | 30 x 22 x 10 | 25 x 22 x 9 |
| Weight (kg) | 8.6 | 5.9 | 4.2 |
| Radiometer receiver architecture | Direct detection | Direct detection | Double sideband downconvert |

### 2.2 Surface-based Radiometer (SBR)

A surface-based dual-polarized radiometer (SBR) built by Radiometrics Corporation collected microwave emission at 19, 37, and 89 GHz and at V and H polarizations. These radiometers contain the same frequencies and polarizations as flown on past and current satellite systems (e.g. SSMI, AMSR2). Internal sensitivities at 19, 37 and 89 GHz are 0.04, 0.03 and 0.08 K, respectively [Radiometrics, 2004]. Each antenna system has a 3 dB half power beam width (HPBW) of 6 degrees (**Table 2**). Polarization switches are operated at $\pm$ 45 degrees from the central position, with an isolation better than 20 dB.

During MOSAiC, the instrument was mounted to a scanning positioner that allowed the user to scan across viewing zenith angles ($\theta$). This positioner was attached to a 80 cm metal tube and fixed to a 30 cm high komatik sled, giving a field-of-view (FOV) at nadir of about 5.7 cm. To avoid seeing the sled, the SBR scanned the surface between $\theta$ = 40 and 70$^{o}$ in steps of 5$^{o}$. Sometimes the instrument was configured to only collect data at $\theta$ = 55$^{o}$, the viewing angle of many satellite platforms. The FOV at this angle is 10.1 by 17.6 cm.

To calibrate the SBR during field work operations requires two temperature blackbody target calibrations. This can be achieved by using an ambient blackbody target and a liquid nitrogen target, or a sky-calibration target (e.g. pointing the instrument to an elevation of 120$^{o}$). Calibration of each 19 and 37 GHz radiometer was performed using a 0.5" thick Lossy Flexible Foam absorber, whereas the 89 GHz channel was calibrated using a 0.5" thick radar absorber foam. Physical temperatures were also made of the absorber pads and the ambient air temperature. Comparison of the slope of the radiometer measurements to the physical measurements provides for a calibration of the brightness temperatures. A calibration was performed during Leg

5 on the 9th of September. However, for 89 GHz, results of the absorber calibration were unstable and calibration coefficients from Leg 3 were used instead.

Output is given as brightness temperature (Tb) in Kelvin (K), which is the product of the effective emissivity ($\epsilon$) and the absolute temperature (T). **Table A2** lists the measurement periods of the SBR during Leg 5. As for KuKa, a power outage to the SBR instrument created a data gap between 12-13 September. In this study we only show results at $\theta$ = 55$^o$ as this is the most relevant viewing zenith angle for current satellite missions. Note, the 37 GHz channel was unfortunately unstable and thus we only focus on results at 19 and 89 GHz.

### 2.3 Meteorological Data and ERA5 Atmospheric Reanalysis

We analyze 2 m air ($T_{air}$) and skin temperatures from the 10-meter high meteorological tower at MET City [Shupe et al., 2021], as well as precipitation rates derived from an on-ice OTT Pluvio pluviometer [Bartholomew, 2020] provided by the Atmospheric Radiation Measurement (ARM) program, also deployed at MET City. The Pluviometer measures precipitation falling on a 400 cm$^2$ area with an accuracy given by the manufacturer of 0.1 mm/min, but cannot distinguish between precipitation types. Size distribution of the falling particles was captured by the Parsivel (on-ice disdrometer), and vertical velocity from the Ka-band ARM zenith radar (KAZR; deployed at the bow of the ship). Further details on deployment of these instruments and their accuracy can be found in Wagner et al. [2021]. While hydrometeor classification is difficult, near-surface air temperature, combined with vertical profiles of falling speed, can be used to categorize liquid phase precipitation [Shupe et al., 2005]. To correspond with scan intervals from KuKa, we produce hourly averages from the 1-second interval meteorological data.

The above mentioned meteorological data are used to interpret the surface property changes for the ROS event, yet there are many other *in situ* observations available beyond what is presented here. A comprehensive overview of the wide range of observations can be found in Wagner et al. [2021]. Much of these data were also analyzed for this study but will not be included to remain concise. From the available data, atmospheric soundings also provide important context for the thermodynamic structure of the atmosphere and are briefly discussed but not shown. Instead, the suite of meteorological and precipitation data presented here allows for straightforward interpretation of how modification of surface properties impacted the remote sensing signatures.

In order to expand our study beyond the time-period of this particular ROS event and to examine how wintertime ROS may be changing over time, we additionally make use of precipitation amount and type from ERA5 reanalysis [Hersbach et al., 2018]. ERA5 was previously found to agree well with precipitation measurements from north pole drifting station data [Barrett et al., 2020], as well as with snowfall timing and accumulation obtained during MOSAiC [Wagner et al., 2021]. ERA5 is used here to assess trends in cumulative, cold-season, non-frozen precipitation over the Arctic Ocean between 1980 and 2020 inclusive. This is done by first identifying and removing frozen precipitation (snow and ice pellets), leaving rain, freezing rain, wet snow and mixed rain/snow. We further remove precipitation falling at a rate $<$ 0.1 mm/day, since ERA5 forecasts include trace precipitation most days [Barrett et al., 2020]. Results are then time-averaged only where rain falls in grid cells that contain sea ice (concentration $>$ 50%). Results are spatially averaged over the Arctic Ocean, and individually over all regions of the

Arctic Ocean that contain sea ice. Regional averages are computed for regions as defined by the National Snow and Ice Data Center (https://nsidc.org/data/polar-stereo/tools_masks.html#region_masks).

### 2.4 Snow Data

Routine snow pit observations were collected at representative locations around the MOSAiC floe (see **Figure 1** for locations). Density cutters (volume = 100 $cm^3$) were used to measure snow density at 3 cm vertical resolution. Needle-point thermometers recorded temperature at the snow surface, snow/ice interface and at 5 cm intervals in between. SWE was measured using the ETH-SWE tube [Haberkorn, 2019], weighed with a calibrated spring scale. Snow depth was measured during SWE sampling.

Snow samples collected using the density cutter were bagged and melted to room temperature for measuring salinity (ppt) using a YSI30 conductivity sensor (resolution = 0.1 ppt and accuracy = ± 2% or ± 0.1 ppt). A micro-computed tomograph (micro-CT) (Micro-CT 45/90 computer tomography scanner from Scanco Medical AG) measured the 3-D snow structure [Hagenmuller et al. 2016]. Using the snow microstructure data, snow density ($\rho_{snow}$) can be derived as: $\rho_{snow} = V_{ice} / V_{total} \times \rho_{ice}$ [Lagagneux et al., 2002, Hagenmuller et al. 2016], where $V_{ice}$ the ice fraction volume, $V_{total}$ the total sample volume and $\rho_{ice}$ = 917 kg/$m^3$ the ice density [Kerbrat et al. 2008; Hagenmuller et al. 2016]. We also compute the specific surface area (SSA) as: $SSA = SA_{ice} / (V_{ice} \times \rho_{ice})$, where $SA_{ice}$ is the ice surface area. Density-cutter and micro-CT-derived bulk $\rho_{snow}$ are in good agreement, with differences of up to 15% (see **Figure S1**).

## 3 ROS Event: Meteorological and Surface Conditions

**Figure 2** summarizes the meteorological and bulk snow properties before, during and after the ROS events. On 13 September, $T_{air}$ rose above 0°C (10:00 UTC) and mostly remained above freezing until 09:40 UTC on the 14th (**Figure 2(a)**). Precipitation began around 05:00 UTC on the 13th and lasted 8 hours (**Figure 2(b)**), with the majority of liquid precipitation falling during this period. Another precipitation event started around 01:00 UTC on the 14th and lasted ∼3 hours. Sounding observations (not shown) show that the large fall velocities observed by the KAZR (**Figure 2(c)**) correspond to the passing of a warm air mass. Observations during the largest rain rates indicate vertical temperatures above zero to slightly above one kilometer. Combining information from observations of temperature, fall velocities, precipitation rate, and particle size distributions, it is straightforward to conclude that significant liquid deposition at the surface occurred throughout September 13th with minor liquid precipitation continuing early on the 14th. As surface temperatures cooled again midday on the 14th, and the soundings further indicated cooler vertical temperatures, any significant liquid deposition concluded by 12 UTC.

ERA5 is in remarkable agreement with the atmospheric data collected during MOSAiC. In particular, ERA5 is able to show a change in precipitation to liquid form; precipitation began as frozen snow/ice pellets on the morning of the 13th, followed by sleet and rain between noon and 20:00 UTC, until turning to wet snow and a mixture of rain and snow on the 14th (see **Figure S2**). The weather system originated East of Finland and crossed Novaya Zemlya and Franz Joseph Land. Another weather system passed near the MOSAiC floe on the 14th (see animation in the Appendix).

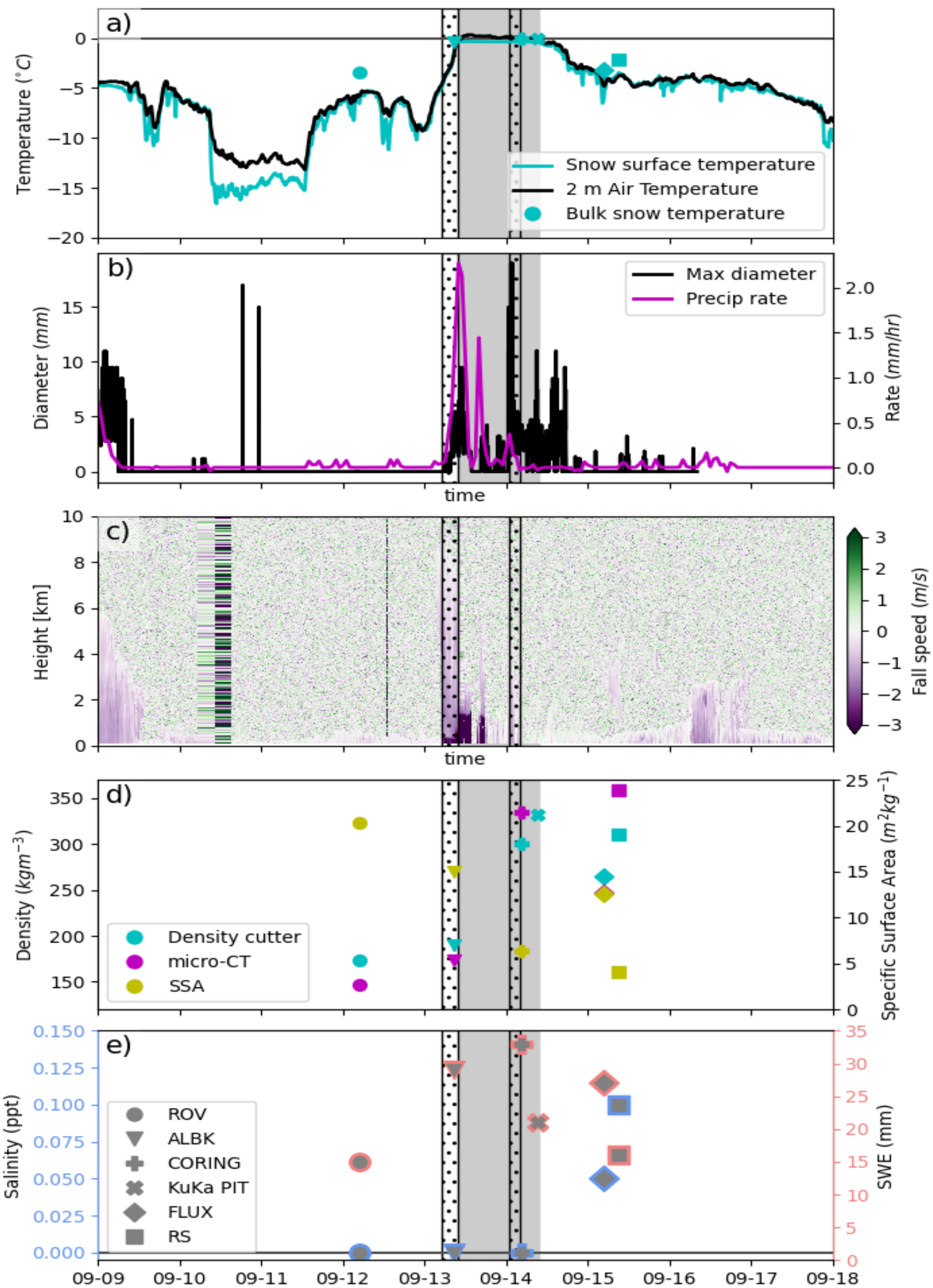

**Figure 2.** (a) 2-m air, skin, and bulk snow temperature; (b) precipitation rate (Pluviometer) and maximum particle diameter (Parsivel); (c) KAZR mean doppler velocity; (d) SSA and bulk snow density; (e) SWE and salinity. Symbols denote snowpit locations. Shaded area indicates air temperatures above 0°C surrounding the two ROS events (dotted areas).

Surface-based observations show the snow was dry prior to rainfall. Snow had been steadily accumulating over bare ice since September 3, without any observed melt. Accumulation 10 days preceding the ROS was a combination of snowfall, wind drift, and surface hoar deposition; the two days prior were characterized by widespread deposition of large surface hoar crystals (11 September) and riming (12 September, **Figure 3, 4(a)**, ROV). Evolution of surface snow conditions between 12 and 15 September is shown in **Figures 3 and 4**.

Prior to rainfall, the snow depth measured at the ROV snow pit on the 12th was 7 cm, and the snow had a bulk density of 173 kg/m$^3$ based on the density cutter (see **Figure 2(d)**). The Micro-CT derived density and SSA profiles show relatively constant profiles with depth except below the snow/ice interface, where the surface scattering layer is present [**Figure 4**]. On the 13th, another 7 cm deep snow pit was dug at the albedo site (08:30 UTC) shortly after it had started to rain. The Micro-CT profile data show only the top part of the snowpack was wet during the time of sampling (large drop in SSA near the surface on the 13th - **Figure 4**), indicating the rain had not yet percolated through the entire snowpack. The bulk density slightly increased (190 kg/m$^3$) compared to the previous day (**Figure 2(d)**). Bulk snow temperature on the 13th was ∼0°C (**Figure 2(a)**) and the SWE increased sharply from 15 mm (12 September) to 29 mm (13 September) (**Figure 2(e)**). However, since SWE measurements include the surface scattering layer (SSL), the SWE increase may also reflect SSL thickness changes due to temperature increase or the increase in liquid water content in the snowpack.

On the 14th, after the rain has percolated into the snowpack, and during the second rainfall event, bulk density increased to 300 and 332 kg/m$^3$ at the coring site and along the KuKa transect, respectively (**Figure 2(d)**). The entire snowpack had low SSA and higher density (**Figure 4**). Photographs show the rain caused local ponding and wet slushy snow (**Figure 3**, KuKa), characterized by rounding of the large grains of rimed surface hoar and compaction of the snowpack (see Micro-CT image in **Figure 4(c)**). The microstructure shows that part of the measured profile was the SSL, which had increased in thickness due to warmer temperatures.

On the 15th, after the rainfall ended and temperatures dropped below freezing, snow pits sampled from the flux and RS sites show a drop in bulk temperature to -3.1°C and -2.2°C (**Figure 2(a)**), respectively, yet significant differences in SWE, density, and SSA between the two snow pits (**Figures 2(d)** and **(e)**, **Figure 4(d,e)**). The earlier pit (flux site) shows surface dusting of snow (**Figures 3 and 4**), resulting in higher SSA at the surface, higher SWE, and lower density (264 kg/m$^3$). The snow pit collected 5 hours later (RS site) shows a mostly refrozen snowpack with new snow only accumulating in small depressions, and reduced SWE and SSA, yet slightly higher bulk density (280 kg/m$^3$). This highlights potential spatial variability in snow conditions, yet it is difficult to separate spatial variability from temporal changes since the snow pits were not sampled at the same time. On the other hand, almost all profiles exhibit a higher-density layer ≈1 cm above the snow/ice interface (**Figure 4**), indicative of a possible refrozen layer. This layer also less permeable, leading to internal ponding of percolated water, as seen in **Figure 4(c,d)** Salinity in the snow pits prior to the 15th was zero. On the 15th, snow salinity measurements were just above 0 ppt, but still less than 0.1 ppt, which is within the accuracy of the conductivity sensor.

Given that surface conditions were generally similar across the floe, these observations broadly represent snowpack conditions viewed by KuKa and the SBR. Next we describe the evolution of radar backscatter and microwave emission corresponding to these changes in snowpack properties.

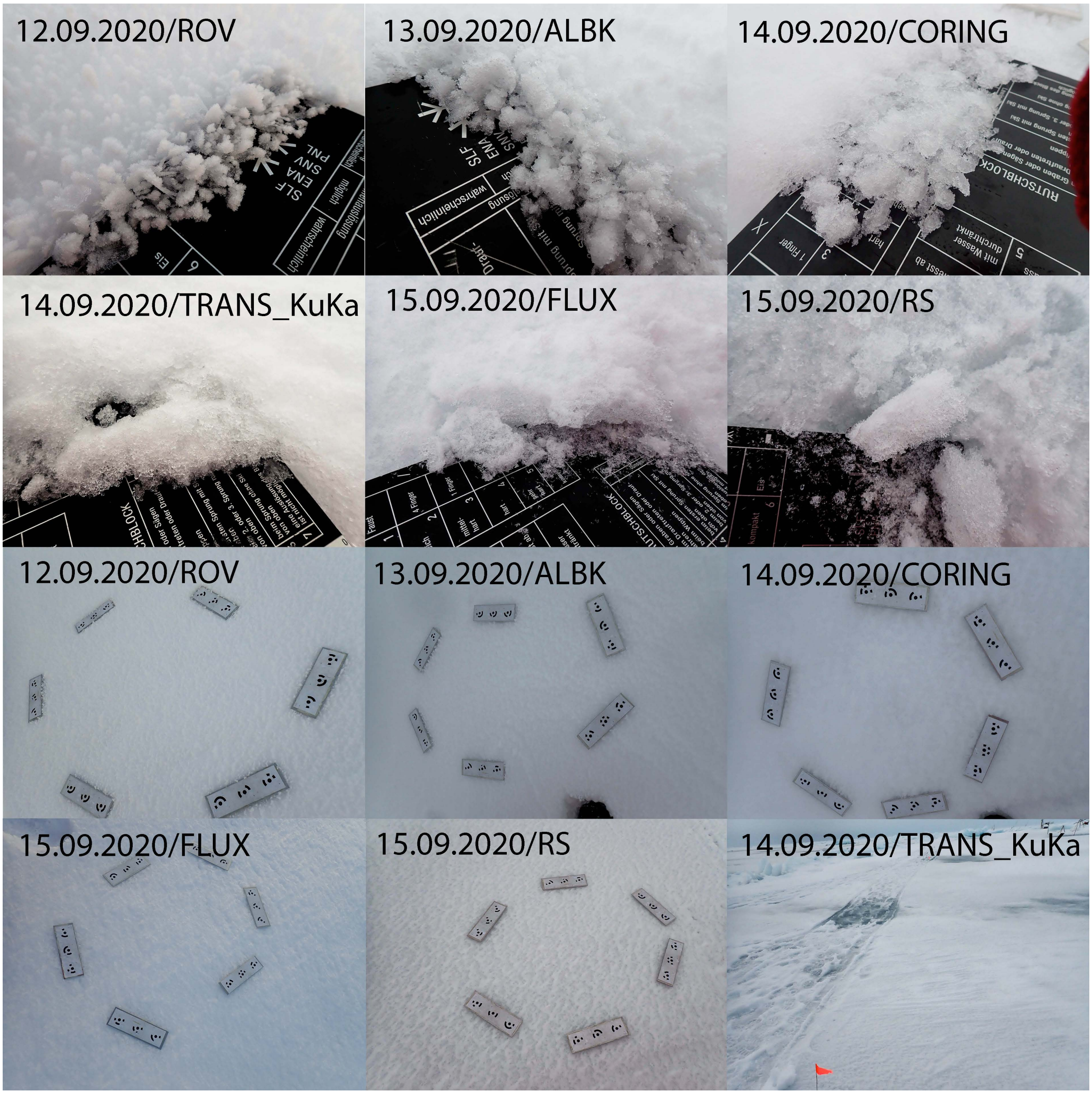

**Figure 3.** Photographs of snow surface grain size and surface conditions at various locations on the MOSAiC floe during the rain-on-snow event. Locations and dates are provided on each photograph.

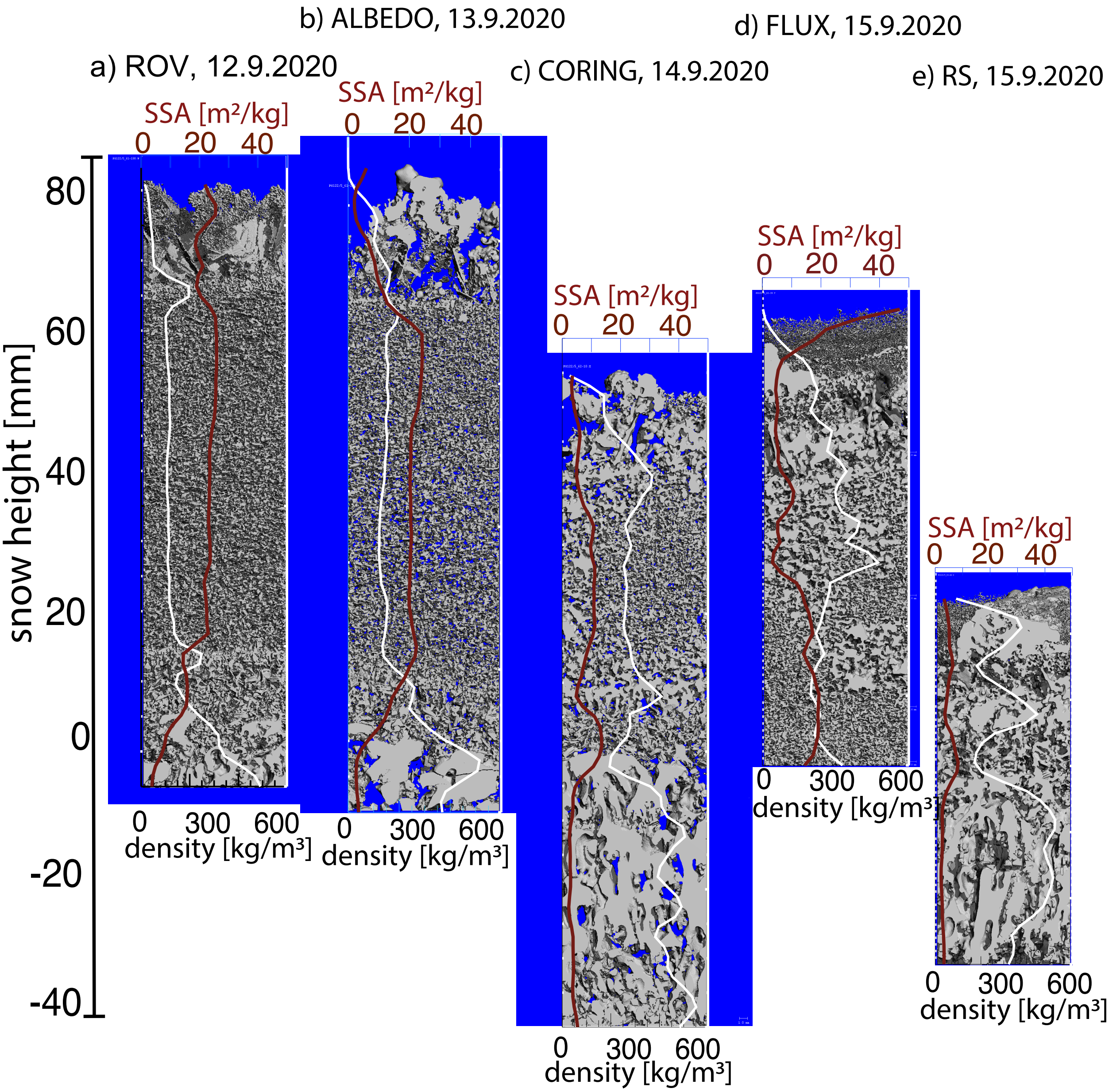

**Figure 4.** Micro-CT images of the snowpack before, during and after the rain-on-snow, together with Micro-CT derived $\rho_{snow}$ (white line) and SSA (dark red line) profiles from which depth-averaged values are calculated (e.g. bulk density and SSA). The bottom of the snowpack (snow/ice interface) is represented by 0. Above 0 is snow, and below is surface scattering layer (SSL).

## 4 Impacts of ROS on Radar Backscatter and Microwave Emission

### 4.1 KuKa Radar

Radar backscatter remains relatively stable at both Ka- and Ku-bands prior to rainfall (**Figure 5(a)-(b)**). At nadir, VV ≈ HH, indicating that the air/snow interface is acting as an electromagnetially rough surface [Onstott, 1992]. At $\theta$ = 45°, VV

backscatter is slightly greater than HH, and backscatter is dominated by volume scattering from snow grains and porous ice layers in the snow volume. Further, the overall magnitude of the co- and cross-polarized backscatter is higher at nadir compared to $\theta$ = 45°, especially for Ku-band HH and VV (i.e. by ≈ 8 dB), also indicating strong surface scattering from the air/snow interface.

During the first ROS event, radar backscatter declines at both frequencies and all polarizations as a result of increasing signature attenuation by liquid water, due to greater dielectric loss [Ulaby and Stiles, 1980]. The decline is larger at Ka-band due to stronger sensitivity of higher frequencies to snow surface changes: Ka-band backscatter reduces ≈ 12 dB (at both VV and HH) and ≈ 18 dB (HV) at all incidence angles. At Ku-band, VV and HH remain relatively stable at nadir but decrease by ≈ 7 dB (at both VV and HH) and by 10 dB (HV) at $\theta$ = 45°. Soon after the onset of the second ROS event, the snowpack transitions to a funicular regime (i.e. liquid water occupies continuous pathways through the snow pore spaces - see blue areas between snow grains in the Micro-CT images in **Figure 4**). This results in downward percolation of liquid water via gravity drainage [Colbeck, 1982a; Denoth, 1999] and likely leads to the large backscatter decline during the second rain event. Ka-band backscatter declines by ≈ 12 dB, while Ku-band declines by more than 15 dB, at all polarizations - the decline is larger at nadir. Overall, Ka- and Ku-band backscatter fell 4 and 6 standard deviations (16.5 and 19.3 dB), respectively, outside the observed backscatter variability before the ROS event.

After the snowpack refreezes, nadir backscatter increases approximately 20 and 25 dB at Ka- and Ku-bands respectively, and remains higher than before the ROS. We speculate that refreezing results in increased porosity in the refrozen snow pore spaces [Colbeck, 1982b; Denoth, 1982] creating air-filled vertical ice channels of various sizes on the order of Ku-band wavelength within the snow. From the MicroCT images, we see evidence of a percolation channel, though more analysis is needed to determine how the porosity changed. If the porosity indeed increased, this could increase Ku-band volume scattering. While the difference in VV and HH backscatter between nadir and 45° increases slightly (≈ 1.5 dB), the angular dependence for Ku-band HV almost vanishes. This indicates dominant volume scattering from the refrozen surface crusts [Strozzi et al., 1997]. An increase in Ka-band HH and VV backscatter relative to that observed prior to rainfall is also observed, which can be explained by an increased dielectric constant difference at the air/snow interface from the glazed surface crust. By end of 17 September, while Ku-band backscatter angular dependence returns to similar values before it rained, Ka-band shows larger angular dependence at HH and VV, while HV decreases.

Next, we consider the impact of the ROS event on radar waveforms (**Figure 6**), i.e. the shape of the power-time plot at various times before, during and after the ROS events. **6(a)** and **(b)** show the HH polarized power as a function of time at both Ka- (a) and Ku-band (b) as a function of range from the antennas (m). Hatch marks indicate locations of the selected waveforms shown in the inset to the left of the echograms. As discussed in Stroeve et al. [2020], the range from which most power is returned corresponds to the air/snow interface in both Ka- and Ku-bands. A feature, possibly some snow pushed into a small dune, appears between 1.4 - 1.6 m range on the morning of 9 September (left side of echograms). As we saw with the backscatter changes, the waveforms are stable before the first ROS event (09-09 up to the black samples, 09-13 08:49 UTC in right inset panel). The black sample shows power returned from the air/snow interface and also from below.

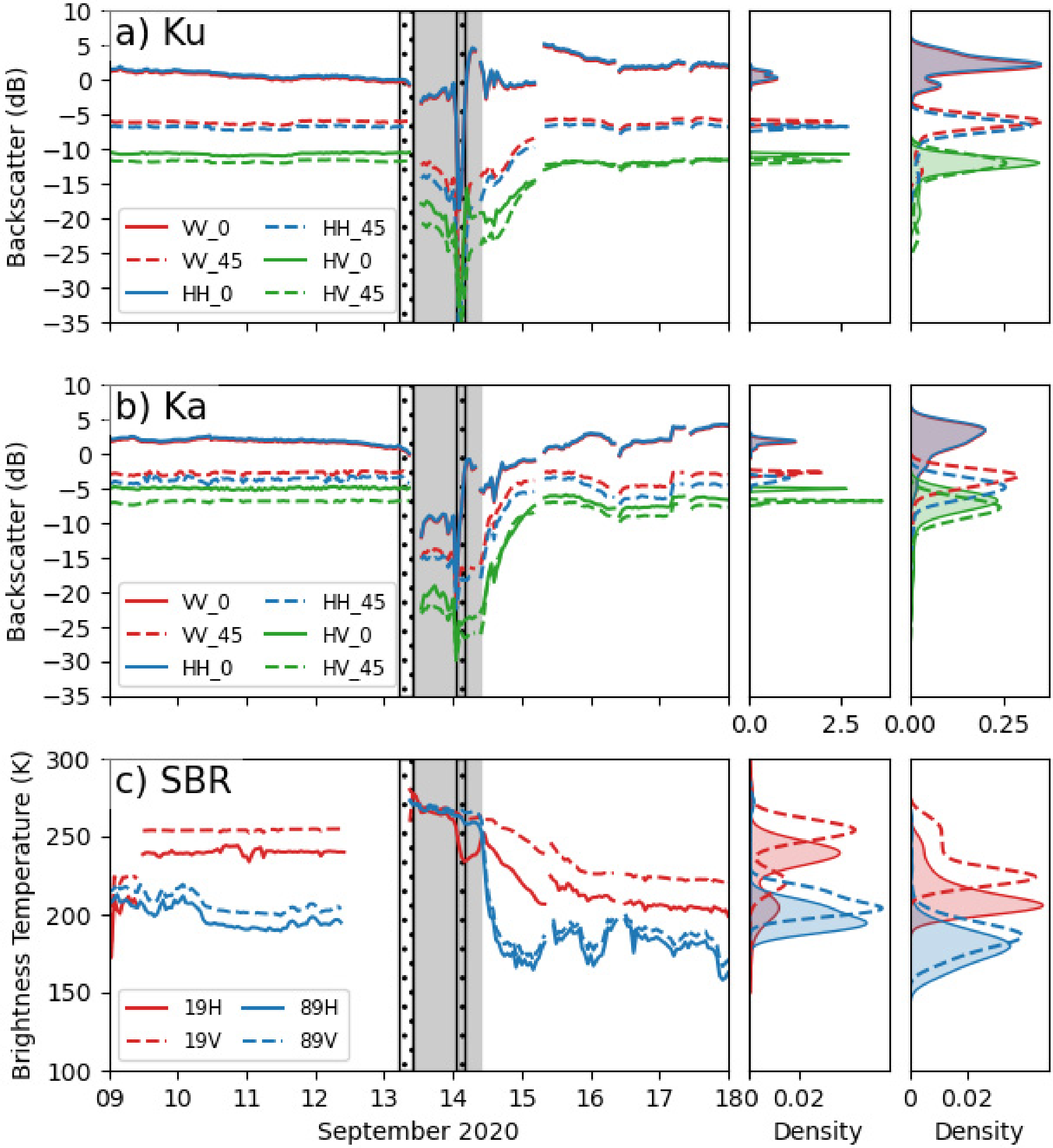

**Figure 5.** (a) Ku- and (b) Ka-band radar backscatter measured at nadir and $\theta = 45^{o}$ ; (c) 19 and 89 GHz brightness temperatures at horizontal and vertical polarizations at $\theta$=55°. Corresponding kernel density distributions are shown in the right panels before and after the ROS event. Shaded/dotted areas are the same as in Figure 2.

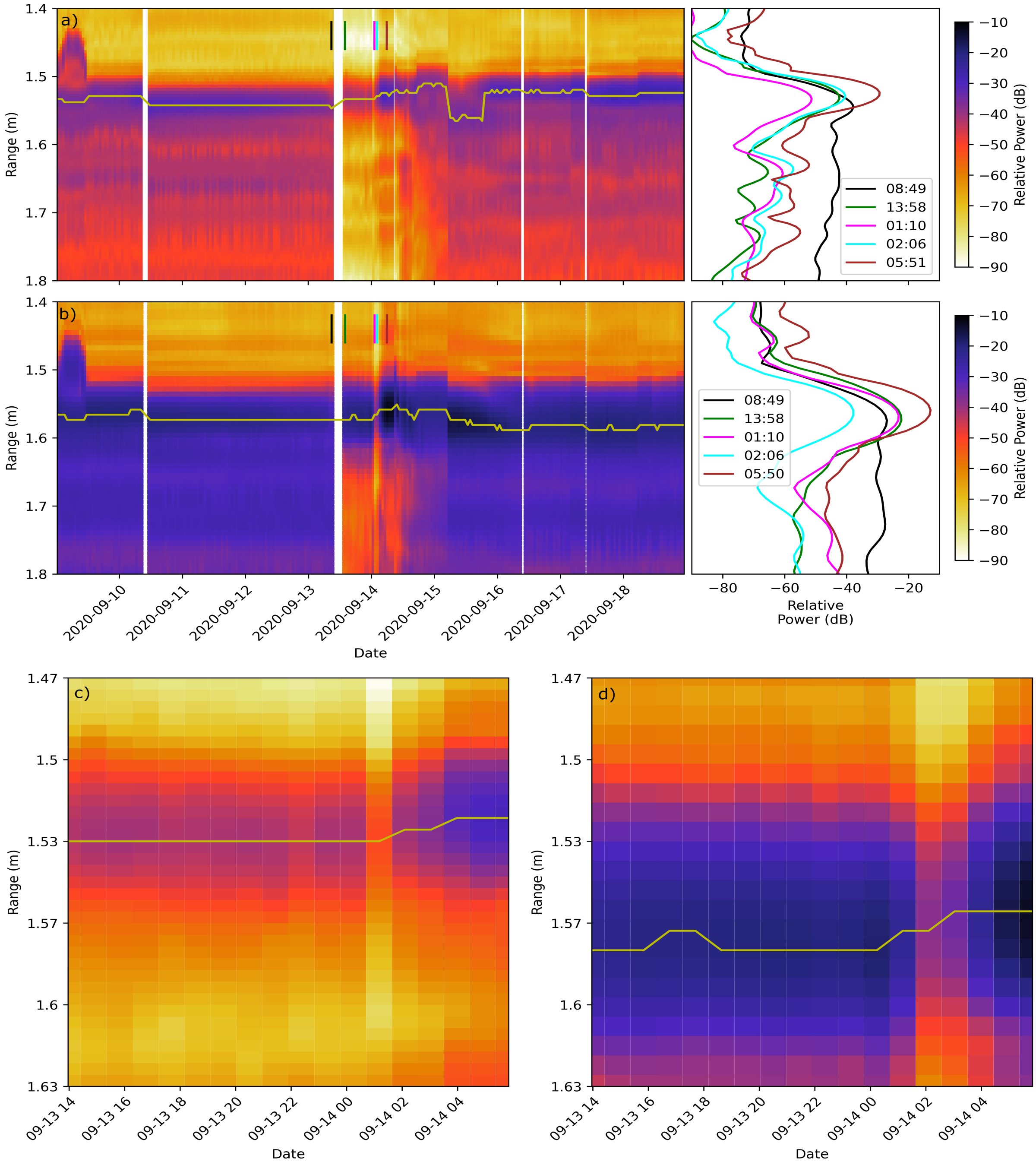

**Figure 6.** (a) Ka- and (b) Ku-band HH echograms at nadir, indicating scattering from different ranges; hence waveform shape over time. Yellow line indicates the range bin with the highest power. White vertical stripes indicate data gaps. Inset panels on the right indicate waveforms selected at the dates/times shown by the colored indicators in the echograms, demonstrating changes in waveform shape, peak range, and leading edges during the 24-hr period. c) and d) are zoomed-in Ka- and Ku-band data, respectively, to show finer detail around the second ROS event.

During the first rainfall (green samples, 09-13 13:58 UTC), most of the power is returned from the snow/ice interface and the power returned from ranges greater than ∼1.6 m drops in both bands, causing a shadow-like region of lower power in the echograms. Note also the shift in the peak return in the green samples relative to the black ones in the inset towards shorter range at Ka-band. During the second ROS event (magenta and cyan samples on 09-14 01:10 and 02:06 UTC), the peak power associated with the air/snow interface decreases and there is a further shift to shorter ranges, now also evident at Ku-band. Following this, the refrozen snow surface increases scattering from the air/snow interface (brown samples 09-14 05:50 UTC). The shifts of the peak could be caused by scattering from higher in the snow pack, or from the instrument sinking into the snow, or some combination; we cannot determine which.

The change in waveform shape from the black to the cyan sample demonstrates that after the two rain events, a greater proportion of the returned power is associated with a peak at the air/snow interface, and a smaller proportion with ranges below the peak. During the ROS event the waveform peak power was reduced and these data also suggest that after a ROS event, refrozen ice layers can shift the apparent surface elevation. Note, the surface elevation drop on September 15th relates to the instrument being moved to a slightly different location.

To see how these results scale up to the satellite level, we investigated the change in CryoSat-2 waveforms during this time period, using AWI (Alfred Wegener Institute) CryoSat-2 L1P data. We plot the peakiness, which is an indicator of waveform shape related to the ratio of maximum to mean power, and sigma0 (**Figure 7**). As shown, during the two ROS events between September 13-14, there are changes to the waveform peakiness near the MOSAiC floe much greater than elsewhere in the Arctic. Since the CryoSat-2 peakiness increases, this suggests that more of the power is returned from a smaller number of range bins as the peak power increases relative to the mean along the waveform. CryoSat-2 sigma0 also changes, increasing on the 13th before reducing over the next few days, but it does not return to the level (16-17 dB) prior to the ROS events.

These results indicate that the ROS altered the CryoSat-2 waveforms, consistent with what we see in the KuKa data. Time-series of peakiness and sigma0 of CryoSat-2 and KuKa Ku-band (to match CryoSat-2) waveforms are overlaid in **Figure 8**. KuKa Ku-band peakiness is computed by taking the subset of range bins above the noise floor (-70 dB) and dividing the maximum power by the mean power in those bins, to calculate a peakiness value for each 24-hour period using a similar methodology to CryoSat-2. The CryoSat-2 data are averaged over a 130 km circle radius centered at KuKa's location each day; 130 km was selected to only included CryoSat-2 data collected as close as possible to KuKa whilst ensuring that at least two CryoSat-2 tracks were within the radius for each date. Due to the locations of the tracks, points were included from CryoSat-2 tracks in the afternoon and evening each day (ranging from 16:41 - 23:17 UTC), whereas KuKa data were gathered at hourly intervals.

The top panel shows waveform peakiness changed in a similar way for KuKa Ku-band and CryoSat-2: increasing sharply in response to rainfall and subsequent refreezing, then reducing. Note this can also be seen in the KuKa Ka-band data in **Figure 6(a)** where the brown echo (following refreezing) is peakier than the black echo (prior to ROS events). The bottom panel indicates that sigma0 also changes in both KuKa Ku-band and CryoSat-2 waveforms, but not in the same way; KuKa sigma0 reduces around the ROS events then increases following subsequent refreezing, whilst CryoSat-2 sigma0 increases following the ROS events, then reduces over time. Unlike peakiness, which is likely to vary in a similar way between the

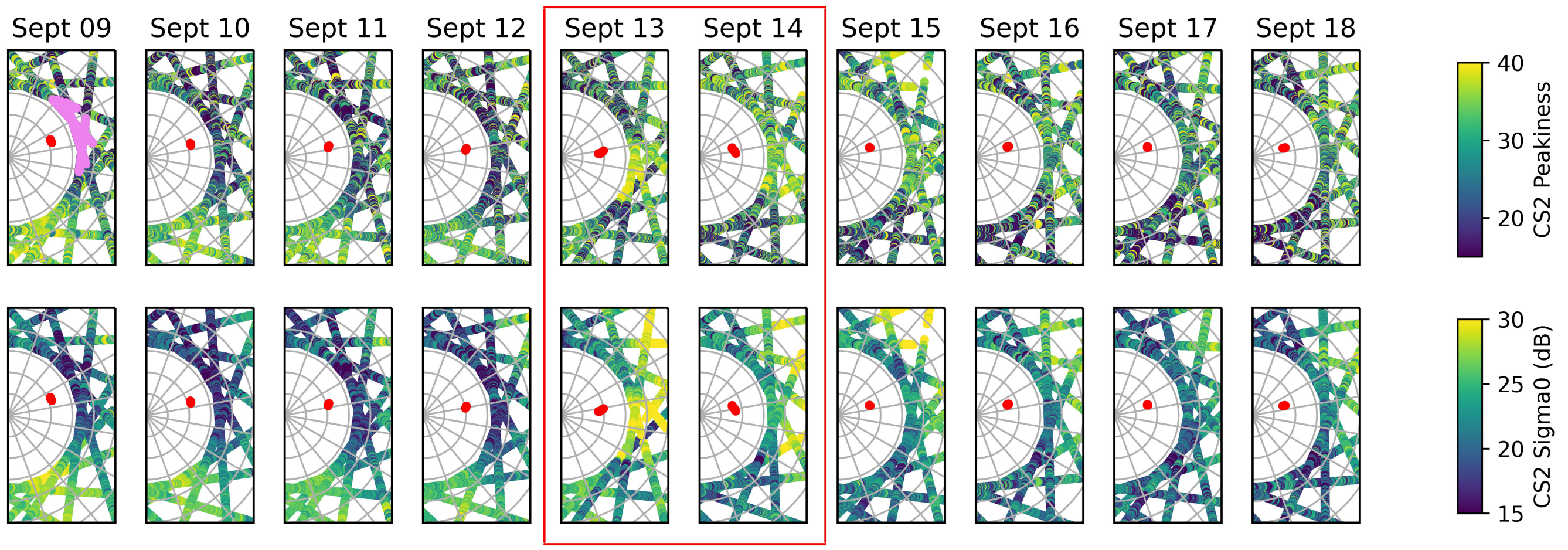

**Figure 7.** CryoSat-2 peakiness (top) and sigma0 (bottom) before, during and after the ROS events on September 13th to 14th (red box). Only waveforms with peakiness less than 40 are shown, as this is the cutoff for classification as a sea ice floe within the ice pack (waveforms with peakiness greater than 40 are classified as originating from leads.) Red circles indicate the locations at which KuKa collected data on that date. The lilac points on the top left panel show CryoSat-2 data points within 130 km of KuKa, from which the averaged quantities shown in Figure 8 are determined.

instruments, sigma0 is likely to be highly spatially variable and this may be one reason that the sigma0 calculated from the two instruments does not vary in the same way. Also, as noted above, due to the CryoSat-2 orbit, the data within 130 km of KuKa was gathered during the afternoon and evening each day, and the ROS events took some time to pass over the CryoSat-2 tracks and KuKa locations, hence the effects would differ in terms of the timing of changes to sigma0. Lastly, CryoSat-2 has a SAR/pulse-limited footprint of 300 x 1500 m compared to the 45 cm beam-limited footprint of the KuKa Ku-band instrument; CryoSat-2 footprints will include ridges, smooth and rough snow and ice surfaces whilst KuKa was surveying one flat area, hence the sigma0 backscatter may vary differently depending on what is included in the footprint and how it is affected by the precipitation.

### 4.2 Surface-Based-Radiometer (SBR)

The bottom panel of **Figure 5** shows the corresponding Tbs at 19 and 89 GHz during the ROS event. Prior to rain, Tbs are relatively stable except for an unnatural jump in the 19 GHz Tb on the 9th that may have been a result of a change in receiver orientation after instrument calibration and/or snow redistribution. Brightness temperatures at 89 GHz are less than at 19 GHz because volume scattering in the snow is larger at 89 GHz. The overall mean and standard deviation of the Tbs prior to rainfall (excluding the erroneous jump) are: $Tb_{19h}$ = 240 ± 2.0 K, $Tb_{19v}$ = 254 ± 0.9 K; $Tb_{89h}$ = 197 ± 6.6 K, $Tb_{89v}$ = 206.8 ± 6.0 K. On the morning of the 13th, the Tbs increase sharply to 274 K at both frequencies in response to liquid water in the snowpack. The increase to 274 K reflects the fact that emissivity of a wet snowpack is nearly 1 because of high absorption, and the physical

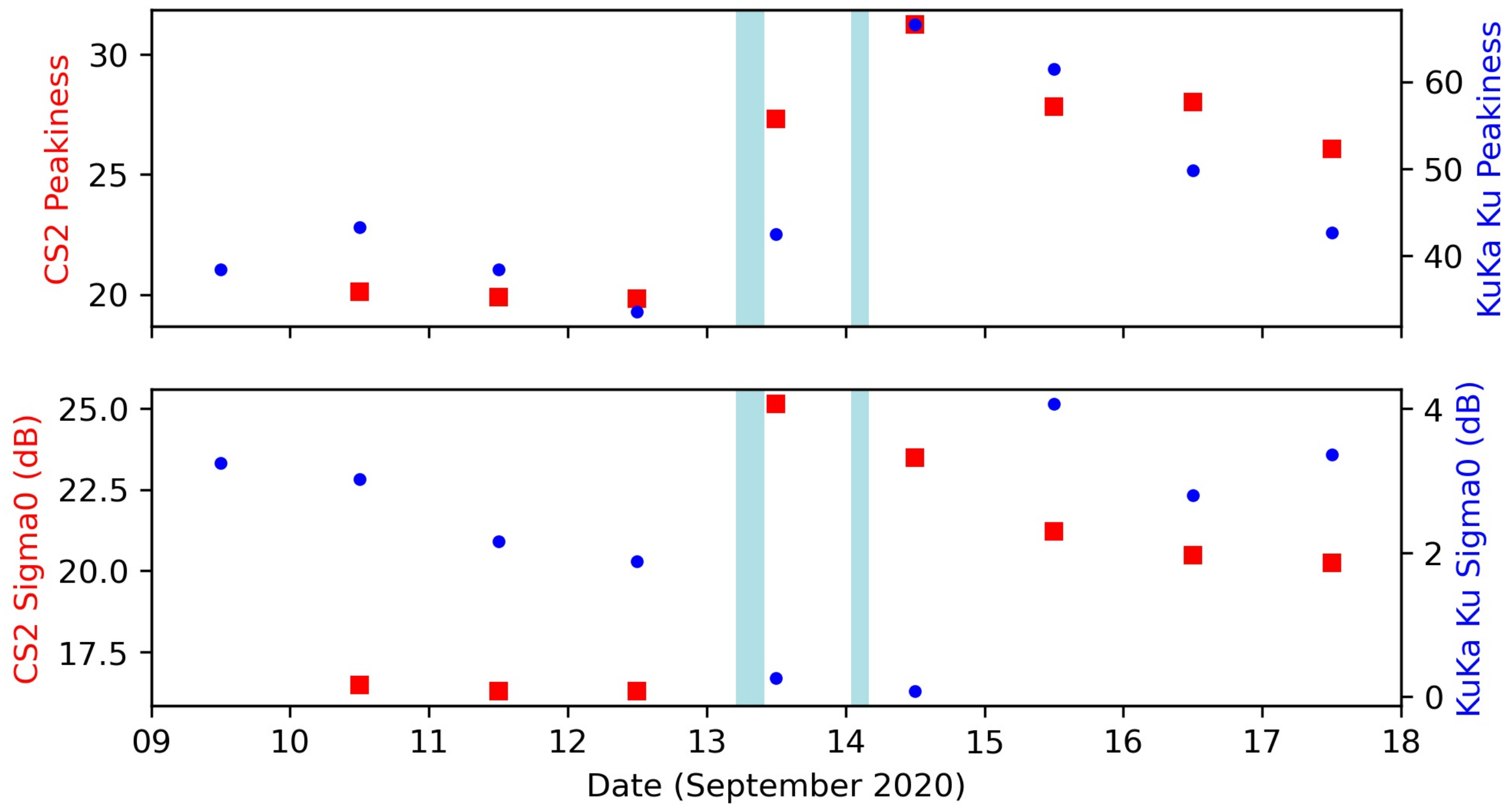

**Figure 8.** Timeseries of peakiness (top) and sigma0 (bottom) for CryoSat-2 (red squares) and KuKa Ku-band data (blue circles) averaged over 24-hour periods. The blue sections indicate periods of rain.

snowpack temperature at the top is slightly above 0°C. This also suggests that at the time the data were collected, the entire snowpack is wet since the 19 and 89 GHz channels are equally impacted.

Midday on the 14th, Tbs drop to cold conditions again, immediately impacting 89 GHz as the top of the snowpack refreezes. The 19 GHz channels, with deeper penetration depth, take longer to respond as the rest of the snowpack takes longer to refreeze or dry out completely from drainage. After the event, the snowpack remains cold, yet it is altered. In particular, the 19 GHz polarization difference (PD) (i.e. PD = $Tb_{19V}$ - $Tb_{19H}$) is larger than before the ROS event potentially as a result of ice layers in the snowpack; the 89 GHz PD on the other hand decreases. Further, grain size increased throughout the snowpack, and thus Tbs are lower than before it rained (more volume scattering), affecting both 19 and 89 GHz.

We further investigated if this particular ROS event is captured in satellite passive microwave observations. The Advanced Microwave Scanning Radiometer 2 (AMSR2) on board GCOM-W1 is able to detect this rain event (**Figure 7**) despite the coarser spatial resolution (12.5 km grid resolution) and temporal averaging (i.e. daily averaged ascending orbits). AMSR2 Tbs sharply increase at all frequencies during the ROS, strongest at 89 GHz (increase of ~27 K ($Tb_{89h}$) and 8 K ($Tb_{89v}$). This is consistent with the larger increase at higher frequencies observed in the SBR. Similarly, the decline afterward is stronger at higher frequencies because they are more sensitive to changes in the top layers of the snowpack and it takes time for the entire

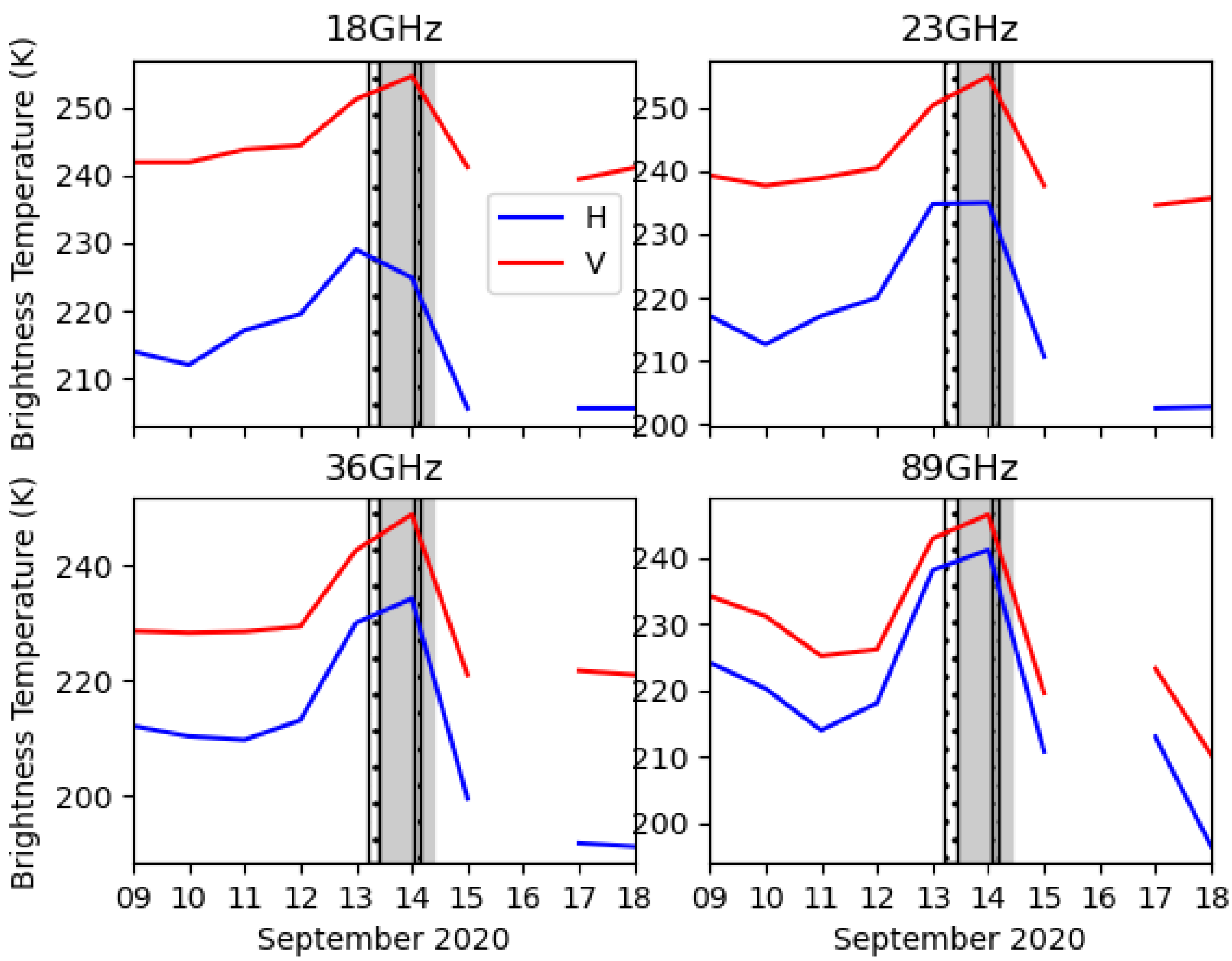

**Figure 9.** AMSR2 brightness temperatures from the 8th of September until the 18th. Data are shown for the descending passes at 12.5 km spatial grid resolution, as extracted for the MOSAiC floe.

snowpack to refreeze. The polarization difference increases for the lower frequency channels after the snowpack refreezes, with the largest polarization change at 18.7 GHz, in line with the SBR observations at 19 GHz. The agreement between AMSR2 and SBR is remarkable given the large spatial and temporal resolution differences in the two measurements and further illustrates that the ROS covered quite a large region.

## 5 Implications

### 5.1 Radar Altimetry

Radar freeboard height retrievals are based on assumptions as to where the dominant scattering horizon is located: i.e. the snow/ice interface at Ku-band, and air/snow interface at Ka-band [e.g. Tonboe et al., 2021]. Our waveform analysis suggests

the peak associated with the dominant scattering surface moved upwards (to smaller range) after the second ROS event and refreezing. The shift is 0.9 cm (Ka-band) and 1.5 cm (Ku-band). This could be due to rainwater refreezing at the surface, raising its elevation, and/or decreasing roughness at nadir. Another contribution could be settling of the radar; it is not possible to verify whether the sled sank into the melting snow and therefore we cannot quantify whether this partially or fully accounted for the apparent change in range.

While satellite radar altimeters cannot resolve the peaks visible in the KuKa data, and there are important differences in range resolution and the echo shapes between KuKa and satellite radar altimeters, the *in situ* shape changes at both Ku- and Ka-band demonstrate that satellite-retrieved freeboards from both CryoSat-2 and AltiKa could be shifted upwards. However, satellite radar return power also depends strongly on the large-scale floe topographic variability (i.e. ridges, rubble fields), which controls the total number of illuminated point scatterers as a function of delay time. While there are challenges in upscaling to satellite footprints, KuKa data combined with measurements of physical snow characteristics allow us to investigate how the combination of snow depth, temperature, salinity and radar-scale roughness control sigma0, and how these facet-scale factors affect the shape return pulse, as a function of transmitted pulse bandwidth. These insights can then be combined with numerical radar simulation approaches that focus on large-scale floe topographic variability, with realistic gain patterns, to understand how the different factors combine to influence the radar return power over km scale satellite footprints.

The change in waveform shape may further impact classification of sea ice vs leads. Lead waveforms can be distinguished from ice floes using waveform 'peakiness' [Laxon, 1994]; when melt ponds form, peakiness increases as the echoes become more specular [Tilling et al., 2018]. The change in waveform shape caused by the ROS events affects the peakiness of the KuKa Ku-band echoes, and similar effects are seen on the CryoSat-2 echoes in that area. This indicates that ROS could affect lead/sea ice discrimination by altering the classification of a floe to a lead, artificially increasing the apparent local sea surface height and incorrectly removing a waveform from the floe dataset. Since conventional thickness retrieval methods from radar altimetry are limited to the cold season months, we cannot fully evaluate the impacts of our detected waveform shape changes on altimetry-derived sea ice thickness. In a follow-on study we will investigate the effect of our observed waveform shape changes on retrieved sea ice freeboard and thickness during the cold season.

Another consideration regarding how ROS can influence ice and snow thickness retrievals is the hydrostatic balance of the ice floe. In particular, the introduction of additional SWE in the form of rainwater into the snowpack will reduce the freeboard by weighing down the floe, hence reducing the elevation of the air/snow and snow/ice interfaces above the waterline. ERA5 indicates an additional 11.5 mm of SWE was deposited onto the snowpack, which would reduce the air/snow interface elevation by 13.6 mm. Snow density changes as a result of rainfall further impact retrieved elevations by altering the speed of radiation propagation through the snow cover [e.g. Mallett et al. 2020; Willatt et al. 2010].

Finally, it is worth noting that on first-year ice (FYI), ROS impacts can be even more complicated. Snow on FYI is saline [Geldsetzer et al., 2009] and the brine content alters the dominant backscatter horizon [Nandan et al., 2017]. ROS can flush brine from the snow, thereby freshening the snowpack [Nandan et al., 2020]. We hypothesize that, upon refreezing, a brine-free and colder snowpack with zero brine volume allows greater penetration of radar signal. Thus, the error introduced by snow

salinity is reduced. In addition, the changes to snow structure and porosity may also affect the radar backscatter, and these effects may interact may interact in complex ways.

### 5.2 Scatterometry

**Figure 5** shows a strong reduction in KuKa radar backscatter when there is liquid water in the snowpack and a backscatter increase as the snowpack refreezes. However, the changes to CryoSat-2 backscatter Figures 7 and 8 were not the same as for KuKa, indicating that changes to backscatter could be highly localised spatially and/or temporally, and/or dependent on the instrument and processing chain characteristics. For satellite-based scatterometers and Synthetic Aperture Radar (SAR), ROS events during winter can thus lead to permanent geophysical and thermodynamic changes to the snowpack leading to complexly-layered snowpacks. These include changes in salinity- and temperature-dependent brine volume (for FYI) [Nandan et al., 2020], density [Denoth, 1999] and snow grain microstructure [Colbeck, 1982].

For example, on the extreme end, ROS-induced melt can lead to melting of all of the snow, leaving bare sea ice, resulting in ice surface scattering. On FYI, ROS events could create a superficial ice layer right above the sea ice, that would constrain upward brine wicking through snow pack. Refreezing of snow after a ROS event can lead to formation of ice lenses and/or air-filled vertical ice channels, inclusions or poly-aggregates of various sizes during refreezing within the snowpack [Colbeck, 1982; Denoth, 1999]. While ice lenses facilitate additional surface scattering, ice channels produce more volume scattering at large incidence angles relevant to SAR and scatterometers.

All of these factors can complicate retrievals of melt-freeze transitions from scatterometers and SAR [e.g. Mortin et al., 2014]. In particular, a ROS event does not necessarily represent the start of the melt season. This is because the ROS event can be considered anomalous and premature prior to sustained microwave-detected melt-onset, by dampening the dielectric permittivity [e.g. Mortin et al., 2012]. In addition, once the snowpack becomes complexly-layered following ROS events, the reliability on microwave signal return for snow/sea ice geophysical retrievals and sea ice classification of FYI and MYI types become ambiguous [Nghiem et al., 1995].

### 5.3 Passive Microwave

Finally, SIC, melt onset/freeze-up timing, snow/ice interface temperature, thin ice thickness, and snow depth are all regularly retrieved using current satellite passive microwave systems operating at frequencies studied here [e.g. Comiso et al., 1997; Markus and Cavalieri, 2009; Markus et al., 2009; Groenfeldt, 2015; Tietsche et al., 2018; Kaleschke et al., 2016; Rostosky et al., 2018]. Since the penetration into the snow/ice depends strongly on frequency, with lower frequencies penetrating further, ice layers in the snowpack will have a larger impact on retrievals relying on low frequency channels. This includes thin ice thickness retrieved from satellites such as the Soil Moisture and Ocean Salinity (SMOS) L-band (1.4 GHz) mission, snow/ice interface temperature retrievals using 6.9 GHz from AMSR2, and snow depth retrieved using a combination of 19/37 GHz or 7/19 GHz channels.

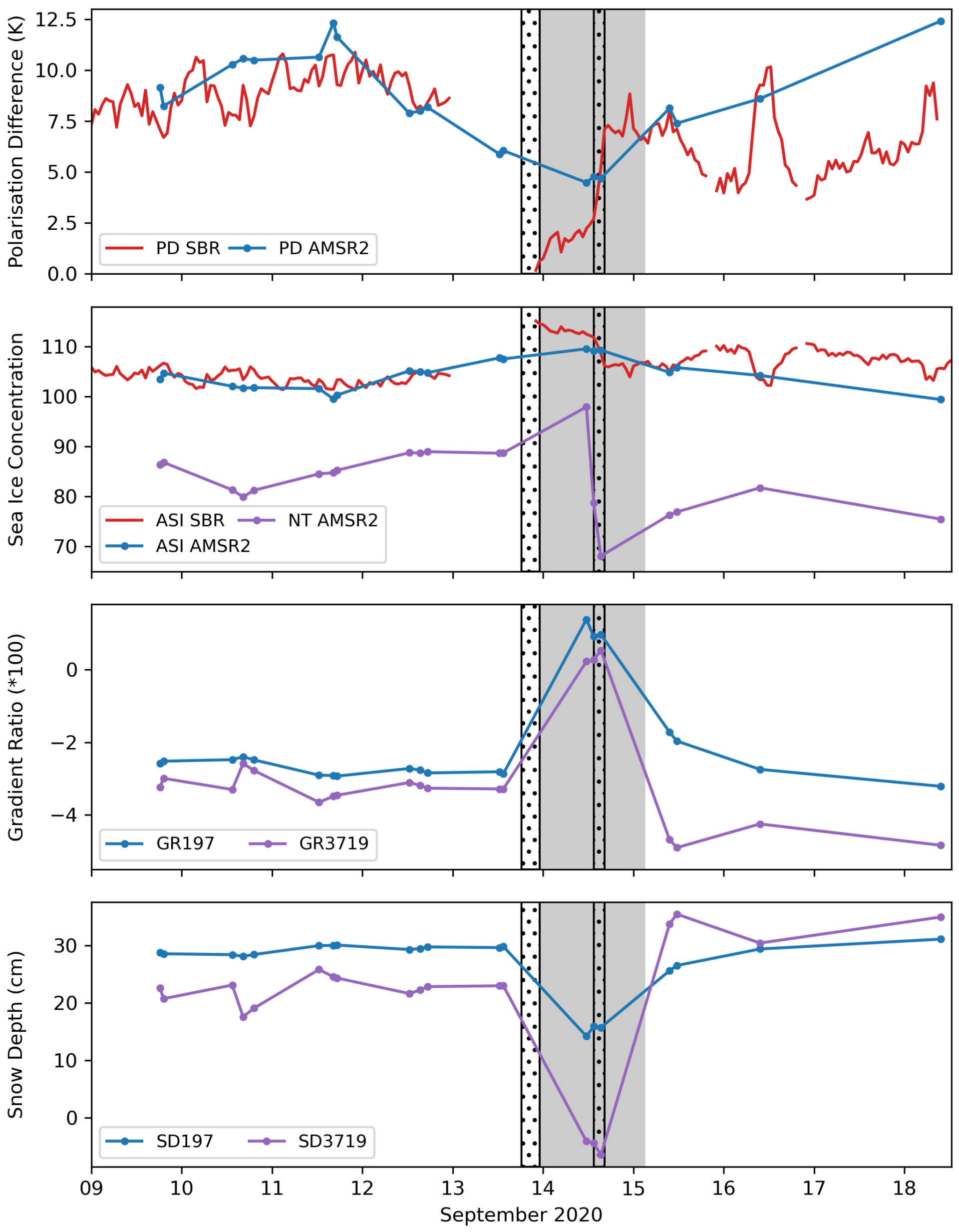

**Figure 10.** Collocated AMSR2 and SBR polarization difference (PD) at 89 GHz (top) and sea ice concentration derived from the polarization difference at 89 GHz using the ASI algorithm [Spreen et al., 2008] (2nd row). In addition, the sea ice concentration from teh NASA Team algorithm [Cavalieri et al., 1997] is shown . 3rd row: gradient ratio (at vertical polarization) derived from AMSR2 37 GHz and 19 GHz (blue) and 19 GHz and 7 GHz (red) observations. 4th row: snow depth derived from GR(19/7) and GR(37/19) during the ROS event.

Nevertheless, impacts occur also at higher frequencies. SIC retrieved with the ASI algorithm [Spreen et al., 2008] is based on 89 GHz polarization differences (PD). Wet snow has an emissivity close to 1 at both V and H polarization, and thus the PD decreases during ROS (**Figure 10(a)**). This leads to a SIC overestimation of 5% (AMSR2) to 10% (SBR) (**Figure 10(b)**). However, because the ASI SIC was already at 100% at the MOSAiC ice floe, and thus limited to the maximum value, this increase cannot be seen in the operational satellite record. But if the ROS event happened in a lower SIC region, the event would have caused an overestimation, e.g., 90% instead of 82%.

On the other hand, The NASA Team algorithm [Cavalieri et al., 1997] that relies on 19 GHz H/V polarization ratio (PR19) and 37, 19 GHz gradient ratio (GR(37/19)) yields a SIC decrease from 90 to 70% during ROS (**Figure 10(b)**). Both PR19 and GR37,19 increase during the ROS event (GR(37/19) shown in **Figure 10(c)**), which causes a SIC decrease for the NASA Team algorithm. The response lasts long after refreeze, as the polarization and gradient ratios are altered, leading to continued underestimation of SIC of about 10% compared to the values prior to the rain event.

Snow depth retrievals over FYI [Markus and Cavalieri, 1998; Comiso et al., 2003] and multiyear ice [Rostosky et al., 2018] are also impacted. While the event during September is too early in the season for a quantitative satellite snow depth retrieval, both the AMSR2 19/37 and 7/19 GHz gradient ratios (GR) were increased during the ROS event (**Figure 10(c)**) because of the reduced penetration depth in the snow and similar scattering at the two frequencies. If Markus and Cavalieri [1998] and Rostosky et al. [2018] snow depth retrieval methods are applied, this would lead to a retrieved snow depth of 0 cm (using GR19/37) or reduction by 50% (GR7/19), respectively (**Figure 10(d)**).

## 6 Changes in ROS

While MOSAiC offered the opportunity for a detailed *in situ* analysis of a ROS event, this event occurred at the end of the summer melt season, at a time before sea ice variables, such as ice thickness and snow depth are retrieved from satellites. However, if these events happen in winter, they would likely have similar impacts on the snow cover, with liquid water percolating through the snow and refreezing. The question then is whether or not ROS or winter warming events are becoming more common. Some studies have already looked at increased frequency and duration of winter warming events using atmospheric reanalysis data [e.g. Graham et al., 2017]. Here we examine ERA5 reanalysis to see if there is a similar increase in winter rainfall.

While there is considerable year-to-year variability, a statistically significant positive trend (at 95% confidence interval) towards more rainfall during the winter season (October through April) is seen for the Arctic Ocean and its marginal seas (**Figure 11(a)**). This is dominated by statistically significant trends in the Central Arctic and the Laptev Sea (**Figure 11(b)** and Appendix). Trends are less steep for the December to February period (not shown), reflecting a larger increase in wet precipitation during autumn freeze-up and melt onset periods (e.g. October November and March-April). While the amount of rainfall is still relatively small in magnitude, we may expect winter rainfall events to become increasingly important as the Arctic continues to warm under atmospheric greenhouse gas forcing and starts to transition to more rainfall earlier than previously forecasted [e.g. McCrystall et al., 2021].

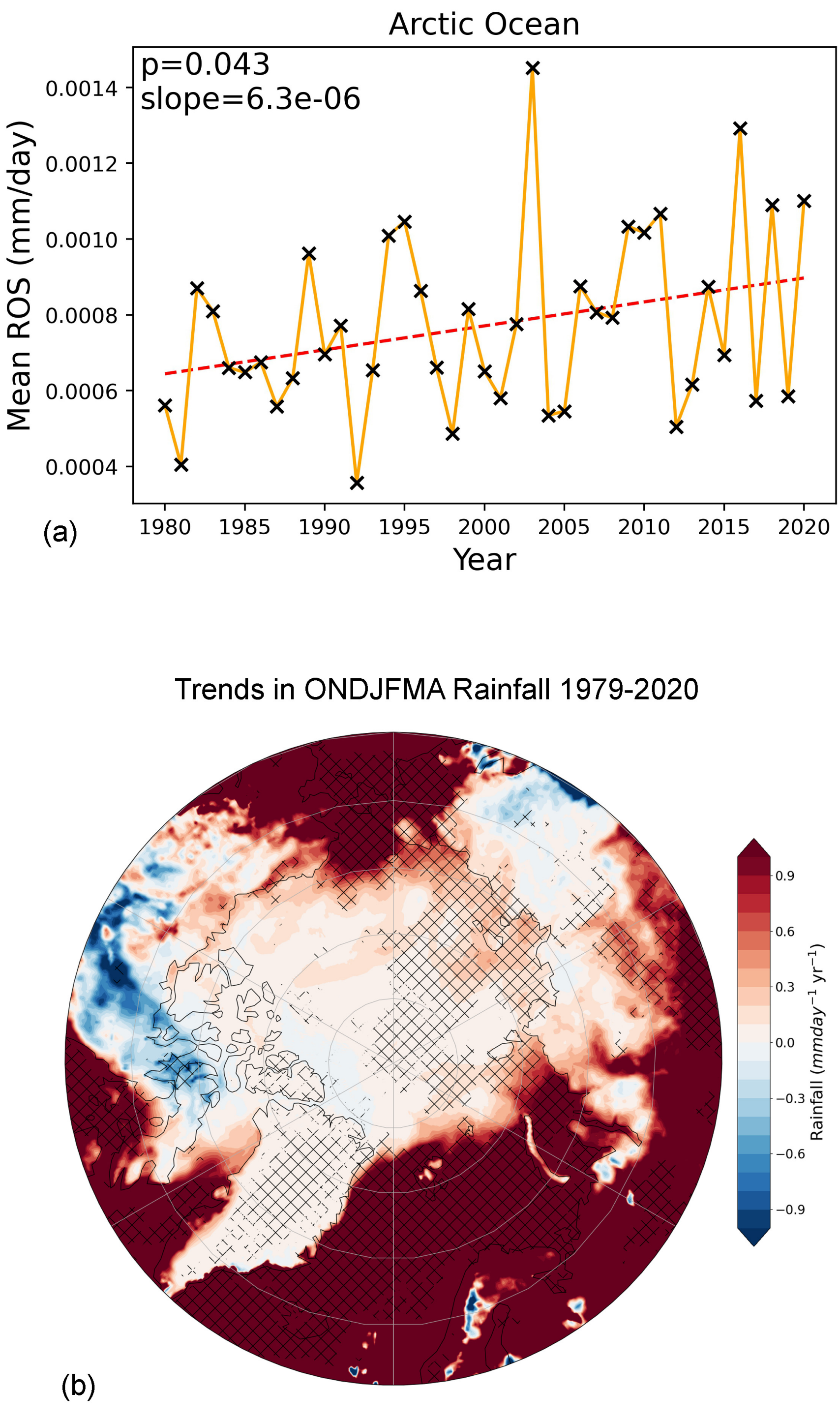

**Figure 11.** (a) Mean wet precipitation (rain, freezing rain, wet snow and mixed rain/snow; mm/day) over sea ice for cold season (Oct-Apr inclusive). Trend determined by a two-tailed test for non-zero correlation in values over time, with p value of 0.043. (b) Spatial trends in rainfall from 1979 to 2020 averaged from October through April.

## 7 Conclusions

During the MOSAiC expedition, several remote sensing instruments were deployed to better understand how snowpack and sea ice properties impact satellite retrievals of various sea ice variables. The instruments spanned frequencies relevant to both current and future (e.g. Copernicus CRISTAL and CIMR) satellite systems. A rain-on-snow event towards the end of the year-long MOSAiC expedition in September 2020 provided a unique opportunity to utilize these assets to study the impact of ROS on active and passive microwave radiation, providing information germane to developing ROS detection over sea ice and for understanding ROS impacts on derived snow depth, sea ice thickness and ice concentration.

Notably, while the geometries of ground-based systems are different from satellite altimeters, we show that ROS can alter waveform shape, with impacts on discrimination of sea ice from leads, and that shifting in the range of the peak can lead to an apparent increase in sea ice freeboard, and hence thickness retrievals from CryoSat-2/AltiKa. Permanent changes observed in microwave emissivity have the potential to impact retrieved ice concentrations and snow depths.

Since Fram Strait to the North Pole is often visited by temporary winter warming events [e.g. Graham et al. 2017], we may expect this region of the Arctic is already more vulnerable to potential biases in satellite-retrieved sea ice variables as a result of ice layers from warmer air temperatures and/or rain-on-snow events. Looking forward, as the Arctic continues to warm, it is reasonable to expect an increase in the frequency and possibly intensity of ROS and winter warming events. Coupled with shallower snowpacks atop the ice cover, due to later ice formation [Stroeve et al. 2020], and that snowpacks atop first-year ice have a non-negligible brine content that also affects microwave emission and backscatter [e.g. Nandan et al. 2017], thickness and snow depth retrievals will become more challenging in coming decades, raising the importance of obtaining ground truth such as collected during MOSAiC.

*Code and data availability.* Data used in this manuscript was produced as part of the international Multidisciplinary drifting Observatory for the Study of the Arctic Climate (MOSAiC) with the tag MOSAiC20192020 and the Project ID: AWI PS122 00. SBR data is available from the MOSAiC Central Data Storage (MCS). KuKa data are available at http://data.bas.ac.uk/full-record.php?id=GB/NERC/BAS/PDC/01437. KAZR and precipitation data are from the Atmospheric Radiation Measurement (ARM) user facility, a U.S. Department of Energy (DOE) Office of Science managed by the Biological and Environmental Research Program. Meteorological data was funded by National Science Foundation (#OPP-1724551). Code for producing the summary plots of KuKa, SBR, atmospheric and snow conditions can be found at: https://github.com/andypbarrett/mosaic rain on snow

*Video supplement.* Animation of storm from ERA5 Reanalysis included in Appendix

## Appendix A: Data Processing and Supplemental Figures

### A1 KuKa Radar Instrument Description

As discussed in the main body of the paper, the KuKa radar was built by ProSensing Inc specifically for the MOSAiC expedition. In the design phase it was decided to use a frequency modulated continuous wave (FMCW) radar, in order to maximize the range resolution for which the observed range interval is small, and where the high processing gain allows low peak transmit powers. The principle of FMCW radar operation have been outlined in a number of publications [Luck, 1949; Griffiths, 1990; Stove, 1992; Meta 2007; Cooper, 2011].

FMCW radar systems typically utilise a constant amplitude sinusoidal waveform that has been linearly frequency modulated in time, also known as a chirp. The KuKa radar transmits Ku-band waves at 12-18 GHz and Ka-band waves at 30-40 GHz, with bandwidths of 6 and 10 GHz for the Ku- and Ka-bands, respectively. The KuKa system uses separate transmit and receive antennas for both Ka- and Ku-bands that allows simultaneous transmission and reception of signals, and a 100% duty cycle. The chirp signal is digitally generated, amplified and then transmitted into the field of view by a transmit horn antenna. The instantaneous transmit frequency, $ft$, is given by $ft = fc + Bt/PRI$, where $fc$ is the carrier frequency, $t$ is time, $PRI$ is the Pulse Repetition Interval, and $B$ is the transmitted bandwidth, which (similar to a pulsed radar) determines the radar range resolution. Deconvolved data are used to reduce sidelobes caused by chirp non-linearity (see Section A1.4).

The reflected echos originating from objects within the antenna beam are collected by the receive horn antenna and then amplified by a low noise amplifier (LNA). For the case of a single object, the received echo signal will be a replica of the transmitted waveform but delayed by time $t$, due to two-way propagation $t = 2R/c$, where $R$ is the object range and $c$ is the free-space propagation velocity (the speed of light). After exiting the LNA, the received signal is first mixed with the transmitted signal generating the intermediate frequency (or beat) signal and then low-pass filtered.

The advantage of FMCW radars is that they allow much lower sampling rates (compared to pulsed radars) as the sampling rate need only correspond to twice the maximum beat frequency (real signal obeying Nyquist criterion) of the furthest object within the field of view. The KuKa system has a 14 bit ADC with a sample rate of 20 MHz, but the range window of interest means that this sampling rate can be reduced (after further low pass filtering) by a decimation factor of 16 or 32 for Ka- and Ku-bands, respectively, before digitization. The digitized beat frequency signals are then converted from their raw binary format into voltages with the signals from the different bands and polarisations/reference signals assigned to separate arrays (i.e. VV, HV, VH ,HH, cal, noise).

The system is configured to operate in both fine mode or coarse mode. The default setting is fine mode, operating at a bandwidth of 6 and 10 GHz at Ku- and Ka-bands, respectively. However, the full bandwidth can be processed within any segment to achieve any desired range resolution above 2.5 cm (Ku-band) or 1.5 cm (Ka-band). In the coarse mode, the range resolution processing is centered on satellite frequencies of CryoSat-2 and AltiKa at 13.575 GHz and 35.7 GHz, respectively, with an operating bandwidth of 500 MHz, yielding 30 cm range resolution. The KuKa radar was designed to be operated in a scanning mode, to simulate a scatterometer, and also in nadir mode to simulate an altimeter. In the scanning mode, the instrument was mounted on a pedestal that was fixed to a sled and deployed at each of the RS sites during Legs 1, 2, 4 and 5;

with an idle time of 30 minutes (Leg 1) and an hour (Legs 2, 4 and 5) between the scans. Scans were made between -30° to +30°azimuth range, and between nadir and 60°, at 5° increments. On 16 January 2020, an external trihedral corner reflector calibration was conducted during Leg 2 at the RS site, positioned $\sim$ 10 m in the antenna's far field region. Table S1 lists the measurement periods of KuKa during Leg 5. A total of 559 scans were obtained on Leg 5 between 25 August and 19 September 2020.

### A1.1 KuKa Radar Geometry

The antennas of each radar are dual-polarized scalar horns with a beamwidth of 16.5° (Ku-band) and 11.9° (Ka-band). The center-to-center spacing between the radar horns are 13.36 cm (Ku-band) and 7.65 cm (Ka-band). The horn spacing and difference in beamwidths means slightly different scanning footprint by the radar system by $\sim$ 60% (Ka-band) and $\sim$ 70% (Ku-band), with every 5° increments in incidence angle $\theta$.

Given the $\sim$ 1.6 m height between the Ka- and Ku-band antennas above the snow surface, the radial distance and range from the pedestal, and the footprint diameter and area increases from nadir to 50° incidence angle. Considering the 11.9° (Ka-band) and 16.9° (Ku-band) antenna beamwidth, the KuKa radar footprint on the snow is dependent on both frequency and incidence angle. This also means the incidence angle dependence on Ka- and Ku-band footprint overlap for a given radar 'shot'. Since the incremental separation between successive incidence angles is 5°, scans sample completely independent snow, every third azimuthal scan in the Ku-band, and every other scan in the Ka-band. Instrument footprint at nadir is 0.25 $m^2$ (Ka-band) and 0.4 $m^2$ (Ku-band); 0.75 $m^2$ (Ka-band) and 2.25 $m^2$ (Ku-band), at $\theta$=45°. In total, 3308 hourly scans were acquired during the year-long campaign. Table S1 summarizes the operating parameters for each Ku- and Ka-band radars.

### A1.2 Ku- and Ka-band NRCS Measurement

The KuKa radar records a data series of six signal states: the four transmit polarization combinations (VV, HH, HV and VH), together with a calibration loop signal (cal) and a noise signal (noise). Each data block with these six signals is then processed separately into range profiles of the complex received voltage, for each frequency through fast Fourier transform. The range profiles for each combination of polarization is power-averaged across the azimuth range, for each incidence angle.

We used the KukaPy python package to convert the Ku- and Ka-band raw data into calibrated polarimetric backscatter (VV, HH and HV). Within every incidence angle scan line averaged at 60°, Ka- and Ku-band VV, HH, HV and VH backscatter coefficients were derived from the second-order complex covariance matrix, assuming scattering from a surface ellipse, as defined by the antenna beamwidths. From the azimuth range-averaged power profiles, the NRCS was calculated following the beam-limited radar range equation by Sarabandi et al. [1990], given by:

$$fNRCS = \frac{8\ln(h^2\sigma_c)}{\pi R_c^4 \theta_3^2 dB \cos\theta} \frac{\overline{P_r}}{\overline{P_{rc}}} \tag{A1}$$

**Table A1.** List of measurement periods for the KuKa radar during Leg 5.

| *Activity - Device Operation* | *Timestamp* | *Action* | *Comment* |
|---|---|---|---|
| PS122/5_58-51 | 23.08.20 14:53 | recording start | Nadir stare mode; Wet ice surface, light snowfall |
| PS122/5_58-51 | 23.08.20 15:27 | recording end | end of nadir stare measurements without scanning |
| PS122/5_58-62 | 25.08.20 06:14 | recording start | Azimuth -50° - 50°, Elevation 0° - 60°, delta 3°, |
| PS122/5_58-62 | 28.08.20 04:41 | recording end | Changed azimuth to -40 to 40. |
| PS122/5_58-62 | 29.08.20 10:34 | recording start | Azimuth -30° - 30°; elevation 0° - 60°, delta 3° |
| PS122/5_58-62 | 04.09.20 07:35 | recording end | End of measurement at remote sensing site |
| PS122/5_58-62 | 04.09.20 09:00 | recording start | Measuring melt pond next to RS site. azimuth: -30° - 30°; elevation: 0° - 50°; delta: 3°; |
| PS122/5_58-62 | 04.09.20 10:14 | recording end | End of melt pond scan. Moving back to RS site. |
| PS122/5_58-62 | 04.09.20 10:28 | recording start | Back at remote sensing site. |
| PS122/5_58-62 | 04.09.20 23:13 | recording end | End of recording due to data storage issues. |
| PS122/5_58-62 | 09.09.20 00:20 | recording start | Azimuth: -30° - 30°; elevation: 0° - 60°; delta 3° |
| PS122/5_58-62 | 10.09.20 08:11 | recording end | Stopped scans to perform transect and melt pond measurements |
| PS122/5_58-62 | 10.09.20 10:09 | recording start | Melt pond scan next to RS site. Azimuth: -30° - 30°; elevation: 0° - 60°; delta 3°. |
| PS122/5_58-62 | 10.09.20 11:19 | recording end | Moved back to regular RS site. |
| PS122/5_58-62 | 10.09.20 11:43 | recording start | Restart of regular scans at RS site after transect and melt pond measurements. |
| PS122/5_58-62 | 17.09.20 08:36 | recording end | Stop of scan at RS site. Preparing for radar transect. |
| PS122/5_58-62 | 17.09.20 10:59 | recording start | Start of regular scan at RS site after radar transect. Azimuth: -30° - 30°; elevation: 0° - 60°; delta 3° |
| PS122/5_58-62 | 18.09.20 04:50 | recording end | End of regular scan at RS site |
| PS122/5_58-62 | 18.09.20 05:00 | recording start | Melt pond scans next to RS site. Azimuth: -30° - 30°; elevation: 0° - 60°; delta 3°. |
| PS122/5_58-62 | 18.09.20 06:16 | recording end | End of measurements at melt ponds. Back to RS site. |
| PS122/5_58-62 | 18.09.20 06:19 | recording start | Back to RS site. Azimuth: -30° - 30°; elevation: 0° - 60°; delta 3° |
| PS122/5_58-62 | 19.09.20 03:53 | recording end | End of measurements at RS site during MOSAiC leg 5. |

where h is the height of the antenna, $R_c$ is the range to the corner reflector, $\theta_3^2$ dB is the antenna half-power beamwidth (one-way), and $\overline{P_r}$ and $\overline{P_{rc}}$ are the power recorded from the footprint and corner reflector, respectively. We discard VH based on reciprocity observed in the cross-polarized channels (i.e. HV = VH).

Uncertainties in NRCS can be attributed to polarimetric calibration error (multiplicative bias) from metal tripod supporting the trihedral corner reflector, utilization of a finite signal-to-noise ratio (SNR), random error, arising from standard deviation in estimated radar return power, and approximation error arising from sensor and target geometry assumptions [Stroeve et al., 2020]. Detailed description of error analysis, along with detailed explanation of radar signal processing, calibration process and NRCS calculation can be found in Stroeve et al. [2020].

### A1.3 Ku- and Ka-band Waveform Analysis

For a single object, the sampled beat signal will have a frequency proportional to the range-to-object, and if there are multiple objects within the field of view, the beat signal will consist of a linear superposition of multiple sinusoids with frequencies representing the echos from different ranges. The mixing procedure in FMCW radars is equivalent to matched filter processing (pulse compression) used in pulsed radar systems; but with the difference that the point target response of a FMCW radar (after mixing) exists in the frequency domain rather than the time domain (as is the case for pulsed radars). To extract the frequency content of the received signals, and to create range profiles of power vs range, a Fourier transform needs to be taken of the beat signal; with the range to different objects within the field of view obtained from their corresponding beat frequencies, $fb$, following the expression, $fb = 2RB/cT$, where $T$ is the chirp period.

For the KuKa system, to obtain range profiles of relative power verses range, each column of the separate voltage data arrays are first windowed using a Hanning window and then zero padded. A column-wise FFT is taken after which these data are multiplied by a conversion factor: transforming the data into an uncorrected power spectrum. To correct for KuKa instrument drift, a correction factor is derived by comparing the values in the calibration data array with those collected under laboratory test conditions. In the next step, this uncorrected power spectrum is squared, the magnitude extracted and the drift correction applied forming a corrected power spectrum array. The data used in this ranging analysis consist of these corrected power spectrum arrays (one for each polarisation) where each of the array columns is a range profile of relative power that has a range bin spacing of 0.46 cm for Ka-band and 0.76 cm for Ku-band. All this data is stored in the NetCDF files output by the KuKaPy programs.

For this study HH polarization data, gathered with KuKa looking at $0^o$ incidence angle (nadir), and averaged over scans from -$30^o$ to -10 $^o$ (Ka) and $0^o$ to $30^o$ (Ku) azimuth angles were used. The full azimuth range was reduced in post-processing to avoid snowdrifts around the sled. Deconvolution was applied (see below) as set in the configuration files which control the KuKaPy processing; the information from these files is copied into the resultant NetCDF files for information as needed. The output NetCDF files contain the 'HH_power' and 'range' variables that were used to examine how the waveform shape changed before, after and during the ROS events.

### A1.4 Deconvolution

For FMCW radars, phase and amplitude errors on the transmitted chirp can degrade radar system performance and introduce range sidelobes [Griffiths, 1990, Budge and Burt, 1993, Piper, 1995, Ayhan et al., 2016]. This paper uses standard techniques to correct for both amplitude and phase non-linearities to mitigate against the sidelobes [Vossiek et al., 1996, Zhaodu et al., 1996, Meta et al., 2006, Dangler et al., 2007]. Sweep non-linearities are characterised from electrically smooth lead surface returns (collected on 23/24 January 2020) and compared with a unity amplitude ideal linear sweep. Non-linearities are corrected by applying the processing flow outlined in [Frischen et al., 2015] with an additional step that utilises high-pass filtering to mitigate the effect of the transmit/receive leakage signal [Stove, 1992]. These techniques will be explored in detail for the optimal case of: temporally coincident KuKa data and deconvolution waveforms in an upcoming publication [Newman et al., in prep]. As deconvolution waveforms from January 2020 were the only available option, and the radar response changes with time, the deconvolution is not optimal. Nevertheless, it provides some sidelobe reduction.

### A2 Surface Based Radiometer (SBR) Instrument

Microwave emission of the snow/sea ice surface were measured using the University of Manitoba surface-based-radiometer (SBR), which consists of three dual-polarized (vertical and horizontal) radiometers (19, 37 and 89 GHz) together with a GPS to record location and date/time. A camera is also housed in the system to capture images of the surface in the approximate field of view (FOV) of the radiometers. Each radiometer contains a 16-bit analog to digital converter, and is equipped with a Texas Instruments MSC1211 microcontroller.As discussed in the Radiometrics user manual, internal sensitivities are 0.04 K (19 GHz), 0.03 K (37 GHz) and 0.08 K (89 GHz).

The radiometers are designed to simultaneously make measurements at all three frequencies, with output provided in three individual ASCII text files stored on a USB and regularly downloaded during the MOSAiC expedition. However, there is a slight time-offset in each radiometer data file. To address this, the data are first resampled to a 1 minute temporal resolution and a moving average of 30 minutes is applied. In addition, during the MOSAiC expedition, the GPS on the instrument was not providing accurate date-tags and all date-tags had to be shifted to match the actual date the data were acquired.

During MOSAiC, a 80 cm tall pedestal was mounted to a 30 cm high komatik sled to hold the rotating positioning system on which the SBR was mounted. The instrument scanned across incidence angles ranging from 40 to 70$^{o}$ in increments of 5$^{o}$ degrees. The time to complete a full scan was approximately 5 minutes. At times the instrument was also set to collect data at fixed incidence angles only. Table A2 lists all the measurements acquired during leg 5 and when the instrument was scanning versus observing at a fixed incidence angle.

**Table A2.** List of measurement periods of the Surface Based Radiometer (SBR) during Leg 5.

| *Activity - Device Operation* | *Timestamp* | *Action* | *Comment* |
|---|---|---|---|
| PS122/5_58-86 SSMI_radiometer_SN002 | 08.09.20 06:08 | recording start | Start continuously scanning at RS site. Elevation from 40° to 120° in 5° steps. |
| PS122/5_58-86 SSMI_radiometer_SN002 | 09.09.20 10:29 | recording end | Stopped recording for calibration |
| PS122/5_58-86 SSMI_radiometer_SN002 | 09.09.20 10:41 | calibration | Calibration 19 GHz with low frequency absorber; Elevation 0°; Temperature absorber: -3.6°C; 10:41:15 to 10:43:15 |
| PS122/5_58-86 SSMI_radiometer_SN002 | 09.09.20 10:46 | calibration | Calibration 89 GHz with MT24 high frequency absorber. Elevation 0°; Temperature absorber: -3.4°C; 10:46:35 to 10:48:35 |
| PS122/5_58-86 SSMI_radiometer_SN002 | 09.09.20 10:53 | calibration | Sky calibration 120° to 180° elevation; end 11:00 |
| PS122/5_58-86 SSMI_radiometer_SN002 | 09.09.20 11:13 | recording start | Start of scan with elevation 40° to 60° and delta=5°. |
| PS122/5_58-86 SSMI_radiometer_SN002 | 12.09.20 09:42 | recording end | End of scan for data download. |
| PS122/5_58-86 SSMI_radiometer_SN002 | 12.09.20 09:53 | recording start | Start of measurements with constant elevation angle of 55°. Was planned as scan measurement. Not clear why scan did not start. |
| PS122/5_58-86 SSMI_radiometer_SN002 | 15.09.20 08:15 | recording end | Recording stopped. Could be due to power outage but not clear. |
| PS122/5_58-86 SSMI_radiometer_SN002 | 15.09.20 08:27 | recording start | restart scan after power outage. Elevation 40° to 70°; delta 5° |
| PS122/5_58-86 SSMI_radiometer_SN002 | 16.09.20 08:48 | recording end | End of recording due to power outage. |
| PS122/5_58-86 SSMI_radiometer_SN002 | 16.09.20 12:13 | recording start | restart scan measurements 40° to 70° after power outage. |
| PS122/5_58-86 SSMI_radiometer_SN002 | 18.09.20 04:20 | recording end | End of measurements during MOSAiC leg 5. For unknown reasons no photos were recorded. Radiometer data looks okay. |

### A3 Snow Pit extra figures

We include two figures, one comparing bulk microCT-derived density and specific surface area (SSA) with the density cutter results. The second figure shows profiles of specific surface area and density also shown in **Figure 4** as a function of date and location.

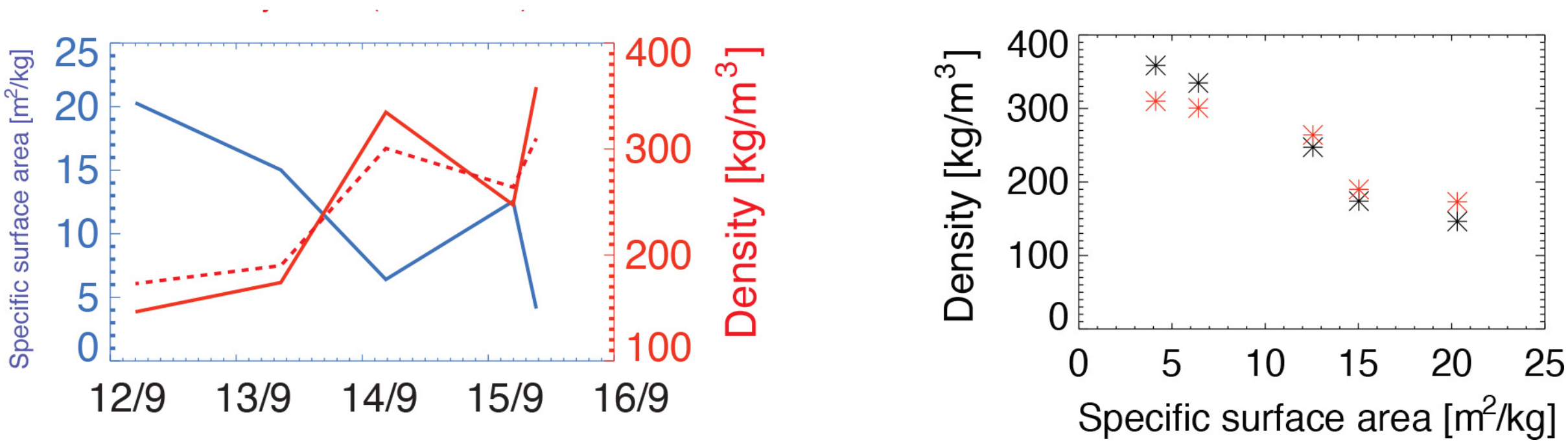

**Figure A1.** Depth-averaged microCT density and specific surface area (SSA) during the rain-on-snow event and comparison with density cutter. Image on left shows the average density (red line) and SSA (blue line) from microCT and density dutter (dotted line). Image on the right shows the SSA vs. density from microCT (black) and the density cutter (red).

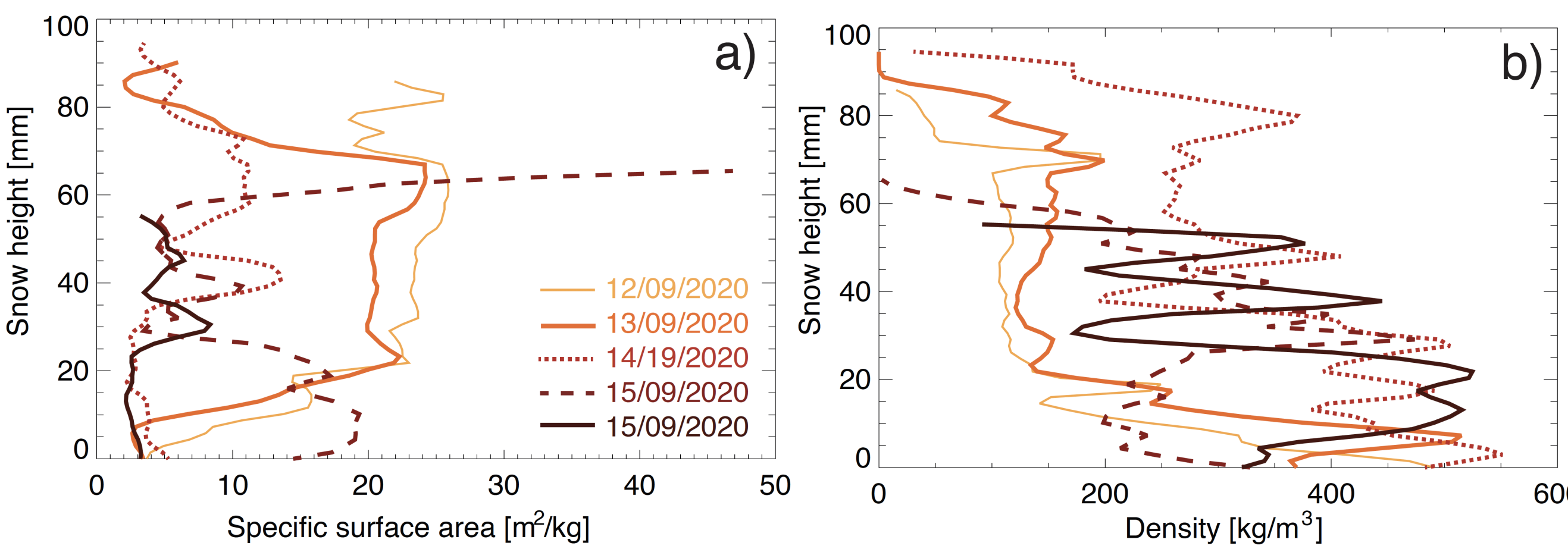

**Figure A2.** Profiles of specific surface area (SSA) (a) and density (b) before, during and after the rain-on-snow event. The bottom of the sample is represented by 0. Please note that the bottom of the sample is not necessary the snow/ice interface, as discussed in Figure 4 in the main manuscript.

## A4 ERA5 Atmospheric Reanalysis Data Processing

We analysed data on precipitation amount and type from the ERA5 atmospheric reanalysis at 0.25°spatial and 1-hourly time resolution. To characterise the precipitation at the floe we produced a timeseries of type and amount for the duration of the event (**Figure A3** (**a**) and (**b**)).

To characterise changes in ROS over longer time periods we analysed the same data during the cold-season (Oct-Apr) between 1980 - 2020 inclusive. After retrieving the data from the Copernicus Data Store, all values of total precipitation corresponding to places and times where the precipitation type was fully 'frozen' (snow or ice pellets) were set to zero. This left the total precipitation corresponding to 'wet precipitation', i.e. rain or mixed precipitation. We also identified all precipitation falling at a rate of $< 0.1$ mm per day and set this to zero. However, while trace precipitation is almost ubiquitous in ERA5 in the Arctic, almost all of this is frozen. The removal of trace precipitation did not therefore significantly impact our calculations of the total amount of ROS for each year. The remaining data was then averaged along the time-axis for each year in the relevant months, only where sea ice was present (concentration $> 50\%$) at the time of the wet precipitation. Sea ice concentration was downloaded with the ERA5 data and is derived from the EUMETSAT sea ice concentration climate data record [Tonboe et al., 2016]. This produced a single, two-dimensional distribution for each year corresponding to the total wet precipitation over sea ice at each grid cell. At this point all the seasonally-averaged grid cells (for each year) were spatially averaged by region (regions defined by the NSIDC; Fetterer et al., 2010). This spatial average was performed on an equal-area grid to match the grid of the mask and avoid the biasing effect of smaller ERA5 grid cells nearer the pole. This spatial averaging produced a single value for each year in each region corresponding to the average rate of wet precipitation when sea ice is present.

We finally produce an animation to illustrate the development of the weather system that brought the ROS event. To improve the visualisation of the event, we employed a more aggressive trace-precipitation masking algorithm, set at 5 mm/day. All remaining, non-frozen, non-trace precipitation is visualised with an hourly timestep (Movie M1).

All code for the processing and visualisation of ERA5 data in this paper can be found at https://github.com/robbiemallett/ros

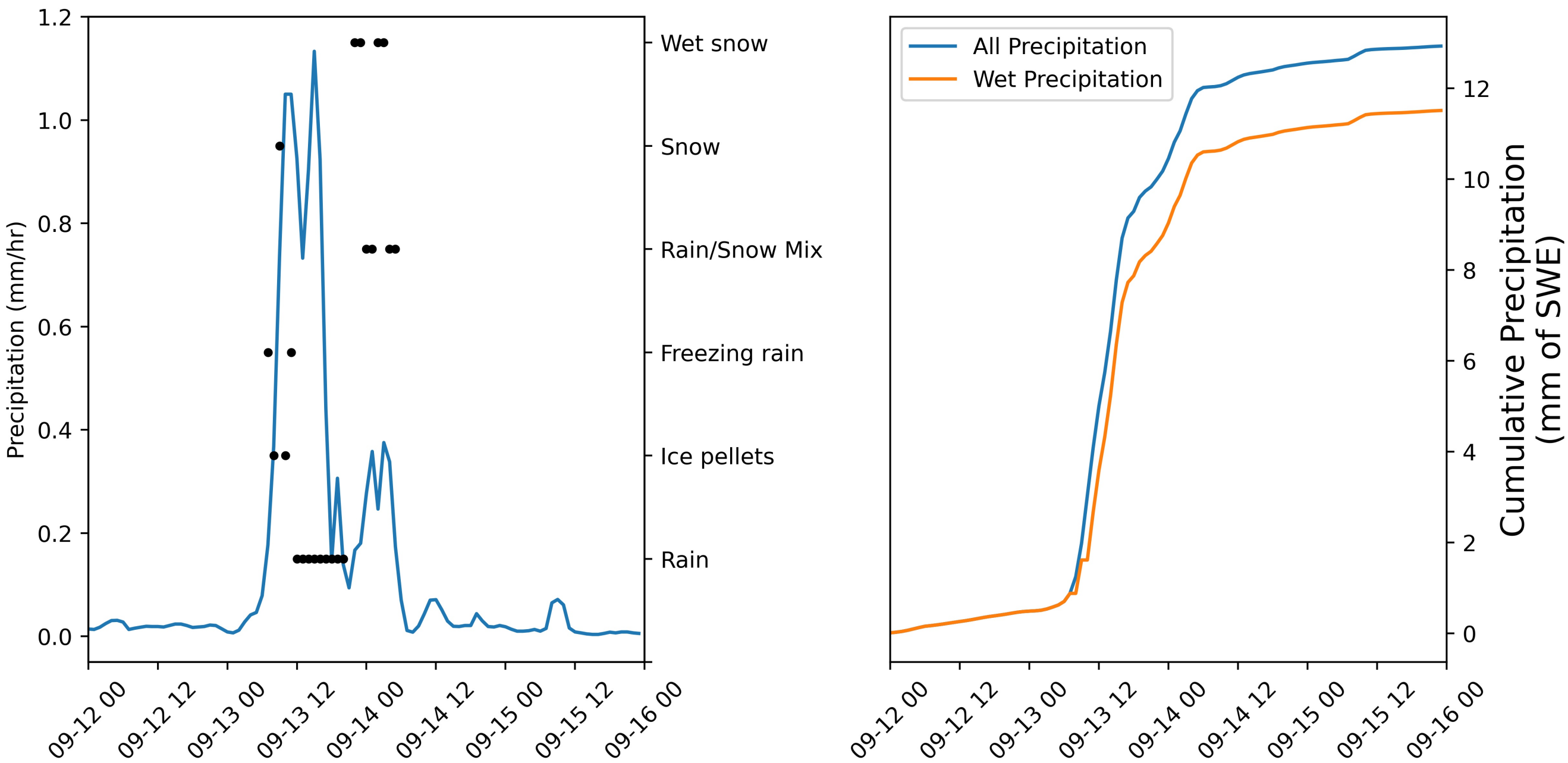

**Figure A3.** Rain-on-snow from ERA5 data (a) Time-series of cumulative precipitation at the MOSAiC floe (nearest 1°grid cell) separated into *all precipitation* and *wet precipitation*, where the latter corresponds to all World Meteorological Organisation precipitation types that are not snow or ice-pellets (b) Time-series of precipitation and precipitation type as in *(a)*.

*Author contributions.* Julienne Stroeve wrote the manuscript with input from co-authors and analyzed the SBR data and AMSR2 brightness temperatures. Vishnu Nandan processed the KuKa data in scan mode while Rosemary Willatt analyzed the KuKa waveforms as well as the CryoSat-2 waveform data (provided by Stefan Hendricks.) Thomas Newman performed the de-convolution of the KuKa data. Philip Rostosky provided calibrated SBR data and the impacts of the ROS on retrieved snow depth and SIC. Michael Gallagher provided the meterological data for the paper while Ruzica Dadic provided the snow data. Robbie Mallett processed the ERA5 reanalysis data. Andrew Barrett helped to improve the figure qualities. Mark Serreze, Rasmus Tonboe, Gunnar Spreen, Stefan Hendricks and Robert Ricker provided valuable editorial comments. Other co-authors helped collect data during MOSAiC.

*Competing interests.* None

*Acknowledgements.* This work was carried out as part of the international Multidisciplinary driftin Observatory for the Study of the Arctic Climate (MOSAiC), MOSAiC20192020 and was funded in part by the Canada 150 Chair program (#G00321321), the National Science Foundation (NSF) Grant #ICER 1928230, and the Natural Environment Research Council (NERC) Grants #NE/S002510/1 & #NE/L002485/1. Funding was also provided by the European Space Agency (grant no. PO #5001027396). VN was additionally supported by Canada's Marine

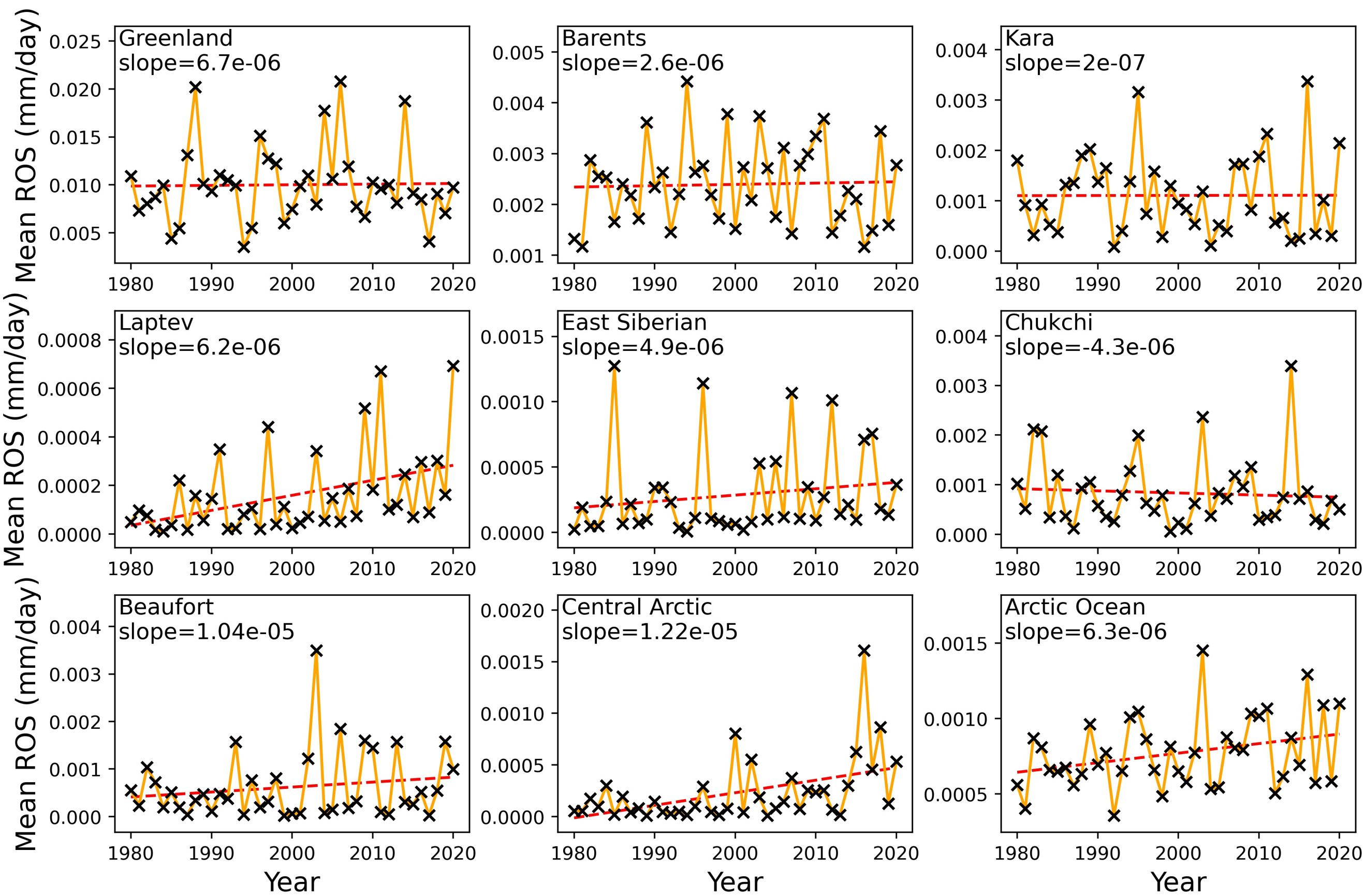

**Figure A4.** Regionally-averaged, time-averaged non-frozen precipitation (mm/day) over sea ice in the cold season (October through April)

Environmental Observation, Prediction and Response Network (MEOPAR) postdoctoral funds. MG was supported by the DOE Atmospheric System Research Program (#DE−SC0019251, #DE−SC0021341). PR, GS, and LT were supported by the the German Ministry for Education and Research (BMBF) through the MOSAiC IceSense project (#03F0866B) and by the Deutsche Forschungsgemeinschaft (DFG) through the International Research Training Group IRTG 1904 ArcTrain (#221211316). MS acknowledges Scanco Medical AG for providing their newest microCT90. MS, RD and AM were supported by the European Union's Horizon 2020 research and innovation program projects ARICE (#730965) for MOSAiC berth fees associated with the participation of the DEARice; and to Swiss Polar Institute grant SnowMOSAiC (#EXF−2018−003). We thank all persons involved in the expedition of the Research Vessel Polarstern during MOSAiC in 2019-2020 (AWI_PS122_00) as listed in Nixdorf et al. [2021].

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
