# Peer review of "Rain-on-Snow (ROS) Understudied in Sea Ice Remote Sensing: A Multi-Sensor Analysis of ROS during MOSAiC"

_The Cryosphere, 2021_

## Author Comment (AC1)

Review of

Rain-on-Snow (ROS) Understudied in Sea Ice Remote Sensing: A

Multi-Sensor Analysis of ROS during MOSAiC

by Stroeve, J., et al.

Summary:

Rain falling on snow changes its physical properties, thereby influencing its microwave signature with potentially far reaching consequences for the retrieval of several geophysical parameters from satellite microwave measurements. The present manuscript deals with presentation and investigation of multi-sensor observations of the impact of rain-falling-on-snow (ROS) events during the middle of September 2020 in the High-Arctic sea ice cover. Observations cover ground-based passive and active microwave measurements including altimeter-type measurements, supported by a set of comprehensive in-situ measurements of snow and meteorological parameters in the framework of the MOSAiC expedition. The manuscript provides a good overview about the various measurements, comes up with a set of interpretations of these measurements, and attempts to put these observations into a wider context, for instance by comparing them with satellite observations, estimating the impact of ROS on retrieval of sea-ice concentration, snow depth on sea ice and sea-ice freeboard.

The manuscript is a valuable contribution to the current state-of-knowledge and should be published in "The Cryosphere".

The manuscript would benefit from a number of clarifications and improvements, though, that I list in my general comments, being detailed further in my specific comments. In addition there are several editoral comments I would like the authors to pay attention to.

**We appreciate the reviewer's positive comment regarding the importance of the study and the careful review of our paper. The reviewer raises some excellent points and below we detail our responses/revision in response to the comments.**

General comments (GC):

GC1: The manuscript should be improved regarding the motivation and the suitability of using a late-summer / fall case as a surrogate for ROS events and their impact during winter / spring. The need for an improvement is given by:

1) Current sea-ice freeboard (and hence thickness) retrieval using satellite radar altimetry typically begins half a month to month later than the case investigated here.

2) The current motivation is build around winter/spring conditions.

3) The environmental conditions encountered during the case investigated differ considerably from those during winter / spring which to an unknown degree limits the relevance of the work presented here.

**The reviewer is correct that ice thickness is currently retrieved during the cold season only, which is not the same as the case studied here, though please note we only have a 2 week difference before ice thickness retrievals start. Our detailed observations of the ROS event are limited by the timing of the event that occurred at the MOSAiC floe and the instruments deployed at the time. However, as we show with the ERA5 atmospheric reanalysis data, ROS frequency and intensity is increasing over time during the cold season, and thus, we can expect such events may play an increasingly important role, especially as the Arctic continues to warm and more precipitation starts to fall as rain. Also, as we point out in the manuscript, it is not just ROS that is important, winter warming events such as reported in previous studies (i.e. Graham et al. 2017) also lead to liquid water and subsequent refreezing. We find it difficult to argue that conditions 2 weeks before the winter season "*officially begins*" are vastly different from early winter season conditions. Below for example are the temperature distributions from atmospheric reanalysis for mid-September, beginning of October and mid-October for the location of the MOSAiC over the 1979-2020 data record. The tails of the distributions overlap, and we also note that *in situ* observations of air temperatures before the ROS event and afterwards were below freezing and remained so until the ship left the floe on September 20. We further had new snow at the MOSAiC floe before the ROS event occurred, which could mimic conditions found early October.**

[Figure]

Further, while the snowpacks may be deeper in winter and the ice surface temperatures may be colder than studied here, the impact of liquid water percolating and refreezing within the snowpack will be the same (i.e. an ice layer could form, and/or other changes to the snow structure could occur). Specifically, water percolates very quickly deep into snow, as preferential flow paths are always forming (Avanzi et al., 2017). This is visible in the micro-CT images for September. So the water penetrates until all rain is frozen, typically about 10-20% of the surface is perforated by the preferential flow path. When we look at the latent heat, the penetration depth is about 5-10 times deeper than estimated by piston flow (homogeneous infiltration). In terms of a warmer or colder snowpack, the rain will freeze in a cold snowpack faster/shallower than in a warm snowpack, so the resulting ice layers/high density layers may be closer to the surface. Yet regardless of where the ice layer forms, ROS can lead to a permanent change in snow geophysical and thermodynamic properties irrespective of winter/summer conditions.

Overall, we disagree with the reviewer that the current study of an event that happened in mid-September is not relevant if the same event occurred two weeks later. We also do not hide the fact that the MOSAiC observations occurred earlier than the official "winter" season, and explicitly start section 6 with a discussion that the event we report on occurred at the end of the summer melt season, at a time before many sea ice variables such as ice thickness and snow depth are retrieved. We also mention that these events will have a similar impact on the snow cover, with liquid water percolation through the snowpack and refreezing, as well as other modifications to the snowpack . However, in response to this concern and the specific comment L67-70 we now add at the end of the introduction:

*"While this event occurred two weeks prior to when key sea ice variables, such as ice thickness and snow depth, are retrieved, similar modifications to the snowpack during the cold season are expected if the event occurred a few weeks later. Thus, this study has relevance to winter sea ice retrievals in the face of increased frequency and intensity of ROS and/or winter warming events."*

We further highlight now in the introduction that winter warming events are also important and the first paragraph now reads as:

*"Over the past 50 years, the Arctic has warmed three times faster than the planet as a whole [AMAP, 2021]. While this amplified Arctic warming is most strongly manifested in autumn as a result of summer sea ice loss [e.g. Serreze et al., 2009], recent studies have also documented an increase in frequency and duration of winter warm spells that can lead to air temperatures reaching 0ºC near the north pole [Graham et al., 2017]. At the same time, there is evidence that Arctic rain-on-snow (ROS) events during the cold season are becoming more common [Meredith et al., 2019; Liston and Hiemstra, 2011]. A recent study further suggests Arctic precipitation will increase more strongly than previously projected through 2100, with an earlier transition from snow to rain [McCrystall et al., 2021]. When rain falls on snow, or when air temperatures rise*

*above freezing, it can dramatically alter snowpack properties such as snow density, grain size, and snow water equivalent (SWE) content [Langlois et al., 2017; Grenfell and Putkonen, 2008]. Upon refreezing, ice layers may form within the snowpack. On land, ROS exacerbates flooding [Li et al., 2019], increases soil temperatures and snowmelt [Westermann et al., 2011; Putkonen and Roe, 2003; Rennert et al., 2009], while subsequent icy layers can impact wildlife, such as seal denning [Sterling and Smith, 2004], or inhibit foraging, leading to massive mortality of caribou, reindeer, and musk oxen [Putkonen et al., 2011; Forbes et al., 2016].*"

 GC2: The manuscript would benefit from relating observations made to what has been published elsewhere. This applies to the presented microwave measurements themselves (are these typical and/or realistic for the conditions encountered?), and this applies to the presented impact on various sea-ice parameters. Given the fact that conditions in the Southern Ocean are likely even more conducive for ROS events I highly recommend to include the other hemisphere into the discussion of your results. In this category I would also like to mention that the authors should  stress that their investigation about the impact on snow depth retrieval are hypothetical because the floe is not a first-year ice floe.

**We disagree with the reviewer that we should include the southern hemisphere in our discussion. While it is certainly interesting, there is not enough known about southern hemisphere snow and the much heavier snowpacks there already depress the ice freeboard below the ocean surface, leading to flooded sea ice and slushy snow packs. Since the reviewer is already concerned about "speculative statements",  and this would be a huge speculation especially given the vastly different estimates of precipitation from atmospheric reanalysis in that region (see Boisvert et al. 2020), we prefer not to add additional complication by also discussing the southern ocean. Instead we keep our focus on the Arctic region. We currently have KuKa deployed in the Weddell Sea so a forthcoming paper will focus on snow in that region.**

**Further, snow depth over FYI and MYI is retrieved with the Rotosky et al. algorithm, though MYI retrievals are only done in springtime due to the nature of the algorithm. However, the influence of the ROS event on retrieved snow depth should be similar in winter and also over MYI and FYI. Both the FYI and MYI snow depth retrieval algorithms take advantage of scattering of microwaves in the snow. Wet snow prohibits penetration, and thus scattering is almost 0. Also, for wet snow, the underlying ice type does not influence the microwave emission anymore since the penetration is limited to the upper few cm in the snow.**

**As for examples of Tbs published elsewhere, we can turn to Carsey's monograph chapter 4. There are many interesting figures in this chapter but looking at table 4-1 and figure 4-18, it is safe to say that the SBR Tbs are within the range of Tbs and emissivities measured by others, both in situ and from satellite. Also, 89 GHz Tbs of 160K is still within the range measured by others.**

GC3: The credibility of the results and their impact would benefit a lot from a better consideration and critical review of the uncertainties and limitations involved in the measurements themselves and in their interpretation. Examples of this are

1) Unclear location of the SSL and the SWE measurements with respect to it.

**The SSL was included in the SWE measurements made with the ETU SWE tube, and thus taken at the same location as all other snowpit variables. The SSL in the field is treated as a snow layer if it's soft enough to be sampled with our instruments. It is not possible to distinguish soft SSL and snow in the field. However, we have now redone the analysis to only use the Micro-CT data for the snow depth, density and SWE and we discuss in the text both values, as well as show the micro-CT only values in Figure 2.**

**We have clarified the SSL position in caption to Figure 4: "*To illustrate the changes in the snowpack and changes in the SSL thickness in this figure, the bottom of the snowpack is represented by 0. Above 0 is snow, and below is surface scattering layer (SSL)*."**

**We have further recalculated SWE also from the Micro-CT data and only for the snow (see responses further down).**

2) Unclear representativity of the KuKa-radar / SBR measurements site with respect to other snow measurement sites as well as a lack of information how that site actually looked during the measurements.

**The idea behind the snow sampling during MOSAiC was that representative snow pits were made around the MOSAiC floe and that these were generally representative for the snowpacks encountered. We briefly discuss this around L220. But with all that, there is still spatial variability that we cannot account for, especially around the sensors in the RS area as individual sensors created localized snowdrifts and scouring. In the case of September 15th, we cannot confidently distinguish between spatial variability between the FLUX and RS sites and temporal variability (FLUX pit was sampled in the morning, RS site in the afternoon). However, as we already mentioned, we can see from all profiles in Figure 4, that despite the potential spatial variability, the snowpack is relatively similar across the floe; for example, there is a denser layer at around 10 mm (see figure 4), which becomes emphasized after the rainfall, on September 14th. That layer is also visible in both profiles on September 15th. Figure 3 shows the surface conditions for all sites, and these conditions are representative of the conditions found at the RS site.**

**Below is a new figure we can add in the appendix material if the reviewer requests it, which shows the location of the KuKa radar and the RS snowpit. NOTE: this image is from 10.09.2022, but does show the spatial variability, and the influence of the RS instruments changing the deposition patterns.**

[Figure]

Leg 5, Remote Sensing area, 10.09.2022
Image courtesy, Ch. Finkebeiner and S. Graupener

snowpit RS

3) Sub-optimal treatment and discussion of the limitations of the KuKa-radar measurements in nadir-looking mode and the interpretation of the results obtained. While I acknowledge that ROS events such as the one observed certainly have an impact on radar altimeter measurements (and this is also stated sufficiently clear in the manuscript - but was kind of known before), the respective measurements and their interpretation do not back up this very well the way presented, leaving a lot of doubts in the capabilities of instrument and experimental set-up to detect the stated vertical displacement of the main scattering layer reliably.

**Thank you for highlighting the need to clarify the instrument information needed for this interpretation, and the fact that we appear to have over-emphasised the importance of the vertical displacement of the scattering layer.**

**In the abstract we state: '*During the Arctic Ocean MOSAiC Expedition, there was an unprecedented opportunity to observe a ROS event*' - i.e. this was an opportunistic set of observations that provided a fantastic timeseries of data over the particular area scanned by KuKa during the event. There are a couple of data gaps as acknowledged but we do not interpret your comment as referring to this as it is addressed elsewhere. The KuKa instrument provides 1.5 and 2.5 cm vertical resolution in the Ka- and Ku-bands respectively, as stated in the paper. This fine vertical resolution combined with fully polarimetric data in two highly-relevant frequency bands makes KuKa an ideal instrument for studying the effect of geophysical changes in the snow pack to radar scattering characteristics, and relating to Ku- and Ka-band satellite-borne instruments.**

We understand that KuKa detects a very small shift in the vertical position of the dominant scattering surface, and this isn't the focus of our results - it is the change in waveform shape that we wish to highlight. We have therefore amended the text in the paragraph starting on line 280, to clarify that it is the change in waveform shape, rather than the shift in the range to the peak, which is of key interest for comparison with satellite data. Specifically we now state:

*"A small shift in the range to the peak as shown in Figure 6 panels (c) and (d) could be caused by scattering from higher in the snow pack, or from the instrument sinking into the snow, or some combination; we cannot determine which and we therefore do not draw conclusions based on this observation, placing higher importance on the effect on waveform shape."*

GC4: I do understand that the authors would like to emphasize the importance of their findings and therefore, for instance, discuss issues like the potential impact on satellite active microwave (AMW) (scatterometer / SAR) observations and their interpretation, and show re-analysis based trends in ROS events. However, the impact on AMW observations and interpretation is not overly well elaborated, the ROS event trend analysis appears to be quite global, not taking into account that the case made here is from September while that trend analysis is for winter in general. Finally, the results presented are not that overwhelmingly convincing that emphasizing their value the way done appears to be a well-selected element for the discussion.

We were limited in the timing and location of the analysis based on when and where the event actually occurred during the MOSAiC expedition when we had several remote sensing instruments deployed. That does not mean that the impacts observed both at the satellite level and in situ are not applicable at other locations or during other times of the year. We clearly show that for this ROS event and subsequent refreezing, impacts were seen both in the in situ data and in the satellite data, so impact is not a question here. We additionally acknowledge in section 6 that this event occurred before ice thickness and snow depth are retrieved, but since we see that ROS events in the cold season (Oct-April) are increasing in frequency and amount, and with statistically significant changes in the central Arctic Ocean, as well as the Kara, Laptev and East Siberian seas (i.e. we show a spatial trend map in addition to Arctic Ocean averaged values), it is likely that these regions may have times when ROS will impact satellite retrievals during the winter season. Previous studies have also documented an increase in frequency and intensity of winter warming events that would also result in liquid water in the snow pack. It is unclear what the reviewer is really concerned with here. Our focus on October to April with the reanalysis ROS detection was purely to coincide with the time over which ice thickness and snow depth are retrieved from satellites to show that ROS could be something to consider in regards to uncertainties in satellite-based retrievals. Remember, this is a discussion on potential implications. A follow-on paper will look at reanalysis to find examples of specific ROS events over sea ice and evaluate their impacts on satellite data directly, yet we will not have corresponding snow pit analysis to verify modifications in the snowpack like we have with the MOSAic expedition. Thus this first study provides some of the baseline understanding that we can then scale up to detected events with satellites in winter.

In regards to impacts for AMW, the KuKa radar results clearly show first-hand understanding of large relative changes in radar backscatter (by many order of dBs) during pre and post ROS events, at both frequencies, incidence angles and all polarizations, relevant to and mimicking system parameters on board satellite radar altimeters and SAR/scatterometers. The radar observations are also strongly supported by in situ snow property (density, temperature, microstructure) changes showing liquid water percolation and snow refreezing. Of course, we did not conduct destructive sampling of the RS footprint to illustrate snow property measurements. However, we do show snow geophysical changes from representative snow pits at multiple locations around the MOSAiC floe, including the RS site - demonstrating consistent ROS effects on snow geo- and thermophysical conditions across scales.

As for the satellite-based scatterometer/SAR implications, we have described how ROS events, followed by refreezing, can almost permanently change the geo- and thermophysical state of the snowpack, irrespective of snow conditions during winter or during the melt period. So beyond a doubt, based on the large magnitude of KuKa backscatter responses, we believe that the ROS effects on the snow pack will be much stronger on scatterometer/SAR backscatter, compared to melt-onset detection, where previous studies using SAR/scatterometers have been used to effectively detect melt-onset. The ROS impact on snow and resultant backscatter will also vary as a function of sea ice type (eg. FYI and MYI), depending on the presence/absence of salinity in the snow pack (via modifications to brine volume, dielectrics etc). We plan to conduct a follow-on analysis to track winter ROS events from scatterometer/SAR data from previous years from regions that has satellite data coverage (unlike MOSAiC floe located close to satellite pole hole), to quantify relative changes in backscatter pre- and post-ROS events. Therefore, observations from the KuKa radar provide us with process-scale understanding that we can use for reliable detection of ROS events from satellite-scales in winter. To show the importance of our study at satellite-scales, we have rewritten the SAR/Scatterometer implications section as the following below:

*"Figure 5 shows a strong reduction in KuKa radar backscatter when there is liquid water in the snowpack and an increase in backscatter as the snowpack refreezes - visible in both frequencies and incidence angles, and all polarizations. Since the change in backscatter during rainfall is large compared to normal day-to-day backscatter variability, this information can be used to detect ROS events from spaceborne scatterometry and synthetic aperture radars (SAR) and differentiate between ROS-induced melt and naturally occurring melt-onset events, that presently depend on time series threshold techniques to determine the timing of spring/summer melt onset [e.g. Mortin et al., 2012].*

*The effects of ROS on scattering after the snowpack has refrozen could also be significant. Over FYI, if an ice layer forms right above the sea ice, it would constrain upward brine wicking through the snowpack. Formation of ice lenses within the snowpack and/or air-filled vertical ice channels, inclusions, or poly-aggregate snow grain clusters of various sizes [Colbeck, 1982b; Denoth, 1999] may also occur. While ice lenses facilitate additional surface scattering, vertically oriented ice channels produce more volume scattering, especially at large incidence angles. A more extreme*

*impact could be complete melting of the snow cover from the rainfall, leaving bare ice, resulting in ice surface scattering.*

*Overall, ROS and subsequent refreezing can create geophysical and thermodynamic changes to the snowpack at various spatial scales, leading to complexly-layered snowpacks, with changes in salinity- and temperature-dependent brine volume (for FYI) [Nandan et al., 2020], density [Denoth, 1999] and snow grain microstructure [Colbeck, 1982a], all altering surface and volume scattering contributions to the total backscatter. These modifications will in turn affect the reliability of SAR and scatterometer algorithms to accurately retrieve snow/sea ice geophysical properties, classify FYI and MYI types [Nghiem et al., 1995] and timing of melt onset. Our results highlight that modifications during and following ROS events and its effect on microwave scattering at these frequencies is not trivial. Overall, our results provide a detailed and first-hand understanding of the geophysical impacts of ROS and its effect on the radar scattering behavior. These findings will help us to further aid interpretation of radar backscatter changes due to ROS events at satellite-scales."*

Specific comments:

L58-62: There are a few approaches to derive melt onset on sea ice in the Southern Ocean based on satellite microwave imagery that should perhaps be mentioned here as well (Willmes, S., et al., 2009, doi:10.1029/2008JC004919 and Arndt, S. et al., 2016, doi:10.1002/2015JC011504 )

**We appreciate the reviewer's concerns about the southern ocean, but we feel including a discussion on the southern ocean is outside the scope of the present study. We currently have KuKa deployed in the Weddell Sea so a follow-on paper could focus more on the southern ocean and then we can include these important references then.**

L67-70: Between this last paragraph and the previous paragraphs or at the end you should perhaps write something about the fact that most (if not all) studies you cited so far, were dealing with cold season / winter and/or winter-spring transition conditions. In contrast, the data you are dealing with during MOSAiC are from a completely different season with also completely different physical properties of the sea ice underneath. Here you are dealing with end of summer / commence of fall freeze-up. I am sure you will get back to this inconsistency in environmental conditions later in the paper. But it will be very helpful if you prepare us, the readers, for the fact that you attempt to further knowledge about ROS events by using late summer / early fall conditions as a surrogate for winter/spring conditions.

**We disagree with the strong comment of a completely different season; it is only 2 weeks prior to when research groups start to retrieve ice thickness with CryoSat-2 (see responses to previous comments on this topic). We believe we cannot say that conditions in mid-September are vastly different than at the beginning of October (see temperature figure posted above in response to general comments).**

However, as discussed previously in the response to general comments, we have now added this sentence at the end of the Introduction: "*While this event occurred two weeks prior to when key sea ice variables, such as ice thickness and snow depth, are retrieved, similar modifications to the snowpack during the cold season are expected if the event occurred a few weeks later. Thus, this study has relevance to winter sea ice retrievals in the face of increased frequency and intensity of ROS and/or winter warming events.*"

Figure 1: This is a busy figure that contains a lot more information than is relevant for this paper. I suggest to get rid of all the unnecessary information to be able to concentrate on the conditions encountered at the RS site

**We respectfully disagree. This photo provides an overview of the entire MOSAiC floe during leg 5 and shows where all the snow pit observations were made (yellow areas), as well as the location of MET City relative to the Remote Sensing Site, which are relevant to the study and the data we present.**

L91/92: You refer to a calibration of the KuKa radar during leg 2 here. How about during leg 5? Is the radar that stable that it did not need a re-calibration even though it was unmounted during leg 4 and then deployed again for leg 5?

**According to the manufacturer, the calibration should be stable for long periods. No calibration was made during leg 5 by the crew manning the RS site. However, we saw from leg 2 that the calibration was stable over the entire leg 2 time-period. Comparison of calibration coefficients by the manufacturer using calibration data conducted during Leg 2 and after MOSAiC show that the instrument was stable throughout the expedition. Further, calibration impacts the absolute values of sigma0, but we are focused on the relative changes in backscatter and radar waveforms, and these are large changes (many orders of dB) from the ROS, relative to any small calibration changes or small backscatter offsets.**

- You are pointing out the antenna's far field. Where does that begin? Possibly close enough to the antenna that both regular measurements and calibration measurements were carried out in the far field?

**The antennas were selected such that the Fraunhofer far field distance at both frequencies is about 0.95 m (Ka-band) and 1.01 m (Ku-band).**

 - Given the height of the antennas above the ice surface of about 1.6 m (see A1) I can guess that the calibration measurements were carried out by pointing the antennas such that they looked parallel to the surface and that the corner reflector was mounted on a tripod at exactly the same height as the antennas such that it opened into the direction of the antennas. Is that correct? It would not hurt to mention this detail, I think.

**The software has an automatic calibration routine that scans the scene to find the peak corner reflector signal. The height of the reflector is not the same as the height of the antenna. As**

long as the corner reflector is pointing in the general direction of the antennas-within +/- 15 degrees or so, the radar cross-section of the reflector should be close to the theoretical value. (This is a bit complicated since the radar cross-section varies as 1/lambda$^2$, lambda=c/freq and the frequency varies from 12-18 GHz at Ku-band and 30-40 GHz at Ka-band. We use the average frequency to find lambda, which is reasonable). This detail is now added in the appendix.

L96: "at nadir and at 45 degrees"

**Done**

What is the motivation to focus on an angle of 45 degrees? Is this the common angle currently used by spaceborne Ku-Band scatterometers? It might be useful to tell the reader.

**Yes, we already stated that on L97:** *We focus our analysis on the normalized radar cross-section (NRCS) values at nadir and at 45º, mimicking θ of satellite radar altimeters and microwave scatterometers*.

L117: What is the resulting height of the antenna above the surface then?

**The distance between the bottom of the positioner to the rotational axis (where we attach the boxes with the antennas) is 30cm. Thus the total height was 110 cm. We have added this detail to the revised manuscript.**

L127/128: What was different in the calibration of the 89 GHz SBR channel using the absorber between the attempts during leg 3 and leg 5 that those during leg 5 were not useable? Or in other words, what caused the calibration during leg 3 to be realiable?

**During the calibrations, several cold-sky measurements were performed. We noticed a large scatter of cold-sky brightness temperatures for leg 5, even after atmospheric correction, leading to high uncertainties in the linear fit. During leg 3, a different absorber material was used and the cold-sky brightness temperatures are much more aligned. Therefore we trust the leg 3 calibration more.**

**This is also supported by the regression coefficients we derived for each leg (see below). We expect slope and intercept close to 1 (like for leg 3), but that was not the case for 89 GHz during leg 5. However, the absolute calibration is not the important part of the study, rather the relative changes in the Tbs. We now state this is a *relative* calibration (not absolute), which is also one of the reasons Tbs approach 274K during the rain.**

*Leg 3*
*Tb 19V: Slope=0.983, Intercept=6.83*
*Tb 19H: Slope=0.981, Intercept=0.36*
*Tb 89V: Slope=1.034, Intercept=-0.83*
*Tb 89H: Slope=1.033, Intercept=-2.35*

*Leg 5*
*Tb 19V: Slope=1.025, Intercept=4.96*
*Tb 19H: Slope=1.023, Intercept=-2.48*
*Tb 89V: Slope=1.172, Intercept=-10.63*
*Tb 89H: Slope=1.113, Intercept=-7.05*

L130/131: I note that the data gap is substantially longer for SBR than the KuKa radar; I suggest to reformulate this accordingly. While the power outage of the KuKa-radar coincides with the worst ROS conditions that of the SBR is not linked to that.

**Instruments had different lengths and timing of being offline from a power outage. This could be partly a result of the fact that the KuKa radar had a UPS, and it was able to record for a longer period of time than the SBR. Therefore, the UPS ran the KuKa system the moment the RS site lost power on the 12th. So SBR goes off, KuKa runs with UPS power throughout the 12th to until 0850 hrs on the 13th. Kuka loses UPS power until 1300 hrs on 13th after which the RS site power came ON and both KuKa and SBR continued to make observations. We have now clarified that in the respective sections for KuKa and the SBR.**

- In addition I note that this power outage is not reflected in Table A2 which content suggest continuous data acquisition from Sep 12 to 15. This should be changed for consistency.

**Table A2 shows the field notes for the scan periods of the instrument. Even though SBR was started on September 12 a gap occurred in the recording of the data. We do not fully understand why this data is missing in the data files and can only suspect that it is due to a power outage. To be consistent we have changed the table header:** *List of measurement periods of the Surface Based Radiometer (SBR) during Leg 5. Note that no data was recorded between September 12 10:38:53 and September 13 09:54:10.*

L132/133: Would it be helpful for other scientists to learn what you consider "unstable" in this context?

**The 37 GHz channel had unrealistic Tbs throughout most of the expedition, and we were not able to fix it during our time on MOSAiC. Here is an example of the instability, such as what we observed during leg 2. We believe the reviewer will agree that this channel is not providing realistic data.**

[Figure]

[Figure]

L137-139: One more sentence describing how this pluviometer deals with the different forms of precipitation and what the measurement principle is (Is it heated? Is it just detecting the impact of the precipitation particles?) would be appreciated in addition to the reference Wagner et al. [2021].

**The pluvio is a weight based rain gauge that uses a number of methods to ensure accuracy. The shape of the collection bucket is an important factor, there is indeed a heater although it was off during this period. We've added this sentence: "*The Pluviometer is a sophisticated rain gauge designed to calculate precipitation rates based on accumulated mass. It measures precipitation falling on a 400cm²….*"**

L152-159: Please provide the grid resolution and the forecast interval that you used. I assume ERA5 provides a 6-hourly forecast of the precipitation? I assume you used all four and computed the daily total?

**ERA5 precipitation is at hourly resolution and the grid is 0.25 x 0.25º resolution. We added that now in the description starting after L130: "*Hourly ERA5 data at 0.25x0.25º resolution is used here to assess trends in cumulative, cold-season (October through April), non-frozen precipitation over the Arctic Ocean between 1980 and 2020 inclusive.*"**

- I note that Leg 5 took place August/September while here you refer to the cold-season and/or wintertime precipitation. You should perhaps to provide a better link between the observations carried out during and ERA5 data used for the MOSAiC Leg 5 on the one hand and this investigation of the cold-season ROS events based on ERA5 on the other hand; please define clearly what you understand by "wintertime" or "cold-season".

**Yes, we use ERA5 to assess changes in cold-season precipitation as part of our motivation to see how these events are changing during the time of year that ice thickness and snow depth are retrieved from satellites. We specifically start L130 with the statement that we want to expand our study beyond the time-period of this particular ROS event. Thus, the reviewer's concern of not providing a link with our study is unclear. We also clearly show that ERA5 reliably captured this particular event in September 2020, lending confidence that ERA5 could be used to also assess changes in October through April. Wagner et al. (2021) further evaluated ERA5 over a longer time-period and during the winter season and also concluded it was suitable for representing precipitation events throughout the MOSAiC expedition. As mentioned in response to the previous comment we now state the cold season is October through April: i.e. see above new sentence in response to previous concern.**

L164: "around the MOSAiC floe" ... or "on the MOSAiC floe"?

**Across the floe seems more relevant, as we are stating that surface conditions were similar across the entire floe (which of course is also on the floe). No change.**

- I note that, aside from showing Figure 1, you did not comment and/or describe the surface conditions the SBR and the KuKa radar were looking at. It remains hence unclear how representative the snow pit measurements are with respect to the conditions within the field-of-views of the used instruments.

**We show all the snow data collected around the time of the ROS event at all snow pit locations (see yellow areas in Figure 1), and we show the photographs of the surface conditions as well as detailed analysis from the snow pits. Detailed snow pits were made each day at different locations, thus we do not have snow pits every single day from the RS site and of course we will not have snow pits directly under the KuKa or SBR instruments. However, we see general consistency in the snowpack conditions between snow pits from locations across the MOSAiC floe, suggesting that the snowpack at the different sites was broadly similar and thus these provide for an assessment of conditions the KuKa and SBR instruments viewed. We have to work with the data at hand, and the sampling plan during MOSAiC was to use representative snow pits around the floe rather than daily samples at the same location. This was a necessary tradeoff for the work effort required and to meet all science objectives for the campaign.**

- You state "routine snow pit observations" but I missed an information about the sampling; sub-daily? daily? 3-daily? Depending on / triggered by precpitation events?

**Routine snow pit observations were defined as part of the MOSAiC sampling plan. This is detailed more in Nicolaus et al. (2022). Specifically Figure 3 shows the temporal sampling done before, during and after the ROS event. This was done every day at specific snowpit sites (yellow areas in Figure 1 where the full snowpit sampling took place), and several observations were made along transects for quick snowpits (i.e. just depth, SMP, salinity and temperature). We have observations at least twice a week for every snow site. We added "*Routine snowpit observations were collected at least twice a week per location*" in the first paragraph on the snow data collected.**

- Didn't you perform any observations of the crystal structure of the snow following the Colbeck classification?

**No, we did not follow the Colbeck (or the more updated International Snow Classification by Fierz et al.). The "snow shape" classification is not quantitative and it would have been difficult to guarantee a consistent data set between different (subjective) observers during MOSAiC. If needed, we can determine the "grain shape" from the 3-D microstructure measurements, but we don't see how this information is relevant here. The pictures in Fig. 3 can also be used to show that the snow particles in all snowpits are rounded and wet at the surface. On the 15th, there is a trace amount of decomposed particles, which is also visible in Figure 4. Furthermore, the shape of snow particles is not relevant for the KuKa signal, which has the wavelength of ~20 mm and ~10 mm, respectively. It is in fact, remarkable, that even with such a thin snowpack, the KuKa can detect the ROS event with high confidence.**

L166/167: Sorry to ask but how was the SWE measured? Was it measured for the 3 cm cut-out samples? What happened (in your case of a 7 cm thick snow cover) with the bottom 1 cm?

**L167 describes how the SWE, that is used in the original manuscript, is measured. We are now calculating the SWE using microCT density data, so we can separate the changes in the snow SWE from the changes in SSL thickness. Figure 2, and the text has been changed accordingly. Specifically the snow pit section has been rewritten as follows:**

**"*Routine snow pit observations were collected at representative locations around the MOSAiC floe at least twice a week per location (see Figure 1 for locations). Density cutters (volume = 100 cm³) were used to measure snow density at 3 cm vertical resolution. Snow samples collected using the density cutter were further bagged and melted to room temperature for measuring salinity (ppt) using a YSI30 conductivity sensor (resolution = 0.1 ppt and accuracy = ± 2% or ± 0.1 ppt). Needle-point thermometers recorded temperature at the snow surface, snow/ice interface and at 5 cm intervals in between. Snow height was measured during snowpit sampling. However, during summer it is difficult to distinguish between the soft surface scattering layer (SSL) and snow in the field. Thus, the snow height and the density cutter measurements may include the SSL. Therefore, the snow height, which we discuss below, is calculated from the Micro-CT, as the distance between the surface and the snow/SSL interface.***

*A micro-computed tomograph (Micro-CT) (Micro-CT 45/90 computer tomography scanner from Scanco Medical AG) measured the 3-D snow structure [Hagenmuller et al. 2016]. Using the snow microstructure data, snow density ($\rho_{snow}$) can be derived as:*

*$\rho snow = V_{ice} / V_{total} \times \rho_{ice}$ [Lagagneux et al., 2002, Hagenmuller et al. 2016], where $V_{ice}$ is the ice fraction volume, $V_{total}$ the total sample volume and $\rho_{ice} = 917$ kg/m$^3$ the ice density [Kerbrat et al. 2008; Hagenmuller et al. 2016]. We also compute the specific surface area (SSA) as: SSA = $SA_{ice}$ / ($V_{ice} \times \rho_{ice}$), where $SA_{ice}$ is the ice surface area. Density-cutter and Micro-CT-derived bulk $\rho_{snow}$ are in good agreement, with differences of up to 15% (see Figure S1). We also use the Micro-CT data to derive the SWE for snow only (i.e. excluding the SSL). This was done by integrating the SWE for each Micro-CT profile point, with SWE =\sum_{s}^{int}\*z\*\rho\textsubscript{snow}/1000, where z= layer thickness/resolution = 1.445 mm) between the snow surface (s) and the snow/SSL interface (int) (Figure 4). The snow/SSL interface was determined visually from microCT images."*

L203-206: What is the scientific rationale to include the surface scattering layer (SSL) into the SWE measurement? Did you cross-check the SWE measurements by simply computing SWE using density and depth of the snow? I get about 12 mm SWE and 14 mm SWE for a 7 cm thick snow cover and the bulk densities given by you.

**It is impossible in the field to distinguish where snow stops and the SSL begins, so it was sampled as part of our snow sampling. The distinction between SSL and snow is only possible because we have detailed microstructural information from the Micro-CT, which we now use to recalculate the SWE of snow only, without the SSL. We now get values similar to what the reviewer has calculated above using densities.**

**Also, while the SSL is not of meteoric origin, but rather modified sea ice, the physical properties of softened up SSL are much closer to snow than to sea ice. That said, the SSL could be a reflective horizon for the KuKa.**

 - Apart from that, I doubt that the comparably small increase in density visible in the respective panels of Fig. 4 is responsible for a doubling of the SWE. It is kind of clear that almost 3 hours of rain has caused a certain mass gain but why is that not yet visible in density or SSA? One could hypothesize that the classical way to estimate SWE from snow depth and density fails because there is too much interstitial liquid water between the snow grains.

**We have now calculated SWE from Micro-CT data, and excluded the SSL. The change in (only) snow density between the 12th and the 13th (at the beginning of the rainfall) is ~20 kg/m$^3$, reflecting a slight density increase at the surface only because of the rain. The snow density from the 13th to the 14th increases by ~130 kg/m$^3$, clearly showing a wetter and compacted snowpack due to rain. There is also some small-scale spatial variability of snow in every snowpit, so even measurements from the same snowpit can have different snow thickness. The interstitial water in the snow is maximum 10%.**

L211/212: Why did the thickness of the SSL increase under the warmer temperatures? How warm did the SSL (or ice/snow interface) get? In L166 you write that you measured the ice/snow interface temperature. Also: How do you know that the SSL thickness increased? Do you have Micro-CT measurements that go as deep below the ice-snow interface as in the middle profile in Fig. 4 also for the two left profiles (ROV, ALB)?

**If it's warmer, the sea ice interface melts and the SSL becomes soft (because the ice crystals have liquid water between them) and it can be sampled as part of the snowpack. Because of the hot air intrusion and the rain event, the snowpack and ice surface had become warmer, thus softening the SSL layer.**

**The Micro-CT measurements for this event are as deep as we could sample without drilling. While we were able to drill and sample the hard sea ice in certain snowpits, the drilling of the extremely wet snowpack during this event was not possible. The middle profile from September 14th had a thick soft SSL, which is why it is part of our sampling (e.g. it could be sampled without drilling). For the ROV and ALB profiles, the SSL was still frozen, so we could only sample the thin soft layer of it, as seen in Figure 4. We added the sentence "*The snow-ice interface temperature was slightly negative (-0.5°C) on the 13th, 0°C on the 14th of September and negative on the 12th and the 15th." in the text. This indicated the warmer snow-ice interface and the softening of the SSL.*"**

L214: Such considerable differences in SWE are also observed earlier between the Kuka radar PIT and the coring site; you don't explain those. Why?

**The difference in those two snow pits could have been due to slight changes in freeboard (coring site was on thicker multi-year ice, while the KuKa transect was on thinner ice), which could have caused the difference in "slushiness" and density. Also, the snow depth at the Coring site was ~8cm, and at the KuKa site only ~3.5 cm. The density at the bottom of the Coring pit was almost the same as the density at the KuKa site. We added the sentence "*Local ponding and smaller snow height could also explain the higher density at the KuKa pit."***

- I might be wrong but, despite the fact that the SWE measurements could be really helpful to understand differences in the mass accumulation at the surface, my impression is that it is not sufficiently well clear how much of the SSL underneath the snow cover is included in the SWE measurements, for basically all examples shown.

**This is a valid point, and we already discussed above that we can't distinguish between SSL and snow using traditional methods. We recalculated SWE and density from micro-CT using snow only.**

- What I also observe is, that the profiles in Figure 4 appear to be, frankly speaking, randomly put with respect to where the snow cover begins (i.e. 0). It is actually not clear how thick the SSL is. It is absent completely in the 4th profile (FLUX) while it is hard to delineate where the SSL begins in the 5th profile (RS). This is associated with a substantial difference in snow depths: 6 cm for FLUX (at

least, because we don't see the SSL) and 2 cm for RS (if the location of where the SSL begins is correct).

**The profiles are not randomly put with respect to where the snowcover begins. It is clear from the microstructure where the snow/SSL interface is. Even in the RS profile, we can distinguish between coarse wet snow particles and SSL because the SSL shows an anisotropy. We added the sentence "*Figure 4 also shows the clear differences between snow and the SSL (defined as zero in Figure 4), with the SSL having a coarser and anisotropic structure reflecting its origins as sea ice*."**

L220-223: I am sorry but I neither see a clear indication of that higher-density layer about 1 cm above the snow ice interface in the profiles shown in Fig. 4 nor do I understand from these what may indicate internal ponding. Again the question pops up what the snow-ice interface temperature might have been.

**We changed "1 cm" to "10 mm" to be consistent with the figure, and added that it's the white line that shows the density "blip" at around 10 mm. Hopefully this will make it clearer to see. The snow-ice interface temperature was slightly negative (-0.5°C) on the 13th, 0°C on the 14th of September and negative on the 12th and the 15th. We added this information in the revised manuscript.**

- I suggest to remove the salinity observations as long as these are not of further relevance for the study because the jump between the left 3 and the right 2 measurements is visually quite large. But you write it is small and falls within the accuracy with which it could be measured. Then these salinity measurements seem to be a bit misleading here.

**Done**

L231/232: "porous ice layers in the snow volume" --> What makes you think that at the frequencies used here, for reasonably dry snow conditions before the ROS event, the majority of the backscatter is not (also) caused by the sea ice underneath the only 7 cm thick snow over? Also: Which ice layers in the snow volume are you referring to here?

**We agree with the reviewer on the speculation. We have removed the 'porous ice layers in the snow volume' in the revised manuscript.**

L239-242: I don't think that what you write here is well illustrated by the data shown. Firstly, I doubt you can speak of the funicular regime here (I guess Garrity, 1992 in the Book "Microwave Remote Sensing of Sea Ice" mentioned that from her field work). The micro-CT images do not show that. In particular do the 2nd and 3rd micro-CT profile essentially show the same distribution of bluish gaps near the surface and there is not overly much difference further inside the snow pack.

**The second and third microCT profiles do not show the same distribution of blue gaps near the surface. The gaps are smaller. The density throughout the whole profile, but also close to the**

surface, is clearly larger on the 14th. The snow particles are also larger with a smaller specific surface area, making water percolation easier. Colbeck, 1982 show wet snow has two distinct regimes of liquid saturation, the lower range (pendular regime) is one where air is continuous throughout the pore space, and liquid water occurs in isolated inclusions/snow interstices. In the higher range of liquid saturation, which is what we have here (i.e. the funicular regime), liquid water is continuous throughout the pore space and the snowpack becomes completely saturated. To avoid confusion, we have slightly modified the funicular regime description in the revised manuscript as follows:

*Soon after the onset of the second ROS event, the snowpack transitions to a funicular regime (i.e. liquid water occupies continuous pathways through the snow pore spaces - see blue areas between snow grains in the Micro-CT images in Figure 4. This results in downward percolation of liquid water via gravity drainage [Colbeck, 1982a; Denoth, 1999], resulting in a completely saturated snow pack.*

- Secondly, for that considerable change of the snow internal structure during the 2nd ROS one would expect to need also a considerable amount of rain. But this is not the case. According to Figure 2 the precipitation intensity during the second ROS event was much smaller than towards the end and after the 1st ROS event.

True, but the temperatures remained near 0°C and thus more liquid water was also generated from snow melt despite less amount of rainfall deposited. Thus, we believe the change is a reflection of both the liquid water deposited from rain and also from the bulk temperature of the snowpack being at the melting point. The change in snow internal structure is very evident in Figure 4 (i.e. between the 13th and the 14th (or CT scans 2 and 3 shown in Figure 4). It's also evident from the snow temperatures that there was significant melt on the 14th (the entire snowpack, including the snow-ice interface was at 0C).  Also, as already discussed, the profile on the 13th was at the beginning of the rain, so the internal structure on the 14th is also reflecting the first ROS event.

L242-245: What could explain a decrease in backscatter that is larger at nadir than 45 degrees incidence angle? Could it be that a pond or a slush layer developed right below the KuKa-radar because of rain water dripping from antennas and equipment onto the snow?

Possibly and cannot be ruled out. We have added this possibility in the revised manuscript as follows:

*"The steeper decline at nadir from the first ROS event suggests stronger signal attenuation likely from a ponded/slushy snow surface directly in front of the radar, possibly due to rain water dripping from the KuKa antenna horns, though this cannot be confirmed."*

L246-256: As noted by you, quite a bit of what is written in this paragraph is speculative. I am not sure whether the snow property observations and the quality of the remote sensing data you have at hand justify all these detailed speculations.

**We have removed such speculative sentences in the revised manuscript as follows**:

**After the snowpack refreezes, nadir backscatter increases approximately 20 and 25 dB at Ka- and Ku-bands respectively, and remains higher than before the ROS.** *This indicates an electromagnetically rougher air/snow interface due to surface refreezing, resulting in stronger surface scattering from this interface, leading to relatively greater backscatter prior to rainfall.* **While the difference in VV and HH backscatter between nadir and 45 degrees increases slightly (~ 1.5 dB), the angular dependence for Ku-band HV almost vanishes.** *This also indicates dominant surface scattering from the refrozen surface and/or from the refrozen layer ~ 1 mm above the snow/ice interface.* **By the end of 17 September, while Ku-band backscatter angular dependence returns to similar values before it rained, Ka-band shows larger angular dependence at HH and VV, while HV decreases.**

- I don't see "evidence of a percolation channel". I also note that none of the micro-CTs are from the immediate vicinity of the KuKa-Radar, are they?

**The micro-CT on the 15th is from the RS site, and there is general consistency of snow conditions across the floe. Thus the micro-CT scans from all sites are broadly representative of the conditions also at the RS site. The location of the percolation channel is irrelevant, this was just mentioned to illustrate that we can see likely preferential water percolation paths (in which local permeability can be increased), no matter where on the floe we are. Avanzi et al. 2017 show that even in the most homogenous snowpacks, water still percolates in preferential paths. However, we have now marked the percolation channel in the micro-CT figure (Figure 4) as pink areas.**

- You write of "volume scattering from the refrozen surface crust" --> How thick is that crust that you can have volume scattering being dominant over surface scattering?

**We reviewed the analysis and found that the increased backscatter could be due to stronger surface scattering at the rougher air/snow interface caused by surface refreeze, and greater than the backscatter during pre-ROS conditions. We have made appropriate changes in the revised manuscript as follows:**

→ *This indicates an electromagnetically rougher air/snow interface due to surface refreezing, resulting in stronger surface scattering from this interface, leading to a relatively greater backscatter prior to rainfall.*

→ **This also indicates dominant surface scattering from the refrozen surface and/or from the refrozen layer ~ 1 mm above the snow/ice interface.**

- You write of a "glazed surface crust" but I could not see that from your earlier results which point towards rain entering the snow, percolating it, not leaving a hard crust at the surface as would be typical for a freezing rain event with air-temperatures remaining below 0 deg C.

**To avoid confusion, we have removed 'glazed surface crust' in the revised manuscript.**

- Why should pores and channels that were just filled with rain water percolating through the snow (and potentially also some snow melt water) just become air-filled? Would you consider the ice underneath as being that permeable that this water leaves the snow due to gravity drainage? Is this reasonable given the (unknown) ice/snow interface and ice temperatures? Aren't the density measurements for Sep. 15 (Fig. 2) suggesting that densities remain high after refreezing?

**The density measurements show the densities remain high after refreezing, which is expected after the melting/refreezing that results in clusters of grains (Figure 4). Refrozen snow can have densities of 500 kg/m³ and indeed we see this, especially closer to the snow/ice interface. We agree that the pores may not just become air-filled, but Avanzi et al. 2017 show that the formation of preferential flow at early stages of water infiltration leads to spatially heterogeneous coarsening which can lead to locally higher hydraulic conductivity. Local permeability can become bigger if channels develop because of where water preferentially flows.**

- I finally note that the vertical scale of Figure 5 and the way you plotted the observations is not ideal for all the interpretations made. Using thinner lines and avoiding dashed lines when plotting a time series which has gaps anyways are potential solutions for improvement.

**We have zoomed in on the ROS events as recommended below and thus, we feel the new graphic should avoid the confusion the reviewer feels with the older version.**

Figure 5: Your main interest is in the response during the two ROS events. I therefore strongly suggest to focus on these events more by showing days Sep. 11 through 15, i.e. 96 hours of data. That way you would be able to show much better how especially the Ku and Ka-Band radar data changed during the 2nd ROS event.

**We had wanted to show how unusual the event was before and after the ROS ended and did not want to cherry-pick the time-period, but we have now started the observations on September 12 to satisfy the reviewer and they end on the 15th in the updated Figure 5. We keep the atmospheric plot the same though as it is important to understand conditions leading up to the event.**

- What explains the jump in Ku-Band nadir backscatter and 19 GHz TB on Sep. 15 after the data gap?

**The stronger increase in Ku-band nadir backscatter could be due to increased surface scattering from both the air/snow and at the snow/refrozen layer interface. However, this is speculative and our analysis is more focused on relative backscatter changes before, during and after the ROS event. Also, given the previous suggestion to limit the plot to the 15th, we now don't see this jump in the revised Figure 5. That said, the change in the nadir VV and HH**

**signals on the 15th are curious, with also a small change noticed in the 19 GHz channel though we do not fully understand why that is.**

- I suggest to narrow down the TB range for which SBR measurements are shown to something like 150 K to 280 K. I also suggest to use thinner lines. That way it would become clearer how large TBs actually get around the ROS events and how low 89 GHz TBs get after the ROS events.

**We disagree with the reviewer that changing the scale will shed any new light as it is already quite clear how the Tbs jump up, and we also stated the mean Tbs before and when the ROS started. We need the room in the figure to show the legend, and the kernel densities are also quite clear on how the Tbs changed, even for the reduction in the Tbs after the ROS event. Reducing the temporal time-period as recommended above should help the reviewer see the Tb changes better.**

- So far, I found little evidence in the text that used the densities shown on the right hand side of the figure. I suggest to either delete those or, in case you decide to refer more to these, to equip them also with TB and/or sigma0 value axes to be better able to quantify the differences between before and after the ROS events.

**We have expanded our discussion of the kernel densities shown. The Tb and sigma0 values are already on the y-axis.**

Figure 6: The zooms are an excellent idea. Still I vote for reducing the time period shown in panels a) and b) to the same period I suggested to show in Fig. 5 (Sep. 11 through including Sep. 15). This would have the advantage that you don't need to discuss the snow dune issue.

**We have made the change as suggested, though we did feel it was worth showing more data as it sheds light on general variability seen in the data collected and we didn't want to cherry pick to avoid showing things, like the dune that appeared for part of the time. The reality is the changes we saw during the ROS are generally outside the range of variability experienced before and after the event ended.**

- So you indeed have a data gap when the precipitation of the first ROS was strongest. Did it perhaps caused the failure of the instrument?

**No, the rain was not responsible. The RS site lost its power on 12 September. However, KuKa radar has a UPS that led to continuous data collection until 0850 hrs on 13th Nov. Between 0850 hrs and 1300 hrs, the UPS lost its reserve power that led to KuKa not scanning this period when the first ROS event happened. The RS site power came back ON after 1300 hrs.**

- What these data suggest is: if rain increases snow wetness a radar can look deeper into the snow ... is this backed up by theory?

**We have no idea how this is interpreted that way. Snow wetness does not result in the radar looking deeper into the snow and we do not show that either. Perhaps it was because we found a typo of snow/ice interface which should have been snow/air interface.**

- I suggest to use markers at the top and bottom axes (e.g. down- and upward pointing filled triangles) instead of using bars to indicate the locations in the echogram for which you show the profiles on the right. You could connect these markers with thin dotted lines of the same color.

**This has been done as specified.**

- Can you add a 6th profile from, e.g. Sep. 13, 0 UTC from clearly before the onset of the 1st ROS event? I am curious to see whether the strange shape of the black profile compared to all other profiles isn't in fact caused by some beginning failure of the radar. Such a profile could be used much better to illustrate the change in the echograms from before the ROS events to after the ROS events.

**This has been added**

- In addition, simply because there a quite many jumps of the yellow line in the echograms, particularly at Ku-Band, which appear to be caused (according to your writing earlier) by a temporary relocation of the KuKa-radar, I suggest to plot a 7th profile when signals have stabilized again and any relocation effects have ceased, e.g. for Sep. 15, 23:50 UTC or so.

**We have added a 7th profile. Because KuKa was moved again on 15th at 4:53 am, we did not include data from the 15th and have instead added a profile from the 14th to illustrate this.**

- I note that by far not every change in these echograms and the yellow line is understandable. Why, for instance, does the Ka-Band range of maximum relative power decrease (less) than the Ku-Band one at the beginning of the relocation (around 6 UTC on Sep. 15) while there is no change at the end of the relocation (around 21 UTC on Sep. 15) at Ka-Band but Ku-Band jumps back to almost the same range as before? I have to admit that I do not necessarily trust the vertical displacements in the range associated with the maximum relative power right after the 2nd ROS event; more explanation might help here.

**We agree that the yellow line itself is quite confusing and have reconsidered that it is probably not useful on this plot because of the range resolution (1.5 and 2.5 cm in Ka and Ku, respectively.) We have instead added a profile as above, which we feel is more useful in understanding what is going on.**

- For me, looking at Fig. 6 as it is, the main conclusion is that the location of the maximum relative power gets a bit closer to the radar at Ku-Band (slightly lower range) but bit farther away from the radar at Ka-Band (slightly larger range). Given the fact that Ka-Band observes at the smaller wavelength of the two I don't understand this immediately as this would mean that after the ROS the Ka-Band penetrates deeper into the snow-ice system than the Ku-Band.

There are of course multiple effects which would determine the interaction of the radiation with the snow and ice, including roughness (at the scale relevant for each wavelength), penetration of the radiation etc. We have removed the yellow line indicating the peak as we do not think, as the reviewer notes, that analysis at these tiny scales << the range resolution are necessarily helpful and could be misleading.

L271/272: You are talking about a vertical shift of between 0.5 and 1 cm? How sure are you that this is not just some kind of noise? I note that the range bin resolution is about 0.5 and 0.8 cm for Ka- and Ku-Band, respectively.

Yes that is correct, and we have changed the manuscript to explain that this is not something that we are able to determine as we cannot control for any sled settling. We hope that it is now clear that although we see a small shift we cannot attribute it to either a change in the scattering horizon relative to the snow pack, or settling.

L276/277: Moving the instrument should impact both frequencies, right? But we only see a shift in the Ku-Band. Why?

The movement would determine which frequency was more impacted e.g. if one side of the sled was depressed relative to the other, or if it was pivoted about a particular point etc. - ie depending on the movement it could impact the Ku- and Ka-band signals differently.

L312: "because volume scattering in the snow is larger at 89GHz" --> Are you sure this is the reason? What is the ice type below the just 7 cm thick snow layer which is comparably cold and potentially more or less transparent at both frequencies. Wouldn't it be more likely that the observed difference in the observed TBs results from the emissivity of the underlying ice? Please check with earlier experiments and provide 1-2 references for your (revised?) statement then.

The emitting layer depth at 89 GHz is within the snow layer even when the snow is only 7 cm thick. For example, sea ice concentration algorithms exclusively using 89 GHz channels do not distinguish between different ice types because the signature of sea ice is insensitive to ice type. Instead, the 89 GHz sea ice emissivity is affected by processes in the snow such as the ROS event. Also, the changes in scattering is happening over two days and while the snow cover metamorphosis can respond on this timescale, the underlying ice and its structure responds on longer timescales than a single ROS event.

- Please put the observed value (L313) into context of existing knowledge.

Observed values (and absolute calibration) of the SBR is not the important part of this study. It is the relative change that is important here. Many studies report values for satellite observations which we cannot directly compare to the SBR because of spatial mismatch. A study by Harasyn et al. 2020 shows Tbs over FYI in Hudson Bay showed Tbs at 19 GHz varied between 220 and 270K and at 89 GHz between 185 and 270K. Surfaces sampled included dry snowpacks and wet snowpacks. Thus, our recorded values are in range of those observed over

**FYI for different ice conditions. We struggled to find similar in situ values for second year sea ice. If the reviewer knows of a study over second year ice we can refer to, we would be happy to include it in our discussion and reference list. In the meantime we can list the first year ice study of Harasyn et al. 2020 and how our results compare to those.**

**Harasyn et al. also compared AMSR2 Tbs to the in situ measured ones, and state that comparing in situ Tb with the satellite Tbs is difficult due to the large difference in spatial scale. Since melt ponds were prevalent we cannot directly compare our results with the AMSR2 data extracted for the MOSAiC floe, and this is likely also a problem with any other comparison from other satellite-based studies.**

L315: I agree that the emissivity of a wet snow pack is nearly 1 ... but how near is nearly? According to Figure 2 the snow surface temperature was 0degC, i.e. say 273K. In order to observe a TB of 274 K requires the emissivity to be larger than 1, which appears not to be realistic unless you proof differently. Even for an emissivity as high as 0.99 the observed TB would be 270.4K. Not knowing how accurate and reliable the calibration of the SBR actually is, I suggest to not put too much emphasis on the discussion of these perhaps artificially high TB values.

**We agree that the emissivity is not larger than one and without going into a discussion about reflected atmospheric emission we have reformulated the sentence so that this is clear. Thus we change:**

**L315: change "The increase to 274 K reflects the fact that emissivity of a wet snowpack is nearly 1 because of high absorption, and the physical snowpack temperature at the top is slightly above 0°C."**

**to**

**"*The increase to 274 K gives an indication of the absolute calibration to the radiometer (it should not increase above 273.15K) and it also reflects the fact that emissivity of a wet snowpack is nearly 1 because of high absorption, and the physical snowpack temperature at the top is close to 0°C.*"**

Figure 8: I like the delay in the pulse peakiness between CS2 and KuKa-radar data with CS2 showing an earlier increase; this seems to be in line with the direction from which the cyclone responsible for the ROS event was moving into the region of interest - aka from the South / Southeast; hence it arrived earlier at the location of the CS2 overpasses than the MOSAiC floe.

I recommend, though, to check the KuKa radar sigma0 values prior to the ROS events in Fig. 8 because Fig. 5 shows values around 0 dB here, hence a discrepancy between the values shown in both figures.

**Thank you for spotting this, it has been corrected. The incorrect data have been replaced with the Figure 5 values which show the same pattern but are a little offset.**

L316/317: How thick a wet snow cover needs to be to mask the emission of the underlying sea ice? Is the observation that both frequencies show a similarly high TB enough evidence that the ENTIRE snowpack is wet?

**The penetration in wet snow depends on the water content. On the 13-14 the entire snow-pack is wet and both the 19 and 89 GHz TB's are equal in magnitude. The emission is originating from the top of the wet snow-pack at both frequencies. When the air temperature drops on the 14th the decreasing TBs have a different slope at 19 and 89 GHz, 1) these two channels are affected differently by scattering in the snow pack and 2) they have different penetration. We added a clarification to ease the reviewer's concern and now added (in red):** *This also suggests that at the time the data were collected, the entire snowpack is wet since the 19 and 89 GHz channels are equally impacted, and thus the emission is originating from the top of the wet snowpack at both frequencies.*

L318: "TBs drop to cold conditions again" --> what you write in the following lines misses the observations that the TBs drop to values considerably lower than before the ROS event. Please quantify this change and also quantify the changes in the PD before and after the ROS events.

**The reason for this drop is scattering in the snow-pack (from metamorphosed snow structure). We already state in L323: "***Further, grain size increased throughout the snowpack, and thus Tbs are lower than before it rained (more volume scattering), affecting both 19 and 89 GHz.***" As for a quantification of the change, since we are limiting the analysis now to a shorter range in updated Figure 5 all numbers would have to change if we wanted to be consistent with the figure. The problem is that the variability is large immediately after the ROS event since the Tbs are still decreasing for some time after the snowpack starts to refreeze entirely. So if we use the updated Figure 5 time-range you would get these numbers:**

**Prior to ROS event: Tb19H=240.0+/-3.05, Tb19V=254.4 +/-1.06, PD_19=14.4 +/-2.48**
                                 **Tb89H=193.1+/-3.03, Tb89V=202.6+/-2.41, PD_89=9.4+/-1.33**

**After ROS event: Tb19H=225.0+/-11.83, Tb19V=250.0+/-8.32, PD_19=25.0+/-5.83**
                             **Tb89H=178.1+/-12.09, Tb89V=184.7+/-12.03, PD_89=6.5+/-1.38**

**However, the real question is what are the changes in Tbs after the Tbs have stabilized. It takes a while for the entire snowpack to lose the liquid water, and thus ideally we should pick when exactly that happens. We could select that based on when the Tbs stabilize again, which happens sometime on the 16th. So we could either report the values after the ROS event ended and for the entire duration of the collected data, which is until September 18 at 04:30 UTC or we could report the data starting on the 16th until the end of the time-period. Using either after ROS time-period we would find the following metrics after the ROS event has ended, showing both a reduction in the Tbs and an increase in the PD at 19GHz .**

**after ROS through end of record:**

**Tb19H=220.2+/-13.25, Tb19V=243.3+/-13.58, PD_19=23.1+/-6.13**
**Tb89H=178.8+/-11.51,Tb89V=185.2+/-11.17, PD_89=6.4+/-1.57**

**Period between September 16 and 18**

**Tb19H=206.3+/-4.19, Tb19V=223.7+/-2.29, PD_19=17.46+/-2.42**
**Tb89H=180.8+/-9.39, Tb89V=186.8+/-8.02, PD+89=6.1+/-1.97**

**We decided to report the values after the entire snowpack has refrozen. Thus we now state:**

*"In particular, the 19 GHz polarization difference (PD) (i.e. PD = $Tb_{19V}$ - $Tb_{19H}$) is larger than before the ROS event, increasing from 14 to 17 GHz after the entire snowpack is refrozen (e.g. between September 16 and 18). This increase is likely the result of ice layers in the snowpack (e.g. high density layers shown in Figure 4(e)); the 89 GHz PD on the other hand decreases from 9.4 to 6.1 GHz. Further, grain size increased throughout the snowpack, and thus Tbs are lower than before it rained (more volume scattering), affecting both 19 and 89 GHz, which decreased to 206±4.2/224±2.3 (19H/V) and 181±9.4/187±8.02 (89H/V) GHz. The larger standard deviation at 89 GHz is likely because this frequency is particularly affected by snow grain/structure scattering."*

- What explains the super low 89GHz TB values of almost as low as 160K at H-polarization? Has there been evidence for such low values before in the published literature?

**We now list the mean values after the ROS event, giving values between Sep 16-18 when the entire snowpack has dried out. However, at 89 GHz the response is faster since it does not penetrate as far as 19 GHz. Thus, the values are around 180K not, 160K at H-polarization. Scattering magnitude is a function of frequency and the 89 GHz channel is particularly affected by snow grain/structure scattering. We now add: "*The larger standard deviation at 89 GHz is likely because this frequency is particularly affected by snow grain/structure scattering.*".  Such low values are not completely unusual as shown in the Carsey et al. Chapter 4 from AGU monograph**

- What explains the fluctuations in 89 GHz TB after the ROS? Is this real or an artifact of the SBR?

**The observed TB variations are real. The measured TB is a function of snow temperature (TB=emissivity\*emitting layer temperature), and also reflected atmospheric downwelling radiation (affecting 89 GHz much more than 19GHz, high reflectivity and sensitivity to atmosphere). In addition, temperature could affect penetration if the snow is not completely dry or saline and therefore the depth of the scattering layer.  We see the variations in the snow surface temperature in Figure 2. Thus, we added the sentence:**

*"Further, the 89 GHz channel is also impacted by atmospheric downwelling radiation, and small temperature fluctuations at the snow surface will influence penetration, leading to temporal fluctuations in Tbs.'*

L322: "grain size increased throughout the snowpack" --> one could speculate how much these 4 cm of apparently icy snow (see RS profile in Fig. 4e) still resemble snow and how one could distinguish between ice layers (see L321) such a snow pack / refrozen slush cover can contain.

**Figure 4 clearly shows that the grain size increased throughout the snowpack. Arguing that it is no longer snow because now ice layers have formed does not change the fact that the snowpack has been altered because of the ROS and subsequent refreezing. We do not feel a change is needed here.**

Figure 9: Please provide information in the text how you co-located these AMSR2 data with the MOSAiC floe location (see further below).

**We simply chose the AMSR2 gridcell that corresponded to the lat/lon of the MOSAiC ice floe on September 9 and followed the floe through time until the 18th. This is now stated in the Figure caption. However, we have now updated this plot to use the swath data (see below) and the distance of the MOSAiC floe to the AMSR2 pixel is now included in Table A3.**

- I suggest to omit the 23 and 36 GHz data and instead, similarly to the comparison between KuKa-radar and CS-2 (Fig. 8), plot the daily average TB-values. This would make the comparison more consistent. Please check, like you did for CS-2, the respective AMSR2 overpass times to figure out how you could optimally compare the SBR observations with the AMSR2 data.

**We disagree, the data are informative as is, showing the impacts across all frequencies which is relevant for sea ice retrievals from sensors such as AMSR2 or SSM/I. As for the data we show from AMSR2, we previously showed daily averaged gridded data from the ascending pass. We have now updated to use only the swath data and chose the location closest to the ice floe. We don't feel that we need to optimally compare the SBR and the AMSR2 data since the point is to show the event could also be seen at the satellite level.**

- Why are no data shown for Sep. 15/16? I assume that this is because of the observation gap around the North Pole. If this is the case then I recommend to, instead of using TB of a single 12.5 km grid cell (which I assume has been done), computing the mean TB of a slightly larger area. Why not using the same radius as you used for CS-2?

**The reason is that we had *actual data* that did correspond to the floe location (in contrast to the CS2 data), and we chose to use that data instead of a larger area. The point is about showing the event was significant enough to be observed in the satellite observations which Figure 9 highlights. We could make it a larger region but there is no strong scientific reason to do so.**

- While the PD at 89GHz before the ROS events is kind of in line with the SBR observations, the PD at 18.7 GHz is with 25-30 K about 10 K larger than the one observed by the SBR. Why? I also note that SBR 19 GHz TBs are higher than AMSR2 TBs while SBR 89GHz TBs are lower. Why?

**We explain why in the text: L320: "In particular, the 19 GHz polarization difference (PD) (i.e. PD = Tb19V - Tb19H) is larger than before the ROS event potentially as a result of ice layers in the snowpack; the 89 GHz PD on the other hand decreases."**

**AMSR/SBR: the AMSR-2 19 GHz footprint is ~22 x 14km while the SBR is measuring a point. The difference in scale explains the differences in TB. Further, atmospheric effects will impact the 89 GHz channel. However, as before the absolute magnitudes are not as important here as simply the fact that the impact is observed both in situ and at the satellite level.**

L336/337: I have difficulties to find this statement in the cited paper. Therein it is clearly described that there are so-called track point differences between Ku- and Ka-Band radar altimetry and that penetration into and attenuation / scattering within the snow is a function of snow depth and grain size but there is not this clear statement made. I suggest to use a different reference if you want the keep the statement as written. Also, for Ku-Band it has been shown in published literature that the main scattering horizon could be located anywhere from within the upper centimetres of the sea ice through the entire snow pack up to at the snow surface itself.

**In the Tonboe-TC paper, it is stated in the intro: "Several studies suggest that it may be possible to derive snow depth directly using a dual-frequency approach by combining Ka- and Ku-band radar altimetry (e.g., Lawrence et al., 2018; Guerreiro et al., 2016). The underlying principal behind this technique is that the assumption of predominant Ka-band scattering originates at the air–snow interface, while for Ku-band, the dominant scattering originates at the snow–sea-ice interface (Beaven et al., 1995; Lawrence et al., 2018; Laxon et al., 2013; Kurtz et al., 2014)." If you do not feel this reference is ok to use, then we could substitute the Tonboe et al. reference with the Lawrence et al., 2018.**

**Also, the main scattering horizon is not a real scattering horizon, it is the projection of the track point into the snow and ice profile. The actual scattering is happening at the snow surface, at the snow ice interface and potentially from internal layers.**

L339-342: What is written here regarding the change of the elevation of the main scattering horizon appears to be quite speculative. What is the vertical resolution of the two radars when looking nadir? How precise could the altitude of the antennas above the snow surface be tracked? You state yourself that the sled might have sunk into the snow a bit (millimetres? centimetres?). In addition, please see my comment with respect to the echograms when viewed from "a greater distance", suggesting that Ku- and Ka-Band main scattering horizons actually changed differently from before to after the ROS. I suggest some reconsideration of the results and depending on that some clarification of the writing here, taking into account the involved uncertainties and limitations more rigorously.

Perhaps the text can be sharpened here, but as we understand from our data, the snow surface has a clear return in Ku-band and sometimes the snow ice interface. The radar can be referenced to the snow surface and then the trackpoint is somewhere between the snow surface and the snow/ice interface. In our revised text, we have reduced the analysis on the change of dominant scattering surface. As the reviewer says, there are issues relating to resolution, sinking etc. whose contributions are not possible to fully understand and we have therefore reworded our analysis to focus more on the change in waveform shape. Thus now this section reads as:

*"Our waveform analysis suggests the peak associated with the dominant scattering surface moved upwards (to smaller range) after the second ROS event and refreezing. The shift is 0.9 cm (Ka-band) and 1.5 cm (Ku-band). This could be due to rainwater refreezing at the surface, raising its elevation, and/or decreasing roughness at nadir. Another contribution could be settling of the radar; it is not possible to verify whether the sled sank into the melting snow and therefore we cannot quantify whether this partially or fully accounted for the apparent change in range.*

*Instead we focus on changes in the waveform shape. While satellite radar altimeters cannot resolve the peaks visible in the KuKa data, and there are important differences in range resolution and the echo shapes between KuKa and satellite radar altimeters, the in situ shape changes at both Ku- and Ka-band demonstrate that satellite-retrieved freeboards from both CryoSat-2 and AltiKa could be shifted upwards. However, satellite radar return power depends strongly on the large-scale floe topographic variability (i.e. ridges, rubble fields), which controls the total number of illuminated point scatterers as a function of delay time. While there are challenges in upscaling to satellite footprints, KuKa data combined with measurements of physical snow characteristics allow us to investigate how the combination of snow depth, temperature, salinity and radar-scale roughness control sigma0, and how these facet-scale factors affect the shape return pulse, as a function of transmitted pulse bandwidth. These insights can then be combined with numerical radar simulation approaches that focus on large-scale floe topographic variability [e.g. Landy et al., 2020], with realistic gain patterns, to understand how the different factors combine to influence the radar return power over km scale satellite footprints."*

L364: "reducing the elevation of the air/snow ..." --> Which effect on the elevation of the air/snow interface relative to a radar sensor is more important: the compaction of the formerly dry snow becoming wet or even slushy hence much decreased depth, or the weighing effect you describe?

That is a difficult question to answer and one we cannot answer with the measurements we made. We are simply stating that additional SWE could lead to weighing down of the floe, reducing the ice freeboard (and hence also the snow freeboard). Of course the reviewer is right that compaction of the formerly dry snow becoming wet will also reduce the snow freeboard, so we have now added the following sentence:

*"In addition, compaction of formerly dry snow by rainwater and wet snow metamorphism, as implied in Fig. 4 (between 13th and 14th of September) also reduces the snow freeboard".*

L365/366: How did you compute that an increase of the existing SWE by 11.5 mm would cause the explicitly stated change in elevation of 13.6 mm? This is not clear to me. Also, when you say "reduction of the air/snow interface elevation" then you take the ice/snow interface as the reference? Hence, in other words, snow depth decreases by 13.6 mm?

**Archimedes' principle dictates that the weight of accumulated precipitation must be supported by the weight of seawater displaced. Taking seawater density as 1023kg/m3 and freshwater density as 1000kg/m3, the snow/ice interface would be displaced downward by: SWE \* (1000/1023). In the case of 11.5 mm of SWE accumulation, this would be 11.24 mm (so a similar amount)**

L370/371: This is the only place where I find the observations of (very slightly) elevated basal snow layer salinities after the ROS very interesting because it seems to point towards some brine wicking from the underlying ice due to the refreezing process. Alone, the ice underneath is MYI and should be fresh in its upper some centimeters and you state that your values fall within the measurement uncertainty anyways. But, what if the sea ice underneath would be saline FYI and you would have had some brine in the basal snow layer that was flushed downwards by the rain ... would the refreezing kind of suck this brine back upwards again and perhaps at an even larger concentration? I don't find your argumentation here overly conclusive.

**There are two hypothetical possibilities here.**

**1) if ROS events occur on impermeable FYI in winter, ROS events can flush brine within the snowpack to the snow/ice interface (but not rejecting brine through the impermeable FYI volume), potentially leading to a temporary meltwater layer at the snow/ice interface which is then followed by refreezing (depends on snow depth and magnitude and duration of the rain event).**

**2) If ROS events occur after melt/ponding/drainage/snow melt on FYI, then the brine from the snowpack would have already undergone brine rejection through the permeable ice volume through the interconnected brine channels. However, we suspect this is less likely on older sea ice types such as SYI or MYI.**

**In our case, pre-ROS conditions indicate a fresher snow pack. So we cannot confirm a saline FYI as the source of brine that refroze causing the snow salinity to show measurable amounts of brine that could also fall within the instrument accuracy of the salinometer. Also, the presence of snow salinity before and after ROS events would have shown amplified backscatter changes from the snow volume/interfaces which we do not see here.**

L378-393: "For satellite-based ... can this lead to permanent geophysical and ... microstructure [Colbeck, 1982]." --> I suggest to switch the information here: ROS events during winter can create certain geophysical changes at various scales. These can then impact active microwave measurements in various ways.

**Ok, we changed accordingly. Specifically we now state: "*Overall, ROS and subsequent refreezing can create geophysical and thermodynamic changes to the snowpack at various spatial scales, leading to complexly-layered snowpacks, with changes in salinity- and temperature-dependent brine volume (for FYI) [Nandan et al., 2020], density [Denoth, 1999] and snow grain microstructure [Colbeck, 1982a], all altering surface and volume scattering contributions to the total backscatter. These modifications will in turn affect the reliability of SAR and scatterometer algorithms to accurately retrieve snow/sea ice geophysical properties, classify FYI and MYI types [Nghiem et al., 1995] and timing of melt onset. Our results highlight that modifications during and following ROS events and its effect on microwave scattering at these frequencies is not trivial. Overall, our results provide a detailed and first-hand understanding of the geophysical impacts of ROS and its effect on the radar scattering behavior. These findings will help us to further aid interpretation of radar backscatter changes due to ROS events at satellite-scales.***"

- I find the two paragraphs that follow very speculative and not necessarily backed up well by your results. For instance, you refer to ROS as an event that might happen in spring and erroneously interpreted as melt onset ... but your example is from September. I am wondering whether you could delete this 5.2 section completely without missing too much information.

**We don't share the reviewer's view that speculating how ROS could impact melt onset retrievals is not worth considering. Yes our event is in September, but that does not in any way imply that a similar event if it occurred in April would not have an impact on the snow regime that would in turn bias melt onset dates. This section is about implications, so taking a broader view than just focusing on the event itself is what we're discussing. We believe it is important to think about the broad implications. Future studies can focus on finding a winter or spring ROS event from reanalysis data (since in situ data are generally lacking), and then investigate how the satellite systems respond.**

Figure 10: The information given here is partly equivalent to Fig. 8 but the way computations and perhaps also co-locations are done seem to be different. I suggest to either combine Fig. 8 and 10 or, if you keep Fig. 8 then I suggest to use the same co-location and computation procedures for both figures with respect to AMSR2 TBs and SBR TBs. In any case you should describe how the co-location between AMSR2 and SBR data was done.

**We agree with the reviewer that these should have been done the same way. Originally gridded (Figure 9) vs. swath data (Figure 10) were used, in part because different authors of the paper produced these figures. We now update Figure 9 to only use swath data. The ship's GPS data is used to find the location of the floe relative to the AMSR2 swath data.**

L403-408: In order to understand that the PD decreases when both V and H-pol emissivities are close to 1, one needs to state the emissivities differed from 1 by an amount that differs between the polarizations for dry snow.

**Sure we can state that the emissivity of dry snow is less than 1. This is rather self-evident from the data, but we now add a reference to Eppler et al.. i.e. we revised the sentence to read:** "*Wet snow has an emissivity close to 1 at both V and H polarization (compared to emissivity less than 1 for dry snow [e.g. Eppler et al., 1992]), and thus the PD decreases during ROS*"

- Obviously, because ASI SIC is above 100% anyways (almost all the time) before and after the ROS for both AMSR2 and SBR, you are in the saturation regime of that algorithm, i.e. the sensitivity of the PD to changes in SIC is comparably small and changes non-linearly. Is is correct to assume, that because of this non-linearity that occurs here in contrast to the linear change of PD with SIC at lower SIC values you have that larger increase in SIC due to the ROS event from 82% to 90%? I suggest that you state more clearly the reason for these different numbers.

**For computing the ASI SIC, we avoided saturation of the algorithm by modifying the retrieval curve for polarization differences <11.7 (tiepoint for 100% SIC). For PD <11.7 we did not use the polynomial algorithm but a linear dependence calculated from the slope and intercept of ASI between PD = 14 and PD = 11. With this modification, we avoid saturation and can assume that the changes observed here can be transferred to a SIC range between 80% and 90%. To address this we now add:**

*"In computing the ASI SIC during this particular ROS event, we modified the retrieval curve for PD differences < 11.7 (tiepoint for 100% SIC), since due to its polynomial dependency, the ASI algorithm saturates at SIC=100%. To deal with saturation, we compute the SIC using the slope and intercept of the ASI retrieval equation between PD = 14 and PD = 11 (i.e., the SIC = 80% to SIC = 100% region) and then apply a linear fit for PD values < 11.7. With this modification, the SIC changes observed here are also representative for areas with SIC between 80% and 100%."*

L409-413: I suggest to see this in a more differentiated way. Before the ROS NT SIC is at 90%. After the first, much more intense ROS event the NT SIC is close to 100%. It is just after the second (weak) ROS event that the NT SIC drops to 70%. What causes the upswing to near 100%?

**The NT is dominated by the 19 GHz pol-ratio and when the SIC is increasing the pol-ratio is decreasing. During the peak of melt the pol-ratio is decreasing (see fig 5) and after the event crusts/layering in the snow is giving a high pol-ratio and artificially low SIC estimates.**

- I note that you show GR3719V but not PD19. Why?

**While ASI and the snow depth retrievals can somewhat directly be linked to GR and PD89, the NT retrieval does depend on both, GR and normalized PD in a complex way and thus showing PD19 gives only limited help for understanding the behavior of the NT retrieval**

- You state that the response lasts long after refreeze ... but if one compares SIC values, then one has NT SIC of around 80% on Sep. and then again on Sep. 16. So while I agree that certainly there is some longer lasting change in microwave signature I am not so sure whether the SIC is a good indicator here. In order to understand this better a graph showing PD19 could help. What I do note is

the decrease of the GR3719V to even more negative values is very clear which would also mean an increase in the MYI concentration when retrieved with the NT approach.

**The GR3719v is an indicator of scattering magnitude and scattering is higher in MYI than in FYI and that is why it can be used to classify these two ice types. Scattering can also happen in the snow layer as after the melt-event. We now add this statement:** *"Furthermore, the decrease of GR(37/19) to more negative values after the ROS event would result in an increase in the MYI fraction".*

- Did you recognize that GR3719V values approach 0 during the ROS, hence making the MYI to look like 100% FYI?

**That is an interesting comment and we had not specifically looked at the impacts on ice type. However, we agree this should also be mentioned so we now state: "***The increase of the GR(37/19) to 0, would additionally lead to the second year ice floe mapped as 100% FYI using the NASA Team algorithm.***"**

L414: "multiyear ice [Rostosky et al., 2018]" --> this reads as if this paper only deals with snow depth retrieval over MYI which is certainly not the case. You might want to rephrase your statement.

**Rephrased to read:** *Snow depth retrievals using algorithms to derive snow depth over FYI [Markus and Cavalieri, 1998; Comiso et al., 2003] and over both FYI and MYI [Rostosky et al., 2018] are also impacted.*

- On another note: you are standing on an ice floe in mid September ... hence the ice certainly is not FYI but it is at least second-year ice. Therefore, I recommend to reformulate your statements in this paragraph accordingly - aka: "If we assume that the ice floe is FYI then we could retrieve snow depth using the approach of ...." The fact that you retrieve a snow depth which is much too high: 20-30 cm instead of 7 cm measured before, or 4 cm measured after the ROS supports this notion very well. That way you would also demonstrate to the reader that these considerations are purely hypothetical.

**We used the approach of Rostosky et al. to compute the snow depth. Thus, it is valid for MYI. We show both the snow depths using GR19/7 or GR19/37, with the GR19/7 working for MYI. So we rephrase now "***If the Markus and Cavalieri [1998] snow depth retrieval method is applied (valid only for FYI), this would lead to a retrieved snow depth of 0 cm (using GR19/37), or a reduction of 50% if using Rostosky et al. [2018] (using GR7/19), respectively (Figure 10(d)).***"**

L423/424: In the first moment I would think similarly. However, there are (at least) two things fundamentally different here: 1) The snow layer on the ice is quite thin in your case but would be much thicker in a normal winter case. Therefore the water entering the snow would have a different effect.

**First-off you would have to argue evidence for much thicker snowpacks. Yes there can be areas of thin or thick snowpacks. Here is an example from leg 2 and you can see clearly that while the**

**mean snow depth is less than 20cm, there are tails of deeper snow packs, but also regions with thinner snow. Thus, the snow depths for this particular event can be the same as in the middle of winter.**

[Figure]

**So what is important then is not the snow depth itself, but rather the underlying temperature of the snow/ice interface. And what would happen to the water entering the snow, which would likely refreeze faster and shallower for a cold snowpack than a warm one. We clearly see even with our shallow snowpack evidence of ice layers forming within the snow after the ROS event and refreezing. Below is also a figure that shows the snow/SSL depth immediately after the ROS event, on September 14th. The red and pink lines are FYI, and the ROV area is close to a ridge and is MYI with many ponds in the transect.**

[Figure]

One thing that is not sufficiently discussed here is the possibility that the underlying sea ice was still quite warm and potentially contributed to the observed change in the vertical snow structure. What were ice-snow interface temperatures?

**See responses above and modifications to the manuscript, i.e. "***The snow-ice interface temperature was slightly negative (-0.5°C) on the 13th, 0°C on the 14th of September and negative on the 12th and the 15th.***"**

2) The bulk and initial snow surface temperatures would be substantially lower and it is reasonable to expect that a winter-time ROS event would provide freezing rain and therefore a different microphysical environment and hence microwave signature of the snow. I therefore suggest that you rephrase this statement.

**It is reasonable to expect the water would refreeze in a more shallow layer in a colder snowpack, but it would still percolate a ways down. The penetration depth due to inhomogeneous infiltration is about 5-10 times deeper than estimated by piston flow (homogeneous infiltration), likely even in cold snow.**

Figure 11: Is the area for which this trend is computed in panel a) identical to the region shown in panel b)? You write that this is "over sea ice", while panel b) shows all areas, i.e. land, ice-free ocean and sea ice.

**This is computed for the central Arctic Ocean (panel a), which includes the Arctic Ocean regions, excluding Baffin and Hudson bays, East Greenland Sea, the Sea of Okhotsk and the Bering Sea. This information is added to the legend.**

- What is the unit of the slope in panel a)?

**The y-axis already shows the results in mm/day and the x-axis is yearly, so the slope is thus mm/day/yr**

- I note that here you sub-sum also "wet snow" under wet precipitation in contrast to Figure A3 where "ice pellets and snow" do not contribute. Which one is correct? Did you apply the same selection criterion for both figures?

**It was obvious not to include ice pellets and snow under the same umbrella as rainfall, and thus these are excluded as before.**

- The map in panel b) appears to show the trend in mean wet precipitation for the entire period, i.e. the total change and not the change per year. Is this correct? Or do I have to read the map such that there are vast regions where the winter-time wet precipitation increased by 40 mm/day within the 42-year period?

**Thanks for pointing this out, we found that the panel b) graphic had the wrong units since the precipitation and snowfall are in m and a mistake was made in the conversion to mm. We have replaced with the correct plot. The spatial pattern is the same with the correct units now.**

- I note that there are vast regions where the trend is not covered adequately by the legend and suggest to change this.

**This is not a problem now with the revised figure**

- I as well suggest to indicate the sea ice extent.

**That is not practical as the sea ice extent changes dramatically over those months shown**

- While I can guess that the hatched area has something to do with significance you did not mention this in the caption.

**Yes it's the region with statistically significant trends at the 95% confidence interval.**

- The title of panel b) is a bit misleading because you are not only referring to rainfall but you are also referring to wet snow (which can have a completely different effect on an existing snow cover - as well as freezing rain) and the mixed rain / snow events. To what extent this "mean wet precipitation" therefore matches the conditions you experienced during MOSAiC in September remains unclear.

**During the MOSAiC ROS event in September we had a combination of wet precipitation, with sleet and**

- Finally, while the MOSAiC observations, for which you apparently had a good agreement between observed and modeled precipitation type (otherwise you would not dare to carry out such a trend analysis) are from September, these results here are for winter, specifically for winter conditions which in terms of precipitation phase might be more of a challenge for the reanalysis. Hence my question: What is your idea of how credible these results are?

**It was good to see that ERA5 did well capturing the observed event in September but that was to be expected since ERA5 assimilated weather station and balloon data (relative humidity, temperature and pressure from MOSAiC). For how well we think ERA5 does in a broader sense, we already pointed out two other publications that evaluated ERA5 against north pole drifting station data (Barrett et al., 2020) and also other station data around the Arctic (McCrystall et al., 2021). In both these comparisons, which we urge the reviewer to read, we found ERA5 performed better than other reanalysis in terms of precipitation. ERA5 is thus the best data we have for trying to interpret these changes as we do not have station data everywhere in the Arctic Ocean to assess this in situ. Given our work in evaluating ERA5 precipitation, we believe the data are useful for assessing how ROS may be changing over time.**

- Provokative question at the end: How about during legs 1 to 4: How often did you have ROS events?

**We haven't looked at the full annual cycle of precipitation measurements, but we did have two moist air intrusions in mid-April 2020, but we did not have KuKa operational at that time and thus we chose an event when we had both active and passive microwave observations. Note discussion of that event on passive microwave brightness temperatures has been submitted to the Frontiers special issue (Rückert et al. ).**

Editoral comments / typos:

L55/56: Please provide the full name of all sensors upon their first mentioning in the text. This applies also to all sensors mentioned later such as AMSR2 (L59, SSM/I and others.

**Done**

L77: "VV, HH, ..."

**Done**

I guess you need to explain once what is meant by V, H, VV, HH, and the HV, VH - also in view of the next subsection.

**Done**

L77: Here you name the instrument "KuKa radar". I recommend that you use this name throughout the paper; the usage of other acroynms further down ("KuKa system", "KuKa instrument" or just "KuKa") is confusing and does not read overly well.

**We now refer to it as KuKa radar everywhere**

Table 2: Please check whether row "Receive Noise" should read "receiver noise" and whether the unit is really K and not mK.

**Should be receiver noise. Unit is correct as given in K**

- Since you provide the center frequency with 1/10 GHz precision you might want to do that for the bandpass as well.

**Ok, changed to just be 19, 37 and 89 GHz.**

L113: Typo: SSMI --> SSM/I

**Corrected.**

L124-126: Please check these two sentences ... "Physical temperatures were also made of the absorber pads ..." reads strange as does the next sentence.

**We removed the word "also". Otherwise this is a correct statement.**

L182: Please check this sentence. "vertical temperatures above zero" reads strange. See also L186 please.

**We meant temperatures as a function of height in the atmosphere, rather than those at the surface. We have rewritten as: "*Observations during the largest rain rates indicate air temperatures above zero extending to slightly above one kilometer above the surface.*"**

Figure 2 caption: "Symbols denote ..." --> Perhaps better: "Different symbols denote different snowpits in panels d) and e).

**Done**

- I find it a bit unfortunate to have different colors representing different parameters AND different methods in panel d). Perhaps you could move SSA to panel e) should you decide that there is no need to show the snow salinity.

**We removed the snow salinity but we kept the SSA with the density as they are informative as viewed together.**

L200: I would not consider 8:30 UTC "shortly" after it had started to rain (5 UTC, see L179).

**Removed the word shortly**

L224: "were generally similar across the floe" ... I am not sure the reader gets this information from the figures just discussed. Would it make sense to summarize the key changes in the snow properties caused by this ROS event in 2 sentences?

**You can see that pretty clearly from the snow pit observations and photographs of the area. We chose to detail the evolution of the snow properties in relation to the ROS event, which can be found at the beginning of section 3**

Figure 3: Please explain the devices that are seen in the photographs.

**We added a description of the snow grain size cards and the reference targets shown in the figure to the figure caption.**

L263: "samples" reads strange. Please see my comment to Figure 6.

**We have changed to "waveforms".**

L279: "the peakiness" --> Just for my understanding ... usually one speaks of "pulse peakiness", here you use "peakiness" or "waveform peakiness". Are these terms that can be used interchangably?

**We have specified waveform peakiness to avoid confusion. We have checked Tilling et al. (2018) where this is discussed in detail for CS2, and there 'peakiness' and 'pulse peakiness' are used interchangeably. We feel that waveform peakiness is probably the clearest way to specify in this paper, because KuKa is a FMCW system (not pulsed) and we are looking at the peakiness of the CS2 and KuKa waveforms.**

L281/282: "in the Arctic" --> perhaps better "in the region shown" because the region shown in the maps of Fig. 7 appears to be a rather small sub-set of the Arctic.

**Changed as recommended.**

L290: Typo: "included" --> "include"

**Changed**

L296: Please check my comments to Fig. 6 in this context.

Figure 7: I suggest to delete the images at the beginning and end (Sep. 9 and 18) and use the space created to illustrate how the area shown maps with respect to land and sea-ice distribution for Sep. 13/14. Information about latitudes and longitudes (i.e. numbers) would help as well to geolocate the region shown.

**We have instead added the lat/lon boundaries. We thought it was quite clear the region the tracks correspond to are over sea ice areas only.**

L311: "less" --> here perhaps better "lower"

**Changed to lower**

L314: Please note my comment regarding the SBR data shown in Figure 5.

L321: "potentially ... snowpack" --> would it make sense to refer to a) the increase in overall snow density and to b) the observed vertical density gradient (see Fig. 4 e), RS)?

**Done**

L325: Figure 7 needs to read Figure 9

**Corrected**

L326: "temporal averaging ..." --> This sentence reads as if AMSR2 only provides this kind of data (daily averaged) ... I suggest to be more correct here and state that the data product you used comprises daily averaged TB observations of ascending orbits and that even though you used this product instead of single swaths you are able to detect the ROS impact on the TBs.

**We originally used daily averaged brightness temperature values from NSIDC, but now we use swath data from co-authors at University of Bremen. The changes are even more remarkable now.**

L332/333: "the ROS covered quite a large region" --> you could make a cross-reference from the comparison of KuKa-radar and CS-2 data to this statement because there you needed to assume that the ROS event covers a comparably large region.

**We already know it covered a large region by looking at the ERA5 animation (not sure the reviewer viewed the animation). Since we say it *further illustrate*s, this seems sufficient given previous reference to the animation and also to the CS2 data. Thus no change is made.**

L345-347: This looks like a perfect place to cite the work of Landy et al., JGR, 2020, https://doi.org/10.1029/2019JC015820

**We added the reference.**

L348-352: "Kuka data combined ... scale satellite footprints." --> This sounds all very good and reasonable - alone: the experiment which results you showed and discussed here appears not be optimally suited to follow that path yet. You could clarify this in the text and suggest what needs to be improved.

**Yes this refers to possible future approaches and we have modified the text to clarify this. We now state**:

*"While there are challenges in upscaling to satellite footprints, KuKa data combined with measurements of physical snow characteristics could, in future studies, allow us to investigate how the combination of snow depth, temperature, salinity and radar-scale roughness control sigma0, and how these facet-scale factors affect the shape return pulse, as a function of transmitted pulse bandwidth. These insights can then be combined with numerical radar simulation approaches that focus on large-scale floe topographic variability [e.g. Landy et al., 2020], with realistic gain patterns, to understand how the different factors combine to influence the radar return power over km scale satellite footprints."*

L353-354: The sentence refering to Laxon, 1994 should perhaps include the notion that leads actually INCREASE the peakiness because of their specular return. This would make it easier to understand why you then move over to water on top of the sea ice in form of melt ponds, also causing specular returns and hence an increase in pulse peakiness before you then come to your results representing kind of a melt-pond precurser: wet or more or less saturated snow triggered here by ROS events instead of summer melt.

**We have specified that leads result in peakier waveforms to explain this.**

L363: "the introduction ..." --> Perhaps better: "by adding rainwater to an existing snow cover its density and also its SWE is increased"

**Changed as suggested**

Figure 10:

Typo in caption "teh" --> "the"

**Corrected**

L431: "Appendix" --> please refer to the specific figure(s) in the appendix.

**Confusing comment as we do refer to each figure in the appendix in the appendix text.**

---

## Author Comment (AC2)

**Summary**

The authors use a time series of atmospheric and surface geophysical observations to document the effect of rain-on-snow events on passive and active microwave remote sensing emission, backscatter, and radar waveform. The examined passive microwave frequencies (19 and 89 GHz) focus on two of the commonly used in sea ice concentration retrieval algorithms, and the active data are at the Ku- and Ka-bands found in current and planned radar altimeter missions used to infer sea ice thickness from sea ice buoyancy. The authors highlight the strong effect that a ROS event has on these remote sensing signals, and how changes in snow structure caused by ROS events are pervasive in their impact on passive emission and radar waveforms. They argue that there is an increase in ROS events on sea ice and that the topic is under-studied in that the community does not understand how these events contribute to sea ice geophysical retrieval errors. Data used are from the large, multidisciplinary, MOSAiC drift campaign that took place in the central Arctic from 2019-2020. The paper focuses specifically on data collected in late August and early September 2020. The paper is original and relevant to TC, and it should be of interest to the sea ice and snow readership.

**The authors appreciate the reviewer's constructive comments to the paper. Following are our detailed responses to your comments.**

Major and minor comments are as follows.

**Major**

(1) Speculation about emission and scattering mechanisms: The authors use a large volume of data to document the rain event and the changes in snow properties that occurred during and after it. The MOSAiC project affords this opportunity, and the authors should be lauded for putting together such a detailed picture of the event as it happened. The impact of the event on remote sensing signals is well documented. However, despite the effort to incorporate so much detail, many statements made about the connections between snow property and microwave emission/backscatter/waveform are speculative. Examples include: on lines 242-245, where the downward percolation of water in snow is "likely" attributed to a 12-15dB decrease in backscatter; and lines 246-250, where snow porosity is related to an increase volume scattering, yet porosity isn't examined and the authors express the need for more analysis. These speculative comments are not well enough substantiated by the data at hand, or by microwave scattering and emission theory and/or a modelling framework. While it is understandable that there isn't a lot of well-established microwave interaction theory dealing with such a complex scenario, there are still basic principles that would help drive the interpretation.

We agree with the reviewer that some speculative statements are present in regards to backscatter and emission mechanisms. Some of these have already been addressed in our responses to the first review. We updated Figure 4 to highlight the percolation pathway of liquid water (see figure below). Note too that the relationship between porosity and density is reciprocal and we chose to show the density in the paper (and thus also indirectly the porosity).

We have also made the following edits:

[revised manuscript text omitted]

For example, does it makes sense that the drainage of the absorbing water during the second rain event should lead to such a dramatic backscatter decline? Isn't the snow being wetted by absorbing rain?

After analysing the micro-CT images, we found water percolation paths (see darker pink marks in revised Figure 4) after the melt onset and during the second rain event and the snowpack transitions to a funicular regime (i.e. liquid water occupies continuous pathways through the snow pore spaces. This results in downward percolation of liquid water via gravity drainage [Colbeck, 1982a; Denoth, 1999], resulting in a completely saturated snowpack. This likely leads to the large backscatter decline during the second rain event. See above modifications to the text.

---

## Author Response (AR2)

General Comments:
I have no general remarks. You improve the manuscript substantially, discussed most of my points sufficiently well, and I do not see an added value to insist on a more elaborated discussion whether the ROS event discussed in your manuscript is representative for winter conditions or not. You noted that you don't see the point here because the "official" start of winter in October was just two weeks away. But this was not my point. I was referring to winter conditions in general, and more the mid- to end-of-winter conditions - aka January through April - where snow thickness is larger, where ice is colder and where air-temperatures are colder as well. But fine. Readers can judge by themselves whether this ROS can be seen as a representative case of "typical" winter-time ROS events or not.

**We thank the reviewer for their thorough comments. We have responded and made the suggested edits (see below).**

Specific Comments:
L139/140: "Note, the 37 GHz ..." and your reply to my respective comment in the 1st review --> Yes, I do and already did believe this. I was just wondering whether you would specify what you mean by unstable, e.g. "erratic TB fluctuations of the order of 50 K or even more during observations periods when the other two frequencies revleard TB variation less than 5 K". But you decided to not let the reader know this detail. Fair enough.
**We didn't feel it was necessary to show the figure of how the 37 GHz channel behaved. However, we now state the following to satisfy the reviewer.**
***Note, the 37 GHz channel was unfortunately unstable, showing at times unphysical fluctuations of more than 50K and thus we only focus on results at 19 and 89 GHz.***

Figure 4, caption, Line 5: At least I have difficulties to clearly separate darker pink from lighter pink and have (still) difficulties to see the 10 mm (only) thick refrozen icy layer by means of the level of pink. But I note the peaks in density at around 10 mm and 30 mm for the RS site on Sep. 15 and at 35 mm at the FLUX site.
**We changed the light pink to purple and updated the plot and descriptions. We hope this allows the reviewer to more clearly see the dense layer compared to the percolation paths and the SSL.**

L252: "Given that surface conditions were generally similar across the floe" --> As for the vertical ordering of the snow layers possibly yes but not when it comes to the snow depth. But again, fair enough.
**Yes there are slight fluctuations in snow depth as expected from snow drifts, but it is the conditions that is more important here.**

L349-351: Another source of typical TBS observed over sea ice could be sea-ice concentration retrieval papers. Even though the values reported therein might contain a small residual weather influence these are another helpful source of information against which your measurements can be backed up. One such paper would be the one by Ivanova et al., in The Cryosphere, 9(5), 2015, where the Appendix A lists typical values for freezing season conditions. There is also a paper by Ivanova et a., 2014 in Transactions on Geoscience and Remote Sensing on sea ice concentration algorithms. **Thanks for pointing out the values in the Appendix from Ivanova et al. (2015). It is encouraging that our values do match theirs quite well. We have added that reference to satisfy the reviewer.**

L444: Not sure why Kaleschke et al. (2016) is listed here as this is about SMOS based thin ice thickness retrieval, hence L-Band, a frequency you did not include. Since the "main playground" for the SMOS thin ice thickness retrieval is FYI with thickness values below about 70 cm with potentially little snow easily penetrated by the long L-Band waves the relevance of your study on SMOS/SMAP based thin ice thickness retrieval can be questioned. **We listed the L-band in the implications section as a discussion point. It is true we do not evaluate the L-band instrument that was also deployed during MOSAiC, as wet snow may also impact this frequency. With an L-band wavelength of 21 cm, a refrozen ice layer should have a depth of similar order to show a scattering effect. However, the snow densification changes the dielectric contrast at the snow/ice interface and this could still have an impact. Again, this is about potential implications and something that the PIs for the L-band instruments could explore further.**

L465/466: "(GR37/19 shown in Figure 10(c))" --> I see that you did not consider my suggestion to also show the PD19 or (as it is very simple to compute) the PR19 based on the values shown in Figure 5. As you may know, the tie point triangle in PR19 - GR3719 space of the NT algorithm highlights very well that the main contribution to the sea ice concentration derived using the NT algorithm comes from the PR19 which is - as you state in your comments to my 1st review - only a normalized version of the PD19. I am sure, that plotting the PD19 alongside the NT sea ice concentration would be an eye-opener. But, fair enough, you did not see it that way. **We show in Figure 10 the PD at 89 GHz and the GR for 19/7 and 37/19. While we could show the PD at 19 GHz, it can already be inferred from Figure 9 and Figure 5 and we do mention some differences in values in the text. We agree that it would also highlight the NT algorithm dependence, but we already have so many figures we tried to keep things succinct. We do plan in a follow-on study to evaluate discrete events detected in reanalysis to get more insights into satellite retrievals. We would be happy to collaborate with the reviewer on such an effort.**

L472: "over both FYI and MYI" --> I have to admit that I am not overly happy with generally stating that the Rostosky et al. algorithm is THE solution for snow depth on MYI. It has not been evaluated adequately for most of the freezing season (October through March) and it has been developed / trained with data of the same month (April) that is also then used for the evaluation. It is a first step into the correct direction but it is not the solution. Anyways, the interesting piece of information here is that Figure 10 shows that the snow thickness based on GR197 is as large after the ROS as it has been before it - which is an improvement compared to using GR3719.

**We did not intend to imply the Rostosky et al. algorithm is the solution for snow depth on MYI...we are simply using it as an example, we are not advocating any particular algorithm over another. The point is simply to show a ROS event can impact these retrieval algorithms.**

L477: What I again still do not support is that you do not comment at all to the fact that the snow thickness values retrieved with both approaches are considerably higher than those so representative MOSAiC floe snow thickness values. To me this clearly suggestions that both snow thickness products are biased high because of the considerable fraction of non-seasonal ice in the region the MOSAiC floe was located. I am missing a statement clarifying this in your manuscript.

**Again, we were not validating any of the snow depth algorithms, but we've added some statements regarding your two concerns about the snow products, in particular we have added: *"Note that the retrieved snow depth is substantially higher than the snow depth measured on the MOSAiC flow. Both algorithms are not designed to retrieve snow depth during the refreezing season. Sea ice that survived one summer melt is interpreted as deep snow by the algorithms and thus snow depth is overestimated."***

L481/482: "they would ... and refreezing" --> I suggest that you add the information that you were rightly providing also in your comments to my 1st review. Therein you stated that during winter with presumably thicker snow loads and colder snow surface and ice/snow interface temperatures the effect of these ROS / warm air intrusions might affect a smaller vertical portion of the snow cover. Percolation into the snow might not be as deep as during your example shown here on the one hand. But on the other hand, ice layers forming at the top of a deeper snow layer might bias altimeter measurements even more in winter than during your case. I just invite you to consider providing this information to readership and not just us reviewers (and those who eventually have the time to look through discussions of submitted manuscripts).

**Ok, we added your above suggestion near lines 481. Specifically we write: *"A key difference with presumably thicker snowpacks and colder snow and snow/ice***

*interfaces in winter could be that ROS or warm air intrusions may affect a smaller vertical portion of the snow cover. Specifically, percolation into the snow might not be as deep, which once refrozen, would lead to larger biases in ice thickness retrieved from radar altimeter than during the case study shown in this paper."*

Typos / editoral comments:
L131/132: "Physical temperatures were made ..." --> Please check sentence; seems as if a word is missing.
**We don't quite understand what the reviewer means, the sentence already reads as *"Physical temperatures were made of the absorber pads and the ambient air temperature*"**

L177: "during summer" --> you could add "and (early) fall"
**Done**

L227: "the the" --> "the"
**Done**

L229: "additiona" --> "additional"
**Done**

L235/236: the 2nd "including the SSL" can be deleted.
**Done**

L237 and elsewhere: I am wondering whether you would consider to change "flux" to "FLUX" when speaking of the respective site.
**Good suggestion, done.**

Figure 5 caption. I suggest to add: "Note the difference of one order in magnitude in the kernel density distributions between pre and post ROS event."
**Good suggestion, Done**

L360: "GHz" --> "K", same in L362.
**Thanks for catching that. Corrected!**

L470: "increase in the MYI fraction" --> add: "and in the snow depth retrieved using the Markus and Cavalieri algorithm."
**Done**

Figure 10: Please use (a) to (d) to denote the panels in the figure itself and also in the caption to be consistent to your text.

**Done**

L634: "microcontroller.As" --> "microcontroller. As"
**DOne**